# Tropical climate-vegetation-fire relationships: multivariate evaluation of the land surface model JSBACH

Gitta Lasslop[1,2], Thomas Moeller[1], Donatella D'Onofrio[3], Stijn Hantson[4], and Silvia Kloster[1]

[1]Max-Planck Insitute for Meteorology, Bundesstraße 53, 20146 Hamburg, Germany
[2]Senckenberg Biodiversity and Climate Research Centre, Senckenberganlage 25, 60325 Frankfurt am Main, Germany
[3]Institute of Atmospheric Sciences and Climate (ISAC-CNR), Torino, Italy
[4]Karlsruhe Institute of Technology, Institute of Meteorology and Climate research, Atmospheric Environmental Research, 82467 Garmisch-Partenkirchen, Germany

**Correspondence:** Gitta Lasslop (gitta.lasslop@senckenberg.de)

**Abstract.** The interactions between climate, vegetation and fire can strongly influence the future trajectories of vegetation in Earth system models. We evaluate the relationships between tropical climate, vegetation and fire in the global vegetation model JSBACH, using a simple fire scheme and the complex fire model SPITFIRE with the aim to identify potential for model improvement. We use two remote sensing products (based on MODIS and Landsat) in different resolutions to assess the ro-

bustness of the obtained observed relationships. We evaluate the model using a multivariate comparison that allows to focus on the interactions between climate, vegetation and fire and test the influence of land use change on the modelled patterns. Climate-vegetation-fire relationships are known to differ between continents, we therefore perform the analysis for each continent separately.

The observed relationships are similar in the two satellite datasets, but maximum tree cover is reached at higher precipitation

values for coarser resolution. This shows that the spatial scale of models and data needs to be consistent for meaningful comparisons. The model captures the broad spatial patterns with regional differences, which are partly due to the climate forcing derived from an Earth system model. Compared to the simple fire scheme, SPITFIRE strongly improves the spatial pattern of burned area and the distribution of burned area along increasing precipitation. The correlation between precipitation and tree cover is higher in the observations than in the largely climate driven vegetation model, with both fire models. The multivariate

comparison identifies an excessive tree cover in low precipitation areas and a too strong relationship between high fire occurrence and low tree cover for the complex fire model. We therefore suggest that drought effects on tree cover and the impact of burned area on tree cover or the adaptation of trees to fire can be improved.

The observed variation of the relationship between precipitation and maximum tree cover between continents is higher than the simulated one. Land use contributes to the intercontinental differences in fire regimes with SPITFIRE and strongly overprints

the modelled multimodality of tree cover with SPITFIRE.

The multivariate model-data comparison used here has several advantages: it improves the attribution of model-data mismatches to model processes, it reduces the impact of biases in the meteorological forcing on the evaluation and it allows to evaluate not only a specific target variable but also the interactions.

## 1 Introduction

Capturing the interactions of vegetation cover and composition with the climatic drivers and related disturbances in Earth system models is crucial to reliably estimate changes of vegetation for a changing climate. Climate is the main driver of global
vegetation patterns, but also vegetation has crucial impacts on the Earth system, due to its influence on the surface albedo and the water cycle (Bonan, 2008; Brovkin et al., 2009). The importance of the vegetation type has been assessed in various studies: when compared to grasslands, forests in tropical areas cool the climate due to higher evapotranspiration, while in boreal regions, forests warm the climate due to a reduction of the albedo (Bathiany et al., 2010). The relevance of vegetation for the climate also shows when contrasting vegetated and non-vegetated surfaces: in the Sahel region this difference is of major
importance for the climatic conditions (Brovkin et al., 1998).

Interactions between vegetation, fire and climate are particularly important to understand the spatial patterns in tropical vegetation, which is characterized by strong gradients from deserts to tropical rainforests. Remotely sensed tropical tree cover shows a bimodality between forest (T>60%) and savanna (T<60%) states for grid cells with similar climate. Intermediate tree cover fractions (e.g. 60%) are virtually absent (Hirota et al., 2011; Staver et al., 2011b). The occurrence of this "gap" in tree cover was
suggested to be caused by a feedback between fire and vegetation. Although the reliability of remotely sensed tree cover data sets to diagnose this "gap" was recently questioned (Gerard et al., 2017), the bimodality in the distribution is also confirmed by canopy height (Xu et al., 2016) or biomass (Yin et al., 2014). The occurrence of both, forest and savanna states, under similar climatic conditions due to a feedback between fire and vegetation is supported by conceptual (Staver et al., 2011a) and process-based models (Higgins and Scheiter, 2012; Moncrieff et al., 2014; Lasslop et al., 2016).

While data analysis can provide insights on driving factors for certain variables, process-based models summarize the process understanding and allow us to perform experiments that are impossible in reality. Dynamic global vegetation models (DGVMs) were developed to understand ecosystem dynamics, the carbon cycle and biosphere-atmosphere interactions (Sitch et al., 2003). Many of them are part of Earth system models (ESMs), to represent the dynamics of the land surface within the climate system. It is therefore important that DGVMs include appropriate representations of vegetation to obtain reliable simulations of the
Earth system.

The development of remotely sensed global burned area products facilitated the implementation and evaluation of complex fire models within DGVMs (Hantson et al., 2016). Over the recent years these models were applied to address the impact of fire on the carbon cycle (Li et al., 2014; Yue et al., 2016), the land surface temperature (Li et al., 2017) or the sensitivity of the fire model to driving factors (Kloster et al., 2010; Lasslop and Kloster, 2015). Evaluation of fire models mostly focused
on evaluating the burned area and carbon emissions, but also the importance of benchmarking the effects of fire on vegetation has been noted (Hantson et al., 2016) and applied in model development studies (Kelley et al., 2013). The evaluation, however, is based on comparing variables one by one and not the relationships between them. Baudena et al. (2015) go beyond the geographic comparison by analyzing the relationship between tree cover and the main climatic driver (precipitation). The

relationship between climate and fire was also evaluated in previous studies (Prentice et al., 2011). However, to our knowledge, climate, vegetation and fire have not been combined in a multivariate model-observation comparison.

Here, we aim 1) to assess the robustness of observed climate-vegetation-fire relationships across the tropical continents based on two remotely sensed tree cover datasets; 2) to test a multivariate model evaluation to identify opportunities for model im-
provements in JSBACH, the vegetation model used within the MPI Earth system model, and 3) to test the contribution of land use change on the obtained relationships.

## 2    Model and Data

To investigate the climate-fire-vegetation relationships in the tropical regions, we represent climate by the mean annual precipitation (P), vegetation by the tree (TC), grass (GC) and non-vegetated cover and fire as the burned fraction (BF).
We define the tropical region as between -30° and 30° latitude. As continental limits we chose -20° to 60° longitude and -30° to 30° latitude for Africa, -130° to -30° longitude and -30° to 30° latitude for South America, 60° to 160° longitude and -10° to 30° latitude for Asia and 100° to 160° longitude and -30° to -10° latitude for Australia.

### 2.1    Model and simulation description

We use the JSBACH land surface model (Reick et al., 2013), which is the land component of the MPI Earth system model
(MPI-ESM) (Giorgetta et al., 2013). JSBACH simulates the terrestrial carbon and water cycle in a process based way. We use two fire algorithms, a simple empirical model (Brovkin et al., 2009; Reick et al., 2013) and the process-based fire model SPITFIRE (Lasslop et al., 2014; Thonicke et al., 2010). Results referring to simulations with the complex SPITFIRE model are referred to as JSBACH-SPITFIRE, simulations with the simple JSBACH standard fire scheme are indicated as JSBACH-standard. These two approaches span the range of complexity of currently used global scale fire models (Hantson et al.,
2016). The JSBACH-standard fire computes burned area based on a minimum burned fraction which increases as a function of the litter carbon pools and relative humidity averaged over the last three weeks. It was tuned to yield reasonable global emission estimates (around 2PG carbon) and to improve the tree cover, which is clearly too high without fire. SPITFIRE computes burned area based on human and lightning ignitions, fire spread rate and a fire duration. SPITFIRE distinguishes between different fuel particle sizes and uses a combination of minimum and maximum temperature, precipitation and soil
moisture to determine the fuel moisture. Both fire models interact with the vegetation model as follows: JSBACH provides fuel amounts, vegetation composition and soil moisture as inputs to the fire model. The fire model in turn reduces the carbon pools of JSBACH according to the simulated carbon combustion of vegetation fires and reduces the cover fractions of burned vegetation. In the JSBACH-standard fire scheme the burned area directly translates into a reduction of the cover fractions of the plant functional types (PFTs) (100% of the cover fractions on burned area are removed). Whereas in SPITFIRE the
mortality of woody vegetation depends on the fire intensity, fire residence time, the vegetation height and bark thickness. The model's plant functional types for the tropics include C3 and C4 grass, tropical evergreen and deciduous trees, and rain green shrubs. Shrubs and trees compete according to their net primary productivity. Grasses and shrubs have an advantage compared

to trees in regions with disturbances due to their lower establishment time scale (Reick et al., 2013, grasses: 1 year, shrubs: 12 years, tropical trees: 30 years). PFTs do not establish if the 5 years running mean net primary productivity (NPP) turns negative. Trees prevail in grid cells without disturbance and positive NPP. Land use is included following the protocol of Hurtt et al. (2011). The implementation is described in detail in (Reick et al., 2013). Croplands are excluded from fire occurrence while pastures are treated as natural grasslands with a higher fuel bulk density within JSBACH-SPITFIRE (Rabin et al., 2017). The JSBACH-standard fire excludes fire occurrence on both anthropogenic land cover types. JSBACH-SPITFIRE shows a reasonable agreement with remotely sensed data products for present day burned area and carbon emissions for simulations with prescribed land cover (Lasslop et al., 2014). The present setup with dynamic biogeography has been evaluated along the human dimensions population density and cropland fraction (Lasslop and Kloster, 2017). The model tends to overestimate burned fraction for high cropland fractions and underestimates burned fraction for very low and high population densities.

### 2.1.1 Simulation setup

JSBACH was forced with meteorological data for the historical period 1850-2005, which was extracted from a coupled simulation with the MPI-ESM version 1.1. For the computation of ignitions the SPITFIRE model additionally uses a population density dataset (Klein Goldewijk, 2001) with decadal resolution and a monthly lightning climatology (LIS/OTD product of the LIS/OTD Science Team, http://ghrc.msfc.nasa.gov). The model's spatial resolution is 1.875° x 1.875°. The time step for plant productivity and hydrology is 30 minutes, while the disturbance routine is called once per day. During the 1000 year spinup period the first 28 years of forcing (1850-1877) were recycled and $CO_2$ concentration fixed at the value of 1850 (284.725 ppm). At the end of the spinup, PFT distribution was largely in equilibrium with only minor shifts between woody PFTs in few grid cells. The subsequent transient historical simulation (Hist) from 1850-2005 accounts for the changes in atmospheric $CO_2$, climate, population density and land use. A complementary simulation accounting only for the rise in atmospheric $CO_2$, transient climate and population density, but using the land use of 1850 for the whole period (cLU) is used to isolate the effect of land use change on the climate-vegetation-fire relationships. When comparing the model output to observations, the averaging period for the model simulations was 1996-2005, as the forcing was only available until 2005.

### 2.2 Datasets for model evaluation

We averaged the remote sensing datasets over the years that were covered by all datasets (2001-2010). Model output is only available until the year 2005. Using only the overlapping period (2001-2005) would decrease the robustness of the mean fire regime and climate characterization. We therefore use different averaging periods for model (1996-2005) and observations (2001-2010). The presentation of the relationship between precipitation, tree cover and burned fraction based on remote sensing data is based on 0.25° resolution and for the comparison with the model the datasets were aggregated to the model resolution (1.875°x1.875°).

### 2.2.1 Vegetation and land cover

We use two tree cover datasets based on satellite data, one based on the MODIS (moderate-resolution imaging spectroradiometer) sensor (Townsend et al., 2011), the other on the Landsat satellite (Hansen et al., 2013). Additionally we use the non-tree vegetation cover and non-vegetation cover of the MOD44B product version 051 (downloaded 6/February 2017, using the R modis package (Mattiuzzi and Detsch, 2018)). The datasets rely on different sensors, however, the algorithms to derive vegetation cover are very similar and the datasets are therefore not completely independent. Nevertheless using the two datasets can give a first insight on the robustness of the investigated patterns.

The maximum tree cover in the MODIS dataset is 80%. This however corresponds to 100% crown cover (Hansen et al., 2003). The modelled cover fractions represent rather the crown cover with a 100% maximum, we therefore linearly rescaled the tree cover data to improve the consistency between model and observations. The second dataset based on Landsat data builds on a high spatial resolution of 30m (Hansen et al., 2013). The dataset provides annual forest gain and loss over the period from 2000-2012. Alkama and Cescatti (2016) reconstructed the annual tree cover and aggregated the dataset to 0.05° . Here, we used the mean over their reconstructed annual tree cover values from 2001-2010.

The MODIS collection 5 land cover dataset (Friedl et al., 2010) was used to test the influence of shrub lands (open and closed shrub lands), as the tree cover data have a higher uncertainty for shrublands. The filtering was applied on 0.05° spatial resolution. This dataset is distributed by the Land Processes Distributed Active Archive Center (LP DAAC), located at the U.S. Geological Survey (USGS) Earth Resources Observation and Science (EROS) Center (lpdaac.usgs.gov), distributed in netCDF format by the Integrated Climate Data Center (ICDC, http://icdc.cen.uni-hamburg.de) University of Hamburg, Hamburg, Germany in 0.05° spatial resolution and annual time step.

### 2.2.2 Fire

The global fire emissions database (GFED, http://www.globalfiredata.org/) provides globally gridded monthly burned area based on the MODIS sensor. We used the version 4 of the dataset (Giglio et al., 2013).

### 2.2.3 Precipitation

The "TRMM and Other Data Precipitation Data Set" (TMPA) is based on the Version 7 TRMM Multi-satellite Precipitation Analysis algorithm (Huffman et al., 2007, 2010). The product has near global coverage from 50° north to 50° south. The precipitation estimate (including rain, drizzle, snow, graupel and hail) is based on a combination of multiple data sources including precipitation gauges. The dataset is available online (http://disc.sci.gsfc.nasa.gov/gesNews/trmm_v7_multisat_precip). For an evaluation of the climate forcing, e.g. the precipitation seasonality, we use the daily TMPA dataset (Savtchenko and Greenbelt, 2016).

### 2.2.4 Other climate parameters

We used the shortwave radiation and temperature of the CRU-NCEP v5 dataset reanalysis (Wei et al., 2014), which is commonly used as observation based model forcing dataset (Rabin et al., 2017), to investigate whether biases in the climate forcing might explain biases in modelled tree cover. We compute the correlation between the difference in modelled and observed tree cover and MPI-ESM and CRU-NCEP for shortwave radiation and temperature.

### 2.3 Quantile regression

We use quantile regressions to characterize the relationship between precipitation and maximum tree cover. The quantile regressions were computed with the R package quantreg (Koenker, 2018). We use the local quantile regression to characterize the shape of the increase in maxmimum tree cover for increasing precipitation. Moreover we quantify the deviation from a linear increase by also including the linear quantile regression. Both regressions were computed for the 0.9 quantile. For the local quantile regression the bandwidth parameter was set to 300 and the number of points where the function was estimated was set to 10.

## 3 Results

We first give an overview over the geographical distribution of the used observation and model output datasets. The comparison of geographical patterns is an important assessment of model performance, it is however difficult to assess whether the interactions between precipitation, fire and tree cover are well captured. Moreover as the JSBACH model is usually used as a land surface model for the MPI-ESM and therefore also here forced with MPI-ESM output, biases in model forcing can cause geographical biases of vegetation and fire variables even with a perfect fire and vegetation model. To reduce the influence of biases in forcing data on the model-data comparison and allow to more closely evaluate the interactions between model components we propose a multivariate evaluation of climate-fire-vegetation relationships. We assess the robustness of observed relationships for two tree cover datasets and two spatial resolutions and compare them to the model simulations. The last paragraph of this section adresses the influence of land use change on the simulated relationships.

### 3.1 Spatial distribution of vegetation cover, area burnt and precipitation in the tropics

The two observational satellite based tree cover datasets are consistent and show only small differences in their spatial pattern (Figure 1a). The overall clear pattern in tree cover is a transition from very high tree cover in moist rain forest regions to low tree cover in the drier savannas to the absence of trees in the desert regions. Both models reproduce this overall observed pattern, although with marked local differences. Both model versions overestimate tree cover in northern Australia to a similar extent. In the North-Eastern Amazon region the simulations underestimate tree cover compared to the observations. This underestimation is much smaller for JSBACH-SPITFIRE. The simulations overestimate tree cover in Southern Hemisphere Africa, this overestimation is again smaller for JSBACH-SPITFIRE. The simulated grass cover has higher maximum values,

but generally is often lower than observed by satellite (Figure 1 d). The spatial distribution of the non-vegetated fraction is captured well by in the model simulations (Figure 1 e).

Generally JSBACH-standard strongly underestimates the total area burnt and the spatial variability (Figure 1 b). JSBACH-SPITFIRE improves the capability to represent fire regimes with high fire occurrences. The tropical average burned area per year is for JSBACH-standard 65 Mha, for JSBACH-SPITFIRE 242 Mha and for the satellite dataset 315 Mha. In South America spatial patterns in JSBACH-standard are inconsistent with the observations (most burning in the Northeast). JSBACH-SPITFIRE overestimates fire occurrence in South America but the spatial patterns are more similar to observations. In Africa we find reasonable agreement between JSBACH-SPITFIRE and the observations. JSBACH-standard shows a strong underestimation of the burned fraction (max. 10% of the grid cell area year$^{-1}$, while the observations show up to 100%). In Australia JSBACH-SPITFIRE and JSBACH-standard show similar patterns and both strongly underestimate the burned fraction.

Precipitation of the MPI-ESM forcing shows a dry bias in the East and central Amazon region, a dry bias in Asia, and moister conditions in the western part of southern hemisphere Africa (Figure 1 c). The dry bias in South America and Asia is known from previous ECHAM model versions (Hagemann et al., 2013; Stevens et al., 2013). The dry bias in precipiation in the Amazon may for instance explain the high bias in burned fraction in that region.

## 3.2 Climate-fire-vegetation relationships: comparison of observation datasets

Maximum tree cover shows an increase along the precipitation gradient across all continents, with trees being absent until a certain threshold (300-500 mm year$^{-1}$), increasing maximum tree cover and saturation of maximum tree cover for high precipitation (between 1500 and 2000 mm year$^{-1}$). The two remotely sensed tree cover datasets are consistent in their variation along the precipitation gradient (Figure 2). Fire occurrence is much higher for the African and Australian continent compared to South America and Asia. Burned fraction increases with increasing precipitation until around 1000mm mean annual precipitation, due to the increasing availability of fuels. For tree cover fractions higher than 0.8, fire is virtually absent. Beyond this distinction there is no visually clear increase in burned fraction for decreasing tree cover at a given precipitation value. The Spearman rank correlation between burned fraction and tree cover for grid cells with mean annual precipitation higher than 1000mm and tree cover lower than 0.8 is, however, significant for both datasets in the 0.25° resolution, in the model resolution only the correlation with the MODIS dataset is significant. This correlation is much stronger for the MODIS tree cover compared to the LANDSAT tree cover (Table 1). For Australia and Africa fire occurrence is very low below a mean annual precipitation of 300 mm year$^{-1}$, for South America and Asia already below 500 mm year$^{-1}$.

The Spearman rank correlation between precipitation and tree cover is very similar for both tree cover datasets (Table 1). The statistical precipitation thresholds for low (but higher than 0) and high tree cover differ by less than 100 mm. The aggregation to the model resolution shows the strongest effect on the precipitation threshold for high tree cover and shifts this value to higher precipitation. The association between precipitation and burned area is less sensitive to the aggregation: 80% of the global burned area occurs in regions with precipitation between 609 and 1518 mm on 0.25° resolution and between 635 and

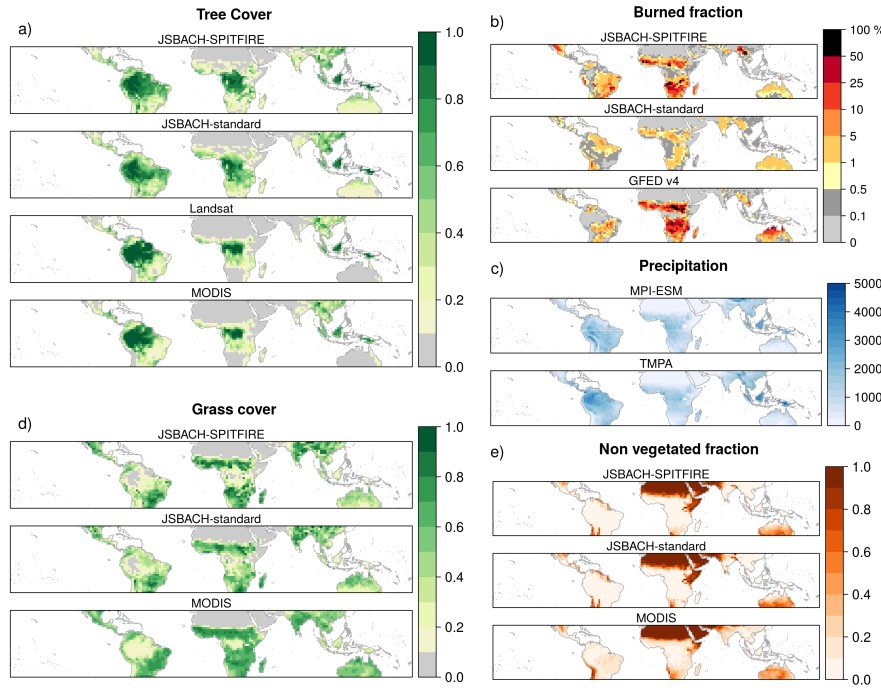

**Figure 1.** Spatial distribution of modelled and observed datasets used in this study. (a): Spatial distribution of tree cover fraction over the global tropics for the JSBACH-SPITFIRE and JSBACH-standard model simulation and the satellite data products from Landsat and MODIS. (b): Burned fraction [year$^{-1}$] as modeled by JSBACH-SPITFIRE and JSBACH-standard and the GFED v4 satellite product. (c): Precipitation in mm year$^{-1}$ of the MPI-ESM and the TMPA dataset. (d): Grass cover fraction, and (e): non-vegetated fraction of the grid cell for the models and the MODIS satellite product. All datasets were remapped to the 1.875° model resolution.

1495 mm in 1.875° resolution.

## 3.3 Climate-fire-vegetation relationships: Evaluation of model results

In the tropics, the observed burned area is strongly constrained by precipitation, around 80% of the burned area is observed
5 in regions with mean annual precipitation between 600 and 1500 mm year$^{-1}$ (Table 1). This precipitation range is slightly larger for the model simulations (Table 1). JSBACH-SPITFIRE reproduces the increase in burned area for low precipitation, but slightly overestimates the contribution of grid cells with precipitation higher than ca. 1300 mm year$^{-1}$ to the total burned area (Figure 3). JSBACH-standard overestimates the contribution of areas with low precipitation, but agrees well on the contribution of areas with high precipitation (>1300 mm year$^{-1}$) when compared to the satellite observations. Fire occurrence is
10 limited in regions with low precipitation due to low fuel availability (Krawchuk and Moritz, 2011). This low fire occurrence is well reproduced by JSBACH-SPITFIRE and for most continents also by JSBACH-standard with the exception of Australia

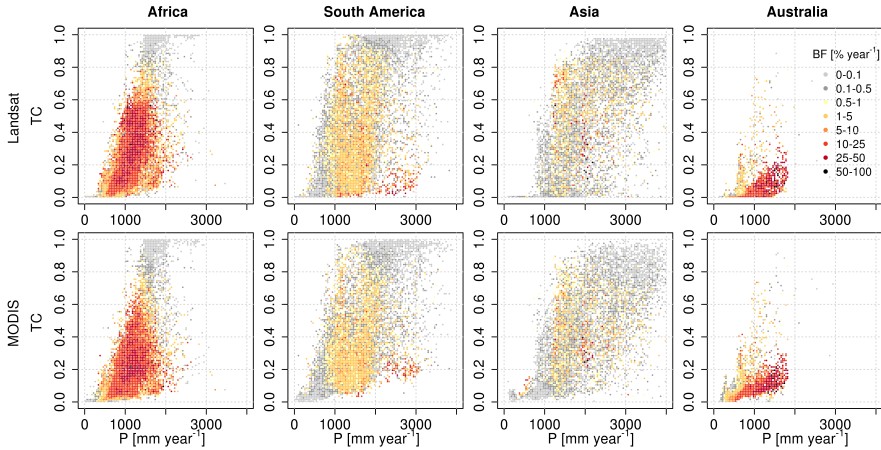

**Figure 2.** Tree cover (TC) versus precipitation [mm year$^{-1}$] with color coded burned fraction (BF) for different continents for the two satellite datasets. Burned area is averaged over data points with the same precipitation (40 mm steps) and tree cover (in steps of 0.01) to avoid over-plotting based on a spatial resolution of 0.25°. For Asia some higher precipitation values were cut off.

where the burned fraction of JSBACH-standard shows almost no variability (Figure 4).

Surprisingly the observations show a higher Spearman correlation between tree cover and precipitation than the models (Table 1). The lower correlation of the modelled relationship most likely originates from the lower precipitation regions (< 500 mm year$^{-1}$), where the maximum tree cover is very low in the observations and both models strongly overestimate the maximum

tree cover (Figure 4).

Models and observations generally agree on the absence of fire for very high tree cover (>0.8) and on the decrease of burned fraction for mean annual precipitation decreasing below 1000mm. However for regions with tree cover < 0.8 and mean annual precipitation > 1000mm we find strong differences. JSBACH-SPITFIRE shows a strong negative Spearman rank correlation between burned fraction and tree cover, the observations show a weaker negative correlation, and JSBACH-standard shows

a positive correlation (Table 1). This can also be seen in Figure 4 where for the JSBACH-SPITFIRE simulation the highest burned fractions (> 50% of grid cells year$^{-1}$) are found in Africa for the lowest tree covers (0.1) and for precipitation between 1000-2000 mm year$^{-1}$. JSBACH-standard in many grid cells shows low fire occurrence for low tree cover, especially for South America (Figure 4), these grid cells have a high fraction of crops or pasture, which both are excluded from burning in JSBACH-standard (in SPITFIRE only crops are excluded). The observations (also Figure 4) show highest values of the burned

fraction for tree cover values up to 0.3 for MODIS and up to 0.5 for LANDSAT.

Burned fraction is much lower in Asia and South America compared to Australia and Africa in the observations. Both models show an underestimation of the fire occurrence in Australia. SPITFIRE reproduces the fire regime with high annual burned fraction in Africa. In JSBACH-standard the difference in burned fraction between the continents is smaller than in JSBACH-SPITFIRE (Figure 4).

**Table 1.** Spearman rank correlation (R) between precipitation (P) and tree cover (TC), and rank correlation between burned fraction (BF) and TC for data points with mean annual precipitation higher than 1000mm and tree cover less than 0.8. The required precipitation [mm year$^{-1}$] for $0.05 < TC < 0.15$ and $0.85 < TC < 0.95$, estimated as 0.05 quantile of precipitation for grid cells with the specific TC only, and precipitation value [mm year$^{-1}$] where 10% and 90% of the burned area (BA) originates from areas with lower precipitation. For the remote sensing datasets TMPA was used as precipitation, for the simulations (Hist, cLU, and JSBACH-standard) the MPI-ESM precipitation was used. Model results are all in 1.875° resolution. Bold font indicates significance (p-value < 0.05 of the correlation.

| Data | R(P,TC) | R(BF,TC) | 0.05 quantile of P for $0.05 < TC < 0.15$ | 0.05 quantile of P for $0.85 < TC < 0.95$ | 10% of BA has lower P | 90% of BA has lower P |
|---|---|---|---|---|---|---|
| Landsat 0.25° | **0.90** | **-0.05** | 568 | 1417 | | |
| Landsat 1.875° | **0.91** | -0.08 | 569 | 1596 | | |
| MODIS 0.25° | **0.91** | **-0.26** | 425 | 1514 | | |
| MODIS 1.875° | **0.93** | **-0.4** | 462 | 1644 | | |
| GFED v4 0.25° | | | | | 607 | 1517 |
| GFED v4 1.875° | | | | | 635 | 1489 |
| JSBACH-SPITFIRE Hist | **0.79** | **-0.5** | 31 | 1268 | 652 | 1663 |
| JSBACH-SPITFIRE cLU | **0.78** | **-0.64** | 13 | 1000 | 700 | 1654 |
| JSBACH-standard | **0.87** | **0.17** | 34 | 1597 | 266 | 1519 |

Models and observations show differences between continents in the relationship between precipitation and maximum tree cover (Figure 5). For Africa, South America and Asia the relationship between maximum tree cover and precipitation shows a saturation for high precipitation. For Australia maximum tree cover increases linearly with increasing precipitation for models and observations, but the precipitation range also does not reach values where a clear saturation is reached for the other con-

5 tinents. For JSBACH-standard the curves are very similar for the different continents. JSBACH-SPITFIRE shows a stronger variation, this must be due to the differences in fire as the model is otherwise the same. The observations show an even stronger variation between continent, with clearly lower tree cover valuse for Australia followed by Asia. For Africa local quantile regression clearly differs from the linear quantile regression for the satellite data, indicating a sigmoid shape, while the other continents show a rather linear increase until the saturation (Figure 5). JSBACH-SPITFIRE reproduces the higher tree cover

10 for South America compared to Africa (albeit the difference is stronger) for mean annual precipitation lower than 1000 mm, but also JSBACH-standard shows a small difference.

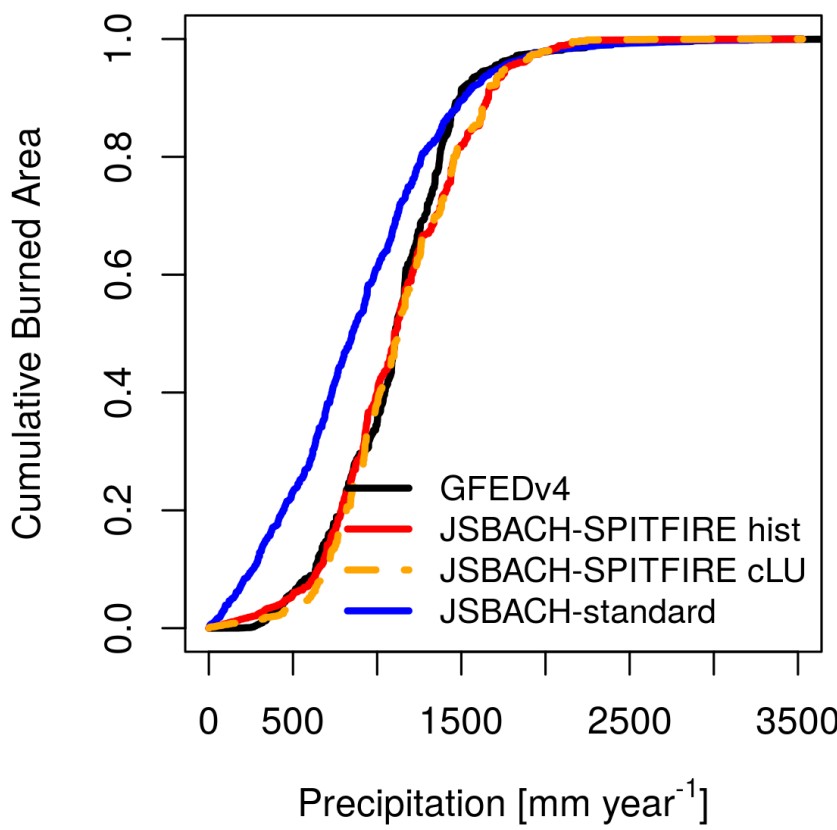

**Figure 3.** Cumulative burned area normalized with the total burned area for increasing precipitation. For the GFEDv4 burned area the TMPA dataset was used, for the model simulations the MPI-ESM precipitation was used.

The grass cover has a much higher variability in the model compared to the MODIS data (Figure 6). The modelled non-vegetated fraction decreases faster with increasing precipitation compared to the observations (Figure 6). The dominance of trees (computed as TC/total vegetation cover) is strongly overestimated in the model for low precipitation (<500 mm year$^{-1}$, Figure 6). While the relationship between precipitation and non-vegetated fraction is similar between the continents, the re-

lationship for grass cover differs (Figure 6). For Australia observations and modelled grass cover increases with increasing precipitation. In Africa, South America and Asia grass cover first increases and then decreases with increasing precipitation.

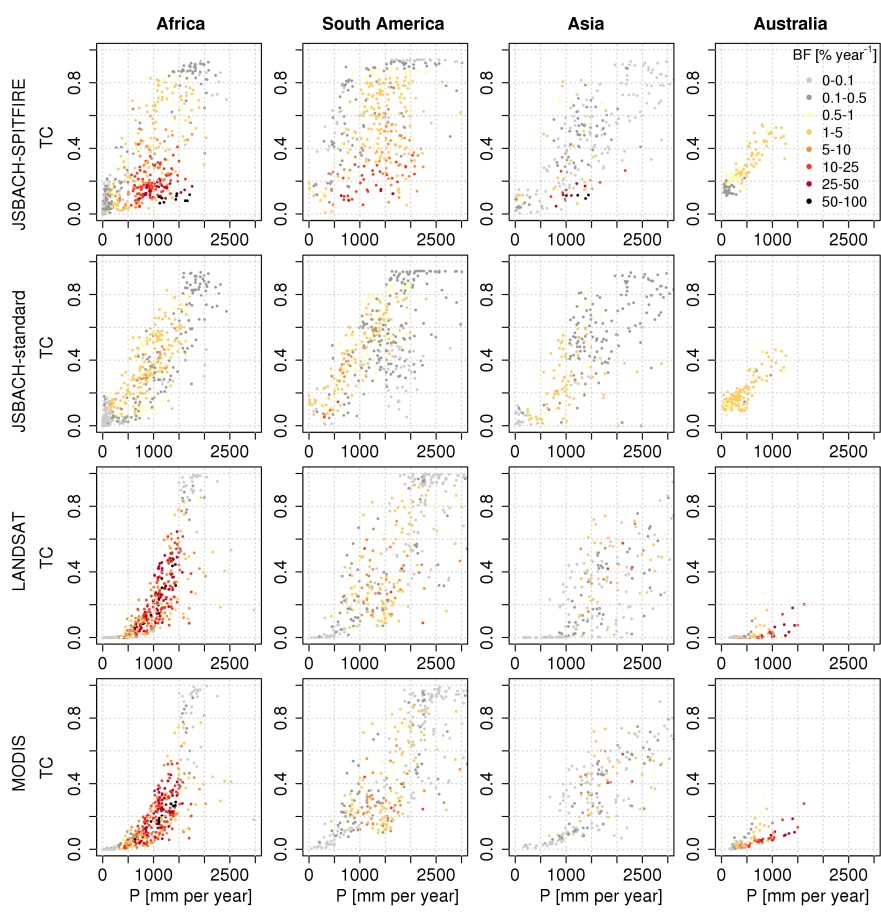

**Figure 4.** Modelled and observed tree cover (TC) versus precipitation (P), color coded burned area fraction (BF). Satellite datasets were aggregated to model grid resolution (1.875°).

## 3.4 Climate-fire-vegetation relationships: Influences of land use change

The simulation with preindustrial land use represents a state with low influence of land use change. The comparison to the historical simulation allows to assess the influence of land use change since 1850. The impact of fire on tree cover, as quantified by the Spearman rank correlation, between burned fraction and tree cover is higher for the simulation with preindustrial land use (Table 1). This indicates that anthropogenic land cover change decreases the impact of fire on the vegetation distribution. Land use change did not affect the rank correlation between precipitation and tree cover. The precipitation range for 80% of the burned area is only slightly narrower for the simulation including land use change (Table 1). Tree cover, however, is even higher for low precipitation and reaches canopy closure for lower precipitation (Table 1 and Figure 7 compared to Figure 4). The simulation with land use of 1850 shows a strong gap between the savanna systems (TC < 40%) and closed forests (TC > 70%)

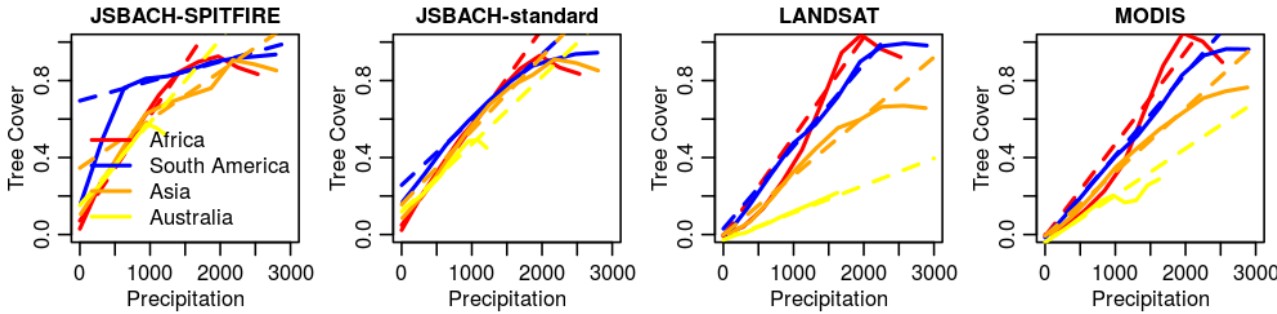

**Figure 5.** Modelled and observed relatioship between precipitation and maximum tree cover based on a linear quantile regression (dashed line) and a local quantile regression (solid line). Different colors indicate the different continents.

for Africa and less strong for South America (Figure 7). For Australia and Asia the simulation does not show this pattern. In the historical simulation land use overprints this gap of the natural vegetation dynamics. The difference in fire occurrence between Africa and South America is smaller for the simulation with preindustrial land use compared to the historical simulation (Figure 7 compared to Figure 4).

## 4 Discussion

The multivariate model-data comparison identified differences and agreements between modelled and observed interactions between fire, vegetation and climate. It goes beyond spatial comparisons by providing better guidance on which processes in the model need improvement. Here we discuss which model improvements can help to address the differences, what causes
agreements in intercontinental differences and whether limitations of the observations might influence our findings.

### 4.1 Opportunities for model improvements

JSBACH overestimates tree cover for low precipitation on all tropical continents. In these dry regions no or only very low burned fractions are observed, and SPITFIRE shows a good response to precipitation while JSBACH-standard already over-estimates the burned area (Figure 3). The improved burned area pattern of SPITFIRE did not lead to an improvement in tree
cover for these dry regions. It is therefore unlikely that further improvements in burned fraction will improve this model-data mismatch for tree cover in dry regions. Satellite data however show that the intensity of fires increases in these regions (Hantson et al., 2017), the impact on vegetation per burned area might therefore be stronger and could support the disappear-ance of trees in these regions. The productivity of vegetation in the JSBACH model depends on the availability of water and is therefore sensitive to drought. The establishment time scale of trees, however, is a constant (30 years for tropical PFTs) and
only if a 5 year average of NPP turns negative, PFTs stop to establish. Other models require a minimum of 100 mm year$^{-1}$ precipitation for sapling establishment (Sitch et al., 2003). The excessive tree cover could be partly improved by improving the

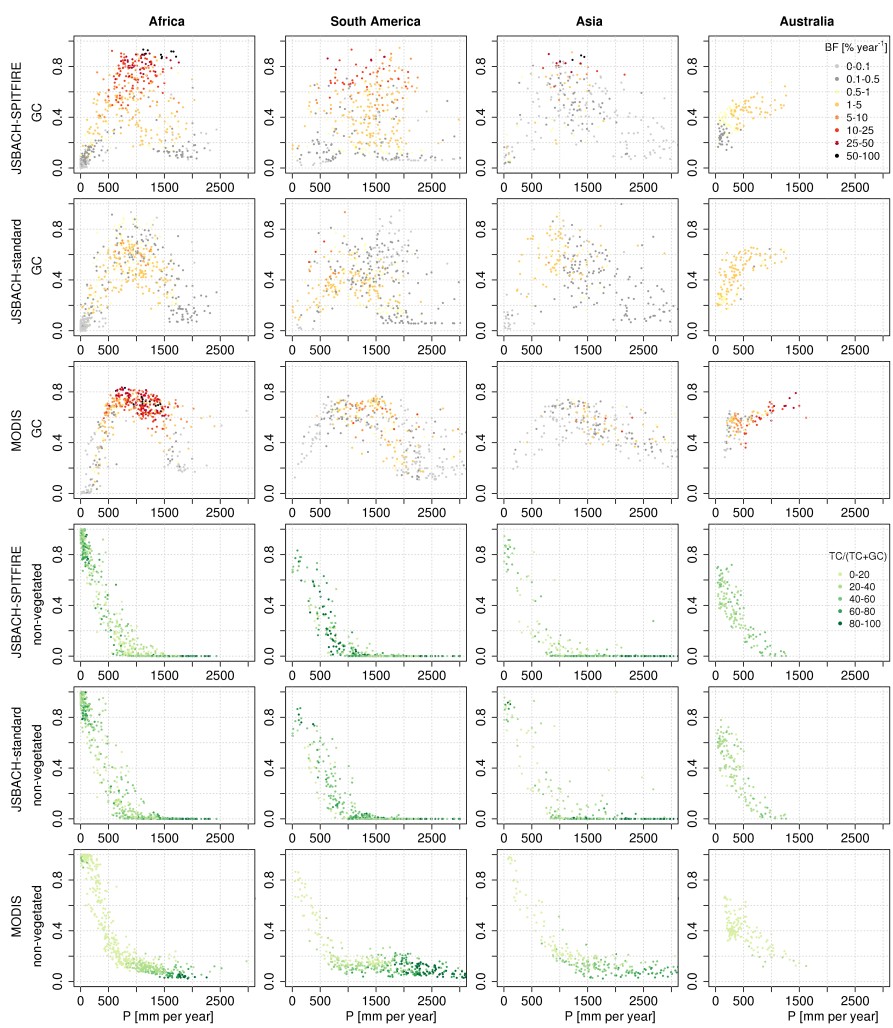

**Figure 6.** Modelled and observed grass cover (GC) and non-vegetated fraction over precipitation (P), with color coded burned area fraction (BF) for the grass cover and dominance of trees as (TC/total vegetation cover) for the non-vegetated fraction.

non-vegetated fraction which decreases too fast with increasing precipitation. In JSBACH, this non-vegetated fraction depends on the productivity of vegetation. Further investigation of effects of the soil moisture memory not only on climate (Hagemann and Stacke, 2015) but also on the vegetation might also lead to useful insights. The excessive dominance of trees (Figure 5) however indicates that also the tree-grass competition is not well represented in the model. Tree-grass competition for water

5 could for example be improved in the model by introducing a sapling stage of trees, which are competitively inferior to grasses (D'Onofrio et al., 2015). Including this mechanism could improve the balance between tree and grass cover, but it could also reduce the establishment rate of trees and therefore, the tree cover in the dry regions with excessive tree cover. Including a PFT-specific rooting depth of vegetation would be an important extension of the model to improve the competition for water

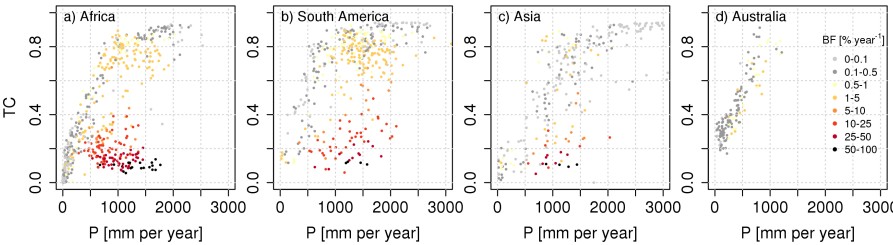

**Figure 7.** Same as Figure 4 for JSBACH-SPITFIRE but with preindustrial land use.

between grasses, saplings and adult trees.

The absence of fire for closed canopies is captured well by JSBACH-SPITFIRE, the modelled strong relationship between higher burned fraction and lower tree cover for open canopies (Figure 4, with the exception of Australia, Table 1), however, is not found in the observations (Figure 2, 4, Table 1). Many general processes determining the savanna-forest boundary

are included in the JSBACH-SPITFIRE model: Increased tree cover leads to a suppression of fire by excluding grasses, higher flammability of grasses leads to increases in fire occurrence with increasing grass biomass (Hoffmann et al., 2012). In JSBACH-SPITFIRE bark thickness is PFT specific and depends on the biomass. Tropical trees are represented by two PFTs one of them has a lower sensitivity to fire due to a higher bark thickness. This is also observed in field studies where savanna species show a higher ratio of bark thickness to stem diameter and are more resistant to fire (Hoffmann et al., 2003). However, the modelled

bark thickness does not adapt to the fire regime as observations indicate (Pellegrini et al., 2017). Kelley et al. (2014) included bark thickness as an adaptive trait in the LPX model, increasing bark thickness for high fire frequencies. This increased and improved the tree cover for Australia. Resprouting is an important plant characteristic that changes the balance between mortality and recovery and also led to an increase in tree cover in fire affected areas in a modelling study (Kelley et al., 2014). A third mechanism to decrease the strong association between high burned area and tree cover could be a negative feedback between

fire occurrence and tree mortality: frequent fire occurrence leads to low fuel loads and low fuel loads allow only low intensity fires with associated lower mortality of trees. In consequence a high burning frequency could lead to lower tree mortality and therefore higher tree cover. This feedback between fire, fuel load, fire intensity and tree mortality is included in the SPITFIRE model. However there is no decrease in fire line intensity with incresing annual burned area (Figure C1). This feedback might therefore be too weak and result in the stronger correlation between burned fraction and tree cover (Table 1).

A more detailed representation of vegetation structure including a sapling state of trees that is more sensitive to fire (e.g. Higgins et al., 2000) and a long-lived adult tree state could also increase the survival of trees. The "fire trap" describes a mechanism where in regions with frequent fires topkill of saplings maintains them in a nonreproductive state (Hoffmann et al., 2009). It explains the importance of the fire free intervals to allow accumulation of sufficient bark to gain sufficient fire resistance. The JSBACH model does not represent the age structure of vegetation, therefore fire always affects the average tree while in reality

only trees that did not accumulate sufficient bark are affected (Hoffmann et al., 2012). Moreover, fire does not influence the tree establishment in JSBACH, it can only lead to mortality. Including a sapling state could therefore increase tree cover in

frequently burned areas, while decreasing tree cover (as described above) in areas that are too dry to provide fuel for frequent burning.

For Australia underestimation of burned area for both fire models is strong (Figure 4). In a previous evaluation where the model was forced with observed climate and vegetation cover was prescribed (in contrast to the dynamic vegetation cover and climate modelled by the MPI-ESM) JSBACH-SPITFIRE showed better results for Australia (Hantson et al., 2015). An improved response of vegetation cover dynamics to precipitation will reduce the underestimation of burned area as in SPITFIRE tree cover and burned area are closely related (Lasslop et al., 2016). Part of the better performance in the previous study might also be due to the use of observed climate forcing.

The rank correlation between precipitation and tree cover is higher for the observations compared to the model outputs (Table 1). One reason might be the lower maximum tree cover for low precipitation in the observations which limits the range of tree cover values in these regions. In JSBACH-standard the correlation between tree cover and precipitation is stronger than in JSBACH-SPITFIRE. In the JSBACH-standard model, fire is only driven by meteorological variables and vegetation properties (which also largely follow climatic gradients). JSBACH-SPITFIRE, however, also uses population density and lightning datasets as input, which are potentially inconsistent with the meteorological forcing derived from the MPI-ESM output. Lightning strikes are strongly related to precipitation (Romps et al., 2014). This decoupling between climate and ignitions might cause the lower correlation for JSBACH-SPITFIRE compared to the JSBACH-standard simulation. For instance in the Northeast Amazon region precipitation of the MPI-ESM is too low, leading to a decrease in tree cover in regions with closed canopy with the JSBACH-standard fire model. The very low ignitions in JSBACH-SPITFIRE in that region contribute to a low fire occurrence compared to JSBACH-standard and in consequence to higher tree cover (Figure 1). Lightning can be computed within climate models (Krause et al., 2014) and using these lightning datasets based on the model not on observations would ensure consistency between meteorological forcing and the ignitions used in the fire model (Felsberg et al., 2018).

The suggested processes are known to be important for the vegetation distribution and it seems plausible that they can help to improve the vegetation distribution. How exactly these plausible modifications would change the patterns of tree cover, fire and their relation to climate likely strongly depends on the exact parameterization and needs to be tested with stepwise model development and factorial simulations.

## 4.2 Difference between continents

We find differences in the climate-vegetation-fire relationships between continents in the satellite products as well as in the model simulations with JSBACH-SPITFIRE and the JSBACH standard model. Differences in the climate-vegetation-fire relationships have been described based on site level datasets (Lehmann et al., 2014). They find that the response of tree basal area to growth conditions (climate and nutrients) and disturbances differs between continents. The study suggests that the one climate–one vegetation paradigm which is an under-pinning of many global vegetation models cannot lead to vegetation patterns that differ between continents under the same climatic conditions as the patterns depend on past environmental conditions and evolution. Evolution is not accounted for in common vegetation models. In simulations with changing climatic forcing,

however, the vegetation is a function of previous environmental conditions and adapts to changes in climate with constant PFT specific time scales. Additionally the human dimension is more and more included in DGVMs, primarily by including anthropogenic land cover change. Moreover, in recent global fire models population density is a commonly used driver for human ignitions and suppression of fires (Hantson et al., 2016).

Our model simulations show that also global vegetation models models can have differences in climate-vegetation-fire relationships between continents. We seperated the effect of land use change by comparing the historical simulation to a simulation with preindustrial land use. We find that land cover change is influencing the differences in the modelled fire regime between Africa and South America. Land cover change influences simulated fire occurrence as cropland areas are excluded from burning and pastures have a higher fuel bulk density in the JSBACH-SPITFIRE model. A reduction in burned area due to increases in croplands is well supported by statistical analysis of satellite data for Africa (Andela and van der Werf, 2014) and globally (Bistinas et al., 2014; Andela et al., 2017). The mechanism behind the reduction in burned area due to croplands is however likely a fragmentation of the landscape, which is not explicitly accounted for in the model. On local scale understanding on these relationships is increasing, for instance the relation between fire and roads (Faivre et al., 2014; Narayanaraj and Wimberly, 2012) or between fire and land management (Morton et al., 2013; Brando et al., 2014). However, a generalization to an approach that would be suitable for global models is still missing.

Vegetation in the MPI Earth system model including SPITFIRE is not only a function of climate but also depends on the history of previous vegetation due to the feedback between fire and vegetation (Lasslop et al., 2016). We did not isolate the effect of the multi-stability in this study but initialized the model with the standard vegetation initialization of the MPI-ESM for the year 1850. The SPITFIRE model also takes into account differences in the fire regime through spatially varying ignitions. In addition to the effect of land use on the differences between continents these spatial differences in ignitions might be important and might explain the smaller differences for the purely climate and land use driven JSBACH-standard model.

The comparison of the increase in maximum tree cover with increasing precipitation shows that although the model shows some variability between continents, it misses a large part of the observed variation. Finding the correct balance of the many influencing factors, e.g. climate, fire, land use, evolutionary differences, will remain a challenge for the future.

## 4.3 Limitations in the comparability between observations and modeled variables

We use two remotely sensed tree cover products, which show coherent patterns. Although these products are derived from imagery with different spectral, temporal and spatial characteristics (MODIS and Landsat), they cannot be considered totally independent because both are derived using a similar classification and regression tree method as well as reference data. The observational tree cover datasets are limited to trees taller than 5 m and do not include shrubs. For the model however we included shrubs and all trees. Previously differences in the threshold where maximum tree cover is reached were attributed to different precipitation datasets and ex- or inclusion of shrub cover (Devine et al., 2017). Filtering modelled and observed tree cover based on the presence of shrubs in the MODIS land cover product leads to only small differences in the relationship between tree cover and precipitation (Figure A1). Excluding grid cells where biomass indicates that the vegetation height is smaller than 5 m according to the allometric relationship used in SPITFIRE-JSBACH (Lasslop et al., 2014) did not lead to

substantially different relationships (Figure A2). Our conclusions are therefore not affected by the limitation of the datasets to observe only trees taller than 5 m.

Compared to the satellite datasets, an African site level dataset shows lower thresholds of precipitation for the absence of trees (ca. 100 mm year$^{-1}$) and for reaching the highest tree cover values (>650 mm year$^{-1}$) (Sankaran et al., 2005). The remote sensing datasets show for Africa an absence of tree cover for precipitation less than ca. 300 mm and canopy closure for 1500 mm year$^{-1}$ in the model resolution (Figure 4). However, the general absence of trees for very low precipitation and increase until a certain threshold is similar to the remote sensing datasets.

The maximum value of a variable can decrease due to spatial averaging. We tested this effect by not using the mean when aggregating the satellite tree cover to the resolution of the precipitation dataset but instead using the maximum value of the underlying 0.05° grid cells of tree cover. Canopy closure can then be reached for all continents for mean precipitation values around 500-1000 mm year$^{-1}$ (Figure A3), which is more consistent with a published site level dataset (Sankaran et al., 2005). This is consistent with the figures in (Hirota et al., 2011) where the MODIS tree cover is shown in 1km resolution. The scale at which maximum tree covers are observed and the spatial scale of the model application therefore needs to be consistent. Moreover, as the thresholds found for the model are closer to the ones found for site-level and high resolution satellite datasets the model performance could improve if the spatial resolution of the model is increased.

Tree cover seems to be a clearly defined variable, but already varies between the two satellite datasets, the MODIS tree cover dataset defines a maximum tree cover of 80%, while the LANDSAT tree cover dataset allows a cover of 100%. In the observations not fully closed canopies due to low foliar biomass might be tracked as a reduced tree cover. In the model, however, tree cover and biomass are two rather independent variables, meaning that tree cover can be high in spite of a low biomass. Biomass datasets might therefore give additional valuable insights and pan-tropical datasets are available (Saatchi et al., 2011; Baccini et al., 2012; Avitabile et al., 2016).

The latest release of the GFED burned area and emissions datasets includes an extension for small fires (Randerson et al., 2012). However these small fires are often related to cropland fires or deforestation fires. Neither of these fire types are modelled explicitly in our model approaches and therefore could cause a mismatch that should not be interpreted as a model bias. Cropland fires are not expected to strongly influence the vegetation cover, while deforestation is prescribed as described in global models and therefore the influence on vegetation cover is considered. Burned area datasets are generally uncertain mainly due to the limited spatial and temporal resolution (Padilla et al., 2015). The difference in global burned area between the dataset including small fires (Randerson et al., 2012) and the one not including small fires is 25%. Missed burned areas due to high cloud cover or vegetation structure certainly also introduces spatial biases. How important such errors are for a comparison as present here is unknown.

By evaluating tree cover and fire for a given mean annual precipitation we account for biases in the MPI-ESM forcing of this parameter. Mean annual precipitation is a strong driver of vegetation patterns especially in the tropics, however other aspects of precipitation and other climatic parameters might be biased and influence our results. Many climate models have problems representing extremes (Sillmann et al., 2013), length of dry periods and tend to generate a permanent drizzle (DeAngelis et al., 2013; Gutowski et al., 2003). Observed rainfall seasonality and number of dry days of the MPI-ESM forcing compares well

between the TMPA observed dataset (see Figure B1), with a small underestimation of seasonality and number of dry days of the MPI-ESM mainly in regions with high rainfall. As we mostly focus on mismatches in regions with low rainfall, the mismatch between observed and MPI-ESM seasonality of rainfall is not concerning. Biases in other climate parameters than precipitation could influence our results, such as temperature or radiation. We therefore explored the correlation between tree
cover biases and biases in the two climate parameters. The biases in mean temperature and shortwave radiation however do not explain any of the variability of tree cover biases, e.g. the correlation is virtually zero and not significant on a 95% significance level (R=0.04, p-value=0.07945 for radiation and R=-0.004, p-value = 0.8842 for temperature). These two parameters were identified previously to explain the impact climate biases on the carbon cycle (Ahlström et al., 2017).

The interactions between climatic parameters are however difficult to disentangle, based on this simple analysis and other ap-
proaches such as multivariate regression or random forrest approaches (Forkel et al., 2017) might help to gain further insights on the effects of specific climate drivers.

## 5   Conclusions

This study combines two satellite datasets with model simulations using a simple and a complex fire algorithm to investigate
relationships between fire, vegetation and climate. Our analysis shows that the two satellite datasets are consistent in terms of the relationship between tree cover, precipitation and fire occurrence, but the spatial scale needs to be considered as some statistical characteristics change with the resolution.

Our analysis showed the strength of the multivariate comparison to detect model inconsistencies and guide model development. It goes beyond the insights gained by standard spatial comparisons. For JSBACH, independent of the fire model used, we find
an overestimation of tree cover for low precipitation where typically fire occurrence is low due to limited fuel availability. The response of burned area to precipitation was captured well for SPITFIRE, but the simple fire scheme showed an overestimation of burned area for dry regions. This indicates that not an improvement of the fire model but improved modelling of drought effects on the vegetation dynamics will improve the response of vegetation to climate in dry regions. Dry regions often show a strong coupling between land and atmosphere (Koster et al., 2006), such an improvement has therefore also a high potential to
improve the performance of the coupled Earth system model.

While fire occurrence and vegetation patterns are well observed by remote sensing, the impact of fire on vegetation is much less constrained by satellite observations limiting the possibilities of evaluating that part of fire models. The multivariate comparison revealed a too strong impact of fire on tree cover for gridcells with very high fire occurrence, which leads to too low tree cover. To boost the tree cover in exactly these regions with high fire occurrence possible model modifications are an adaptation
of trees to fire, by increasing bark thickness in reponse to high fire frequencies, or a stronger negative feedback between fire occurrence and fuel load. This stronger feedback should then reduce fire intensity and consequently fire mortality.

The complex fire model SPITFIRE improves the difference in fire regimes between the continents, especially Africa and South America, compared to the simple fire model. The intercontinental variation in the relationship between precipitation and

maximum tree cover is much smaller for the models compared to the observations. Known variations in vegetation are not sufficiently understood to be represented in models. However, our finding that models do show differences in the fire-vegetation-climate relationships between continents shows that further exploration why models show differences can be helpful to better understand causes for intercontinental differences.

Overall the multivariate model evaluation highlights the potential for more targeted model improvements with respect to the interactions between climate, vegetation and fire, which are crucial for our understanding of future vegetation and climate projections.

*Code and data availability.*  The observational datasets are freely available. The processed data and model output as displayed in this publi-

cation and the processing scripts are available upon request to publications@mpimet.mpg.de.

**Appendix A:  Sensitivity of climate-vegetation-fire relationships to remapping, presence of shrubs and modelled tree height**

**Appendix B:  Evaluation of precipitation forcing**

Additionally to the total amount of rainfall the seasonality can play role for vegetation or the length of dry periods. We

therefore assess here whether the rainfall seasonality and the number of dry days are reasonable in our climatic forcing. We use the TMPA 3B42 daily dataset (Savtchenko and Greenbelt, 2016) as a reference and define rainfall seasonality as the number of days needed to reach 80% of the annual precipitation, and dry days as days with less rainfall than 3 mm. A low number of days need to reach the 80% rainfall indicates a strong seasonality, a high number of days a low seasonality. The MPI-ESM does not show a concerning underestimation of dry days or too low seasonality, there is a small underestimation however and

it is stronger in regions with high rainfall.

**Appendix C:  Relationship between modelled burned area and fire intensity**

*Author contributions.*  GL wrote the manuscript. GL and TM designed the study and performed the analysis. SH, DD, SK helped refine the analysis and to develop and shape the manuscript.

*Competing interests.*  The authors have no competing interests

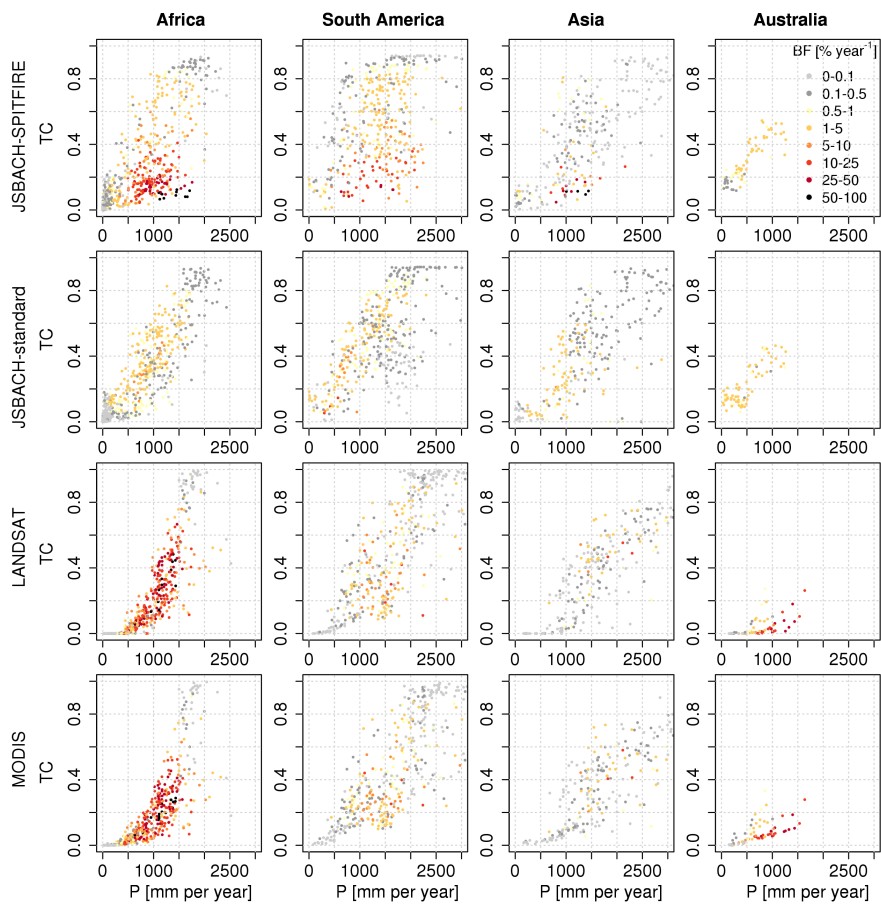

**Figure A1.** Same as figure 4 but tree cover filtered for the presence of shrub lands (using the MODIS open and closed shrub land classification). This indicates a low sensitivity of the fire-vegetation-climate relationships to shrub lands.

*Acknowledgements.* We would like to thank the DKRZ for excellent computing facilities. D. D'Onofrio acknowledges support from the European Union Horizon 2020 research and innovation programme under grant agreement No 641816 (CRESCENDO). S.H. acknowledges support by the EU FP7 projects BACCHUS (grant agreement no. 603445) and LUC4C (grant agreement no. 603542). We thank Victor Brovkin for valuable discussions and comments on this manuscript and are grateful to the three anonymous reviewers for their detailed reviews.

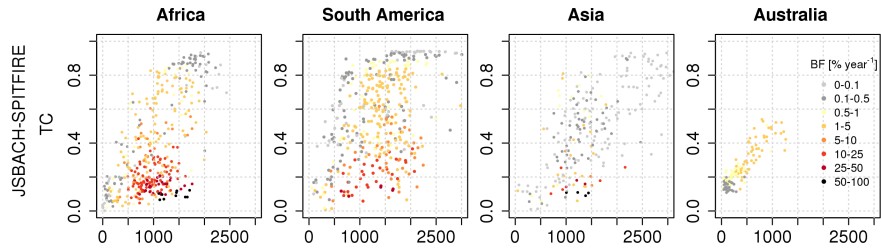

**Figure A2.** Modelled tree cover (TC) versus precipitation (P) [mm year-1]. Modelled tree cover was filtered for vegetation height of trees <5 m using the modelled vegetation height. This value is given as detection threshold for the satellite products. When filtering the model output with this threshold the differences to the unfiltered dataset are very small (compare with Figure 4, panels for JSBACH-SPITFIRE).

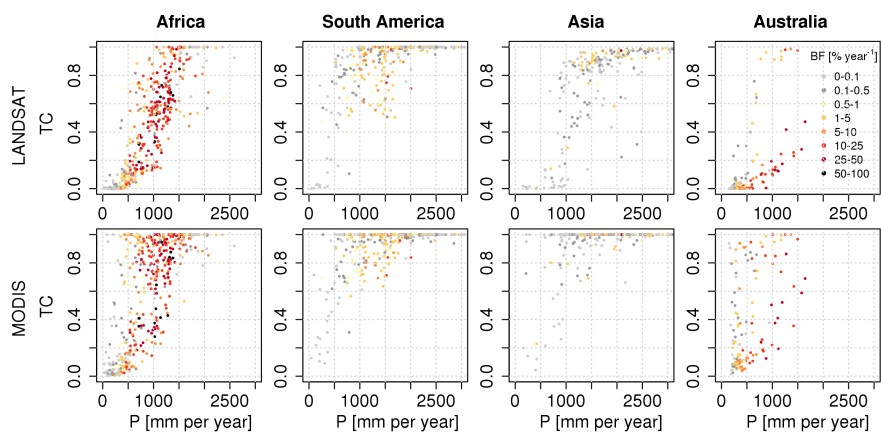

**Figure A3.** Tree cover (TC) versus precipitation (P) with color coded burned fraction (BF). Tree cover was here remapped from 0.05° resolution to 2° using the maximum value of the higher resolution instead of the mean.

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

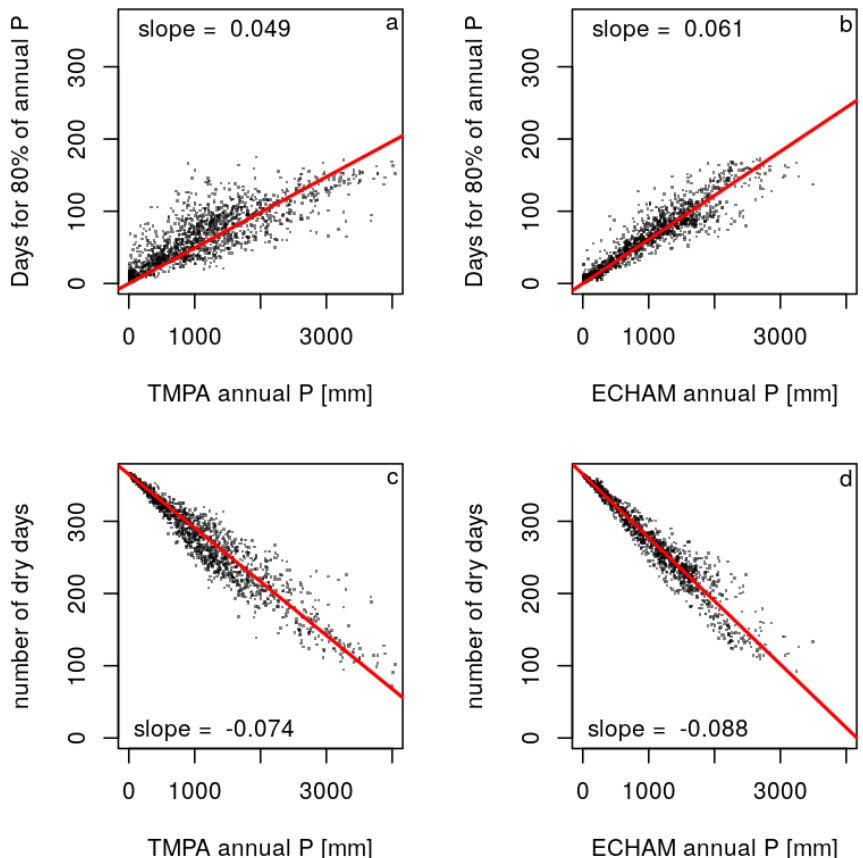

**Figure B1.** Relationship between annual precipitation and precipitation seasonality and number of dry days, respectively, for the ECHAM simulation used as meteorological forcing for the JSBACH simulations used here and the TMPA 3B42 daily dataset. Slope indicates the slope of the regression line.

Andela, N., Morton, D. C., Giglio, L., Chen, Y., van der Werf, G. R., Kasibhatla, P. S., DeFries, R. S., Collatz, G. J., Hantson, S., Kloster, S., Bachelet, D., Forrest, M., Lasslop, G., Li, F., Mangeon, S., Melton, J. R., Yue, C., and Randerson, J. T.: A human-driven decline in global burned area, Science, 356, 1356–1362, https://doi.org/10.1126/science.aal4108, http://www.sciencemag.org/lookup/doi/10.1126/science.aal4108, 2017.

5  Avitabile, V., Herold, M., Heuvelink, G. B. M., Lewis, S. L., Phillips, O. L., Asner, G. P., Armston, J., Ashton, P. S., Banin, L., Bayol, N., Berry, N. J., Boeckx, P., de Jong, B. H. J., DeVries, B., Girardin, C. A. J., Kearsley, E., Lindsell, J. A., Lopez-Gonzalez, G., Lucas, R., Malhi, Y., Morel, A., Mitchard, E. T. A., Nagy, L., Qie, L., Quinones, M. J., Ryan, C. M., Ferry, S. J. W., Sunderland, T., Laurin, G. V., Gatti, R. C., Valentini, R., Verbeeck, H., Wijaya, A., and Willcock, S.: An integrated pan-tropical biomass map using multiple reference datasets, Glob. Chang. Biol., 22, 1406–1420, https://doi.org/10.1111/gcb.13139, http://doi.wiley.com/10.1111/gcb.13139, 2016.

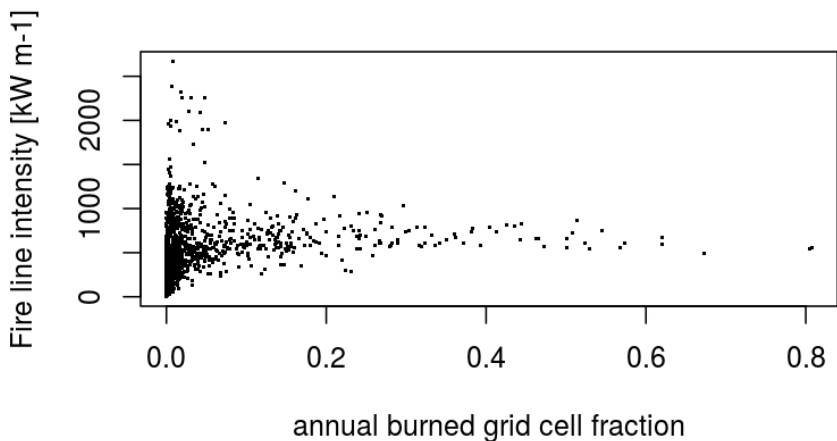

**Figure C1.** Relationship between annual burned area and fire line intensity. The expected decrease in fire line intensity for frequently burning areas due to the feedback between fire and fuel load is not found in the simulation results and might indicate that the feedback between fire occurrence, fuel load and fire intensity is too weak.

Baccini, A., Goetz, S. J., Walker, W. S., Laporte, N. T., Sun, M., Sulla-Menashe, D., Hackler, J., Beck, P. S. A., Dubayah, R., Friedl, M. A., Samanta, S., and Houghton, R. A.: Estimated carbon dioxide emissions from tropical deforestation improved by carbon-density maps, Nat. Clim. Chang., 2, 182–185, https://doi.org/10.1038/nclimate1354, http://www.nature.com/articles/nclimate1354, 2012.

Bathiany, S., Claussen, M., Brovkin, V., Raddatz, T., and Gayler, V.: Combined biogeophysical and biogeochemical effects of large-scale
forest cover changes in the MPI earth system model, Biogeosciences, 7, 1383–1399, https://doi.org/10.5194/bg-7-1383-2010, http://www. biogeosciences.net/7/1383/2010/, 2010.

Baudena, M., Dekker, S. C., van Bodegom, P. M., Cuesta, B., Higgins, S. I., Lehsten, V., Reick, C. H., Rietkerk, M., Scheiter, S., Yin, Z., Zavala, M. A., and Brovkin, V.: Forests, savannas, and grasslands: bridging the knowledge gap between ecology and Dynamic Global Vegetation Models, Biogeosciences, 12, 1833–1848, https://doi.org/10.5194/bg-12-1833-2015, http://www.biogeosciences.net/12/1833/
2015/, 2015.

Bistinas, I., Harrison, S. P., Prentice, I. C., and Pereira, J. M. C.: Causal relationships versus emergent patterns in the global controls of fire frequency, Biogeosciences, 11, 5087–5101, https://doi.org/10.5194/bg-11-5087-2014, 2014.

Bonan, G. B.: Forests and climate change: forcings, feedbacks, and the climate benefits of forests., Science (80-. )., 320, 1444–1449, https://doi.org/10.1126/science.1155121, http://www.ncbi.nlm.nih.gov/pubmed/18556546, 2008.

Brando, P. M., Balch, J. K., Nepstad, D. C., Morton, D. C., Putz, F. E., Coe, M. T., Silverio, D., Macedo, M. N., Davidson, E. A., Nobrega, C. C., Alencar, A., and Soares-Filho, B. S.: Abrupt increases in Amazonian tree mortality due to drought-fire interactions, Proc. Natl. Acad. Sci., 111, 6347–6352, https://doi.org/10.1073/pnas.1305499111, http://www.pnas.org/cgi/doi/10.1073/pnas.1305499111, 2014.

Brovkin, V., Claussen, M., Petoukhov, V., and Ganopolski, A.: On the stability of the atmosphere-vegetation system in the Sahara/Sahel region, Journal of Geophysical Research, 103, 31 613, https://doi.org/10.1029/1998JD200006, http://doi.wiley.com/10.1029/1998JD200006, 1998.

Brovkin, V., Raddatz, T., Reick, C. H., Claussen, M., and Gayler, V.: Global biogeophysical interactions between forest and climate, Geophysical Research Letters, 36, 1–6, https://doi.org/10.1029/2009GL037543, http://www.agu.org/pubs/crossref/2009/2009GL037543.shtml, 2009.

DeAngelis, A. M., Broccoli, A. J., and Decker, S. G.: A Comparison of CMIP3 Simulations of Precipitation over North America with Observations: Daily Statistics and Circulation Features Accompanying Extreme Events, J. Clim., 26, 3209–3230, https://doi.org/10.1175/JCLI-D-12-00374.1, http://journals.ametsoc.org/doi/abs/10.1175/JCLI-D-12-00374.1, 2013.

Devine, A. P., McDonald, R. A., Quaife, T., and Maclean, I. M. D.: Determinants of woody encroachment and cover in African savannas, Oecologia, 183, 939–951, https://doi.org/10.1007/s00442-017-3807-6, http://link.springer.com/10.1007/s00442-017-3807-6, 2017.

D'Onofrio, D., Baudena, M., D'Andrea, F., Rietkerk, M., and Provenzale, A.: Tree-grass competition for soil water in arid and semiarid savannas: The role of rainfall intermittency, Water Resour. Res., 51, 169–181, https://doi.org/10.1002/2014WR015515, http://doi.wiley.com/10.1002/2014WR015515, 2015.

Faivre, N., Jin, Y., Goulden, M. L., and Randerson, J. T.: Controls on the spatial pattern of wildfire ignitions in Southern California, Int. J. Wildl. Fire, 23, 799, https://doi.org/10.1071/WF13136, http://www.publish.csiro.au/?paper=WF13136, 2014.

Felsberg, A., Kloster, S., Wilkenskjeld, S., Krause, A., and Lasslop, G.: Lightning Forcing in Global Fire Models: The Importance of Temporal Resolution, Journal of Geophysical Research: Biogeosciences, 123, https://doi.org/10.1002/2017JG004080, 2018.

Forkel, M., Dorigo, W., Lasslop, G., Teubner, I., Chuvieco, E., and Thonicke, K.: A data-driven approach to identify controls on global fire activity from satellite and climate observations (SOFIA V1), Geoscientific Model Development, 10, 4443–4476, https://doi.org/10.5194/gmd-10-4443-2017, https://www.geosci-model-dev.net/10/4443/2017/, 2017.

Friedl, M. A., Sulla-Menashe, D., Tan, B., Schneider, A., Ramankutty, N., Sibley, A., and Huang, X.: MODIS Collection 5 global land cover: Algorithm refinements and characterization of new datasets, Remote Sens. Environ., 114, 168–182, https://doi.org/10.1016/j.rse.2009.08.016, http://linkinghub.elsevier.com/retrieve/pii/S0034425709002673, 2010.

Gerard, F., Hooftman, D., van Langevelde, F., Veenendaal, E., White, S. M., and Lloyd, J.: MODIS VCF should not be used to detect discontinuities in tree cover due to binning bias. A comment on Hanan et al. (2014) and Staver and Hansen (2015), Glob. Ecol. Biogeogr., 26, 854–859, https://doi.org/10.1111/geb.12592, http://dx.doi.org/10.1111/geb.12592, 2017.

Giglio, L., Randerson, J. T., and van der Werf, G. R.: Analysis of daily, monthly, and annual burned area using the fourth-generation global fire emissions database (GFED4), J. Geophys. Res. Biogeosciences, 118, 317–328, https://doi.org/10.1002/jgrg.20042, http://doi.wiley.com/10.1002/jgrg.20042, 2013.

Giorgetta, M. A., Jungclaus, J., Reick, C. H., Legutke, S., Bader, J., Böttinger, M., Brovkin, V., Crueger, T., Esch, M., Fieg, K., Glushak, K., Gayler, V., Haak, H., Hollweg, H.-D., Ilyina, T., Kinne, S., Kornblueh, L., Matei, D., Mauritsen, T., Mikolajewicz, U., Mueller, W., Notz, D., Pithan, F., Raddatz, T., Rast, S., Redler, R., Roeckner, E., Schmidt, H., Schnur, R., Segschneider, J., Six, K. D., Stockhause, M., Timmreck, C., Wegner, J., Widmann, H., Wieners, K.-H., Claussen, M., Marotzke, J., and Stevens, B.: Climate and carbon cycle changes from 1850 to 2100 in MPI-ESM simulations for the Coupled Model Intercomparison Project phase 5, Journal of Advances in Modeling Earth Systems, 5, 572–597, https://doi.org/10.1002/jame.20038, http://doi.wiley.com/10.1002/jame.20038, 2013.

Gutowski, W. J., Decker, S. G., Donavon, R. A., Pan, Z., Arritt, R. W., and Takle, E. S.: Temporal–Spatial Scales of Observed and Simulated Precipitation in Central U.S. Climate, J. Clim., 16, 3841–3847, https://doi.org/10.1175/1520-0442(2003)016<3841:TSOOAS>2.0.CO;2, http://journals.ametsoc.org/doi/abs/10.1175/1520-0442{%}282003{%}29016{%}3C3841{%}3ATSOOAS{%}3E2.0.CO{%}3B2, 2003.

Hagemann, S. and Stacke, T.: Impact of the soil hydrology scheme on simulated soil moisture memory, Climate Dynamics, 44, 1731–1750, https://doi.org/10.1007/s00382-014-2221-6, http://link.springer.com/10.1007/s00382-014-2221-6, 2015.

Hagemann, S., Loew, A., and Andersson, A.: Combined evaluation of MPI-ESM land surface water and energy fluxes, J. Adv. Model. Earth Syst., pp. n/a—-n/a, https://doi.org/10.1029/2012MS000173, http://doi.wiley.com/10.1029/2012MS000173, 2013.

Hansen, M. C., DeFries, R. S., Townshend, J. R. G., Carroll, M., Dimiceli, C., and Sohlberg, R. a.: Global Percent Tree Cover at a Spatial Resolution of 500 Meters: First Results of the MODIS Vegetation Continuous Fields Algorithm, Earth Interactions, 7, 1–15, https://doi.org/10.1175/1087-3562(2003)007<0001:GPTCAA>2.0.CO;2, http://journals.ametsoc.org/doi/abs/10.1175/1087-3562(2003)007{%}3C0001:GPTCAA{%}3E2.0.CO;2, 2003.

Hansen, M. C., Potapov, P. V., Moore, R., Hancher, M., Turubanova, S. a., Tyukavina, A., Thau, D., Stehman, S. V., Goetz, S. J., Loveland, T. R., Kommareddy, A., Egorov, A., Chini, L., Justice, C. O., and Townshend, J. R. G.: High-resolution global maps of 21st-century forest cover change., Science, 342, 850–853, https://doi.org/10.1126/science.1244693, http://www.ncbi.nlm.nih.gov/pubmed/24233722, 2013.

Hantson, S., Lasslop, G., Kloster, S., and Chuvieco, E.: Anthropogenic effects on global mean fire size, International Journal of Wildland Fire, 24, 589–596, 2015.

Hantson, S., Arneth, A., Harrison, S., Kelley, D., Colin Prentice, I., Rabin, S., Archibald, S., Mouillot, F., Arnold, S., Artaxo, P., Bachelet, D., Ciais, P., Forrest, M., Friedlingstein, P., Hickler, T., Kaplan, J., Kloster, S., Knorr, W., Lasslop, G., Li, F., Mangeon, S., Melton, J., Meyn, A., Sitch, S., Spessa, A., Van Der Werf, G., Voulgarakis, A., and Yue, C.: The status and challenge of global fire modelling, Biogeosciences, 13, https://doi.org/10.5194/bg-13-3359-2016, 2016.

Hantson, S., Scheffer, M., Pueyo, S., Xu, C., Lasslop, G., Van Nes, E., Holmgren, M., and Mendelsohn, J.: Rare, Intense, Big fires dominate the global tropics under drier conditions, Scientific Reports, 7, https://doi.org/10.1038/s41598-017-14654-9, 2017.

Higgins, S. I. and Scheiter, S.: Atmospheric CO2 forces abrupt vegetation shifts locally, but not globally, Nature, 488, 209–212, https://doi.org/10.1038/nature11238, http://www.nature.com/doifinder/10.1038/nature11238, 2012.

Higgins, S. I., Bond, W. J., and Trollope, W. S. W.: Fire, resprouting and variability: a recipe for grass–tree coexistence in savanna, J. Ecol., 88, 213–229, https://doi.org/10.1046/j.1365-2745.2000.00435.x, http://dx.doi.org/10.1046/j.1365-2745.2000.00435.x, 2000.

Hirota, M., Holmgren, M., Van Nes, E. H., and Scheffer, M.: Global Resilience of Tropical Forest and Savanna to Critical Transitions, Science, 334, 232–235, https://doi.org/10.1126/science.1210657, http://www.sciencemag.org/cgi/doi/10.1126/science.1210657, 2011.

Hoffmann, W. a., Orthen, B., and Vargas Do Nascimento, P. K.: Comparative fire ecology of tropical savanna and forest trees, Functional Ecology, 17, 720–726, https://doi.org/10.1111/j.1365-2435.2003.00796.x, 2003.

Hoffmann, W. A., Adasme, R., Haridasan, M., T. de Carvalho, M., Geiger, E. L., Pereira, M. A. B., Gotsch, S. G., and Franco, A. C.: Tree topkill, not mortality, governs the dynamics of savanna–forest boundaries under frequent fire in central Brazil, Ecology, 90, 1326–1337, https://doi.org/10.1890/08-0741.1, http://doi.wiley.com/10.1890/08-0741.1, 2009.

Hoffmann, W. a., Geiger, E. L., Gotsch, S. G., Rossatto, D. R., Silva, L. C. R., Lau, O. L., Haridasan, M., and Franco, A. C.: Ecological thresholds at the savanna-forest boundary: how plant traits, resources and fire govern the distribution of tropical biomes., Ecology letters, 15, 759–768, https://doi.org/10.1111/j.1461-0248.2012.01789.x, http://www.ncbi.nlm.nih.gov/pubmed/22554474, 2012.

Huffman, G. J., Bolvin, D. T., Nelkin, E. J., Wolff, D. B., Adler, R. F., Gu, G., Hong, Y., Bowman, K. P., and Stocker, E. F.: The TRMM Multisatellite Precipitation Analysis (TMPA): Quasi-Global, Multiyear, Combined-Sensor Precipitation Estimates at Fine Scales, J. Hydrometeorol., 8, 38–55, https://doi.org/10.1175/JHM560.1, http://journals.ametsoc.org/doi/abs/10.1175/JHM560.1, 2007.

Huffman, G. J., Adler, R. F., Bolvin, D. T., and Nelkin, E. J.: The TRMM Multi-satellite Precipitation Analysis (TMPA), in: Satell. Rainfall Appl. Surf. Hydrol., edited by Hossain, F. and Gebremichael, M., chap. 1, pp. 3–22, Springer Verlag, 2010.

Hurtt, G. C., Chini, L. P., Frolking, S., Betts, R. A., Feddema, J., Fischer, G., Fisk, J. P., Hibbard, K., Houghton, R. A., Janetos, A., Jones, C. D., Kindermann, G., Kinoshita, T., Klein Goldewijk, K., Riahi, K., Shevliakova, E., Smith, S., Stehfest, E., Thomson, A., Thornton, P., Vuuren, D. P., and Wang, Y. P.: Harmonization of land-use scenarios for the period 1500-–2100: 600 years of global gridded annual land-use transitions, wood harvest, and resulting secondary lands, Climatic Change, 109, 117–161, https://doi.org/10.1007/s10584-011-0153-2, http://link.springer.com/10.1007/s10584-011-0153-2, 2011.

Kelley, D. I., Prentice, I. C., Harrison, S. P., Wang, H., Simard, M., Fisher, J. B., and Willis, K. O.: A comprehensive benchmarking system for evaluating global vegetation models, Biogeosciences, 10, 3313–3340, https://doi.org/10.5194/bg-10-3313-2013, http://www.biogeosciences.net/10/3313/2013/, 2013.

Kelley, D. I., Harrison, S. P., and Prentice, I. C.: Improved simulation of fire–vegetation interactions in the Land surface Processes and eXchanges dynamic global vegetation model (LPX-Mv1), Geosci. Model Dev., 7, 2411–2433, https://doi.org/10.5194/gmd-7-2411-2014, http://www.geosci-model-dev.net/7/2411/2014/, 2014.

Klein Goldewijk, K.: Estimating global land use change over the past 300 years, Global Biogeochem. Cycles, 15, 417–443, 2001.

Kloster, S., Mahowald, N. M., Randerson, J. T., Thornton, P. E., Hoffman, F. M., Levis, S., Lawrence, P. J., Feddema, J. J., Oleson, K. W., and Lawrence, D. M.: Fire dynamics during the 20th century simulated by the Community Land Model, Biogeosciences, 7, 1877–1902, http://www.biogeosciences.net/7/1877/2010/, 2010.

Koenker, R.: quantreg: Quantile Regression, https://cran.r-project.org/package=quantreg, 2018.

Koster, R. D., Sud, Y. C., Guo, Z., Dirmeyer, P. A., Bonan, G., Oleson, K. W., Chan, E., Verseghy, D., Cox, P., Davies, H., Kowalczyk, E., Gordon, C. T., Kanae, S., Lawrence, D., Liu, P., Mocko, D., Lu, C.-H., Mitchell, K., Malyshev, S., McAvaney, B., Oki, T., Yamada, T., Pitman, A., Taylor, C. M., Vasic, R., and Xue, Y.: GLACE: The Global Land–Atmosphere Coupling Experiment. Part I: Overview, J. Hydrometeorol., 7, 590–610, https://doi.org/10.1175/JHM510.1, http://journals.ametsoc.org/doi/abs/10.1175/JHM510.1, 2006.

Krause, A., Kloster, S., Wilkenskjeld, S., and Paeth, H.: The sensitivity of global wildfires to simulated past, present, and future lightning frequency, J. Geophys. Res. Biogeosciences, 119, 312–322, https://doi.org/10.1002/2013JG002502, http://doi.wiley.com/10.1002/2013JG002502, 2014.

Krawchuk, M. a. and Moritz, M. a.: Constraints on global fire activity vary across a resource gradient., Ecology, 92, 121–132, http://www.ncbi.nlm.nih.gov/pubmed/21560682, 2011.

Lasslop, G. and Kloster, S.: Impact of fuel variability on wildfire emission estimates, Atmospheric Environment, 121, 93–102, http://dx.doi.org/10.1016/j.atmosenv.2015.05.040, 2015.

Lasslop, G. and Kloster, S.: Human impact on wildfires varies between regions and with vegetation productivity, Environmental Research Letters, 12, https://doi.org/10.1088/1748-9326/aa8c82, 2017.

Lasslop, G., Thonicke, K., and Kloster, S.: SPITFIRE within the MPI Earth system model: Model development and evaluation, Journal of Advances in Modeling Earth Systems, 6, 740–755, 2014.

Lasslop, G., Brovkin, V., Reick, C., Bathiany, S., and Kloster, S.: Multiple stable states of tree cover in a global land surface model due to a fire-vegetation feedback, Geophysical Research Letters, 43, https://doi.org/10.1002/2016GL069365, 2016.

Lehmann, C. E. R., Anderson, T. M., Sankaran, M., Higgins, S. I., Archibald, S., Hoffmann, W. A., Hanan, N. P., Williams, R. J., Fensham, R. J., Felfili, J., Hutley, L. B., Ratnam, J., San Jose, J., Montes, R., Franklin, D., Russell-Smith, J., Ryan, C. M., Durigan, G., Hiernaux, P., Haidar, R., Bowman, D. M. J. S., and Bond, W. J.: Savanna Vegetation-Fire-Climate Relationships Differ Among Continents, Science (80-. )., 343, 548–552, https://doi.org/10.1126/science.1247355, http://www.sciencemag.org/cgi/doi/10.1126/science.1247355, 2014.

Li, F., Bond-Lamberty, B., and Levis, S.: Quantifying the role of fire in the Earth system – Part 2: Impact on the net carbon balance of global terrestrial ecosystems for the 20th century, Biogeosciences, 11, 1345–1360, https://doi.org/10.5194/bg-11-1345-2014, http://www.biogeosciences.net/11/1345/2014/, 2014.

Li, F., Lawrence, D. M., and Bond-Lamberty, B.: Impact of fire on global land surface air temperature and energy budget for the 20th century due to changes within ecosystems, Environ. Res. Lett., 12, 44 014, http://stacks.iop.org/1748-9326/12/i=4/a=044014, 2017.

Mattiuzzi, M. and Detsch, F.: MODIS: Acquisition and Processing of MODIS Products, https://cran.r-project.org/package=MODIS, 2018.

Moncrieff, G. R., Scheiter, S., Bond, W. J., and Higgins, S. I.: Increasing atmospheric CO 2 overrides the historical legacy of multiple stable biome states in Africa, New Phytologist, 201, 908–915, https://doi.org/10.1111/nph.12551, http://doi.wiley.com/10.1111/nph.12551, 2014.

Morton, D. C., Le Page, Y., DeFries, R., Collatz, G. J., and Hurtt, G. C.: Understorey fire frequency and the fate of burned forests in southern

Amazonia, Philos. Trans. R. Soc. B Biol. Sci., 368, 20120 163, https://doi.org/10.1098/rstb.2012.0163, http://rstb.royalsocietypublishing.org/cgi/doi/10.1098/rstb.2012.0163, 2013.

Narayanaraj, G. and Wimberly, M. C.: Influences of forest roads on the spatial patterns of human- and lightning-caused wildfire ignitions, Appl. Geogr., 32, 878–888, https://doi.org/10.1016/j.apgeog.2011.09.004, http://linkinghub.elsevier.com/retrieve/pii/S0143622811001731, 2012.

Padilla, M., Stehman, S. V., Ramo, R., Corti, D., Hantson, S., Oliva, P., Alonso-Canas, I., Bradley, A. V., Tansey, K., Mota, B., Pereira, J. M., and Chuvieco, E.: Comparing the accuracies of remote sensing global burned area products using stratified random sampling and estimation, Remote Sensing of Environment, https://doi.org/10.1016/j.rse.2015.01.005, http://linkinghub.elsevier.com/retrieve/pii/S0034425715000140, 2015.

Pellegrini, A. F. A., Anderegg, W. R. L., Paine, C. E. T., Hoffmann, W. A., Kartzinel, T., S., S., Sheil, D., Franco, A. C., and Pacala, S. W.:

Convergence of bark investment according to fire and climate structures ecosystem vulnerability to future change, Ecol. Lett., 20, 307–316, https://doi.org/10.1111/ele.12725, http://doi.wiley.com/10.1111/ele.12725, 2017.

Prentice, I. C., Kelley, D. I., Foster, P. N., Friedlingstein, P., Harrison, S. P., and Bartlein, P. J.: Modeling fire and the terrestrial carbon balance, Global Biogeochemical Cycles, 25, 1–13, https://doi.org/10.1029/2010GB003906, http://www.agu.org/pubs/crossref/2011/2010GB003906.shtml, 2011.

Rabin, S., Melton, J., Lasslop, G., Bachelet, D., Forrest, M., Hantson, S., Kaplan, J., Li, F., Mangeon, S., Ward, D., Yue, C., Arora, V., Hickler, T., Kloster, S., Knorr, W., Nieradzik, L., Spessa, A., Folberth, G., Sheehan, T., Voulgarakis, A., Kelley, D., Colin Prentice, I., Sitch, S., Harrison, S., and Arneth, A.: The Fire Modeling Intercomparison Project (FireMIP), phase 1: Experimental and analytical protocols with detailed model descriptions, Geoscientific Model Development, 10, https://doi.org/10.5194/gmd-10-1175-2017, 2017.

Randerson, J. T., Chen, Y., van der Werf, G. R., Rogers, B. M., and Morton, D. C.: Global burned area and biomass burning emissions

from small fires, J. Geophys. Res. Biogeosciences, 117, G04 012, https://doi.org/10.1029/2012JG002128, http://doi.wiley.com/10.1029/2012JG002128, 2012.

Reick, C. H., Raddatz, T., Brovkin, V., and Gayler, V.: Representation of natural and anthropogenic land cover change in MPI-ESM, Journal of Advances in Modeling Earth Systems, 5, 459–482, https://doi.org/10.1002/jame.20022, http://doi.wiley.com/10.1002/jame.20022, 2013.

Romps, D. M., Seeley, J. T., Vollaro, D., and Molinari, J.: Projected increase in lightning strikes in the United States due to global warming, Science, 346, 851–854, https://doi.org/10.1126/science.1259100, http://www.sciencemag.org/cgi/doi/10.1126/science.1259100, 2014.

Saatchi, S. S., Harris, N. L., Brown, S., Lefsky, M., Mitchard, E. T. A., Salas, W., Zutta, B. R., Buermann, W., Lewis, S. L., Hagen, S., Petrova, S., White, L., Silman, M., and Morel, A.: Benchmark map of forest carbon stocks in tropical regions across three continents., Proceedings of the National Academy of Sciences of the United States of America, 108, 9899–9904, https://doi.org/10.1073/pnas.1019576108, http://www.pubmedcentral.nih.gov/articlerender.fcgi?artid=3116381{%}7B{&}{%}7Dtool= pmcentrez{%}7B{&}{%}7Drendertype=abstract, 2011.

Sankaran, M., Hanan, N. P., Scholes, R. J., Ratnam, J., Augustine, D. J., Cade, B. S., Gignoux, J., Higgins, S. I., Le Roux, X., Ludwig, F., Ardo, J., Banyikwa, F., Bronn, A., Bucini, G., Caylor, K. K., Coughenour, M. B., Diouf, A., Ekaya, W., Feral, C. J., February, E. C., Frost, P. G. H., Hiernaux, P., Hrabar, H., Metzger, K. L., Prins, H. H. T., Ringrose, S., Sea, W., Tews, J., Worden, J., and Zambatis, N.: Determinants of woody cover in African savannas., Nature, 438, 846–849, https://doi.org/10.1038/nature04070, 2005.

Savtchenko, A. and Greenbelt, M.: TRMM (TMPA-RT) Near Real-Time Precipitation L3 1 day 0.25 degree x 0.25 degree V7, https://doi.org/10.5067/TRMM/TMPA/DAY-E/7, https://disc.gsfc.nasa.gov/datasets/TRMM{_}3B42RT{_}Daily{_}7/summary, 2016.

Sillmann, J., Kharin, V. V., Zhang, X., Zwiers, F. W., and Bronaugh, D.: Climate extremes indices in the CMIP5 multimodel ensemble: Part 1. Model evaluation in the present climate, Journal of Geophysical Research: Atmospheres, 118, 1716–1733, https://doi.org/10.1002/jgrd.50203, http://doi.wiley.com/10.1002/jgrd.50203, 2013.

Sitch, S., Smith, B., Prentice, I. C., Arneth, A., Bondeau, A., Cramer, W., Kaplan, J. O., Levis, S., Lucht, W., Sykes, M. T., Thonicke, K., and Venevsky, S.: Evaluation of ecosystem dynamics, plant geography and terrestrial carbon cycling in the LPJ dynamic global vegetation model, Global Change Biology, 9, 161–185, https://doi.org/10.1046/j.1365-2486.2003.00569.x, http://doi.wiley.com/10.1046/j.1365-2486.2003.00569.x, 2003.

Staver, A. C., Archibald, S., and Levin, S.: Tree cover in sub-Saharan Africa: Rainfall and fire constrain forest and savanna as alternative stable states, Ecology, 92, 1063–1072, https://doi.org/10.1890/10-1684.1, http://www.esajournals.org/doi/10.1890/10-1684.1, 2011a.

Staver, A. C., Archibald, S., and Levin, S. A.: The Global Extent and Determinants of Savanna and Forest as Alternative Biome States, Science, 334, 230–232, https://doi.org/10.1126/science.1210465, http://www.sciencemag.org/cgi/doi/10.1126/science.1210465, 2011b.

Stevens, B., Giorgetta, M., Esch, M., Mauritsen, T., Crueger, T., Rast, S., Salzmann, M., Schmidt, H., Bader, J., Block, K., Brokopf, R., Fast, I., Kinne, S., Kornblueh, L., Lohmann, U., Pincus, R., Reichler, T., and Roeckner, E.: Atmospheric component of the MPI-M Earth System Model: ECHAM6, J. Adv. Model. Earth Syst., 5, 146–172, https://doi.org/10.1002/jame.20015, http://doi.wiley.com/10.1002/jame.20015, 2013.

Thonicke, K., Spessa, A., Prentice, I. C., Harrison, S. P., Dong, L., and Carmona-Moreno, C.: The influence of vegetation, fire spread and fire behaviour on biomass burning and trace gas emissions: results from a process-based model, Biogeosciences, 7, 1991–2011, https://doi.org/10.5194/bg-7-1991-2010, http://www.biogeosciences.net/7/1991/2010/, 2010.

Townsend, J. R. G., Carroll, M., DiMiceli, C., Sohlberg, R., Hansen, M., and DeFries, R.: Vegetation Continuous Fields MOD44B, 2001-2010 Percent Tree Cover, Collection 5, Version 051, 2011.

Wei, Y., Liu, S., Huntzinger, D. N., Michalak, A. M., Viovy, N., Post, W. M., Schwalm, C. R., Schaefer, K., Jacobson, A. R., Lu, C., Tian, H., Ricciuto, D. M., Cook, R. B., Mao, J., and Shi, X.: The North American Carbon Program Multi-scale Synthesis and Terrestrial Model Intercomparison Project – Part 2: Environmental driver data, Geosci. Model Dev., 7, 2875–2893, https://doi.org/10.5194/gmd-7-2875-2014, http://www.geosci-model-dev.net/7/2875/2014/, 2014.

Xu, C., Hantson, S., Holmgren, M., van Nes, E. H., Staal, A., and Scheffer, M.: Remotely sensed canopy height reveals three pantropical ecosystem states, Ecology, 97, 2518–2521, https://doi.org/10.1002/ecy.1470, http://doi.wiley.com/10.1002/ecy.1470, 2016.

Yin, Z., Dekker, S. C., van den Hurk, B. J. J. M., and Dijkstra, H. A.: Bimodality of woody cover and biomass across the precipitation gradient in West Africa, Earth System Dynamics, 5, 257–270, https://doi.org/10.5194/esd-5-257-2014, http://www.earth-syst-dynam.net/5/257/2014/, 2014.

Yue, C., Ciais, P., Zhu, D., Wang, T., Peng, S. S., and Piao, S. L.: How have past fire disturbances contributed to the current carbon balance of boreal ecosystems?, Biogeosciences, 13, 675–690, https://doi.org/10.5194/bg-13-675-2016, https://www.biogeosciences.net/13/675/2016/, 2016.