# Peer review of "Tropical climate-vegetation-fire relationships: multivariate evaluation of the land surface model JSBACH"

_Biogeosciences, 2018_

## Referee Comment (RC1) · Anonymous Referee #1 · 20 Feb 2018

General Comments:

Reviewer summary: The manuscript presents results from multi-variate comparisons between a simple fire model and complex fire model within JSBACH against those of remote sensing datasets for tree cover, grass cover, and burned fraction for regions within the tropics. The work finds that the resolution of the remote sensing datasets is important for setting precipitation limits on tree cover and burned fraction classifications. The fire models capture broad spatial patterns, but overall the complex fire model has improved performance. The analysis was completed for continental subsets and with and without preindustrial land use. Given the results the authors suggest

improving the drought response of vegetation, including more complex bark thickness for trees, and a representation of size-structure. The multi-variate analysis used here better identifies model-data mismatches to model processes.

Article contribution and overall impact: This study highlights the challenges of simulation of vegetation-fire interactions across the tropics. Strong climate vegetation relationships and a closely interacting fire regime make the vegetation state of this region difficult to simulate. The manuscript does a good job of presenting the challenges of capturing vegetation and fire in the tropics with simulation and with remote sensing datasets. The discussion would benefit from a more detailed description of the connections between recommended improvements and deficiencies of the simulations, as well as inclusion of more references. Please update the discussion to include a reference back to the figure or table being discussed (some of these are highlighted in detail comments). Specifically, more detailed discussion of size-structure and its importance as a mechanism for tree survival in fire prone regions should be included. A key component of the mortality of woody vegetation to fire is its size at the time of fire and the ability to accumulate size between fires. This is central to the work of many of W. Hoffman's papers in the region (Hoffman et al 2003, Hoffman et al 2009, Hoffman et al 2012). This type of work should be referenced as well as important differences between the continents in terms of vegetation survival from fire.

Detailed comments:

Page 6 line 15: Are burned area and burned fraction the same?

Page 7 line 12: "stronger relationship between low tree cover and high fire occurrence than observations" Explain this in more detail. By what measure and for which figure/table?

Page 9 line 6: Why use the preindustrial land use? The observation datasets are for the period of 1996-2005.

Page 10 line 4: Update "We here discuss..." to "Here we discuss..."

Page 10 line 7: Clarify that improvements in the SPITFIRE version cannot improve this mismatch. The standard version does not capture the observations as shown in figure 3.

Page 11 line 3: update "too high tree cover" to "excessive tree cover"

Page 11 line 5: update "too high dominance" to "excessive dominance"

Page 11 line 6-7: Explain how saplings being inferior to grasses would improve the representation of tree-grass competition? How would these saplings alter the resulting tree cover in areas where grasses exist? Are there processes in the model that would need to be added to include grass suppression of saplings?

Page 11 line 8-9: Include the figure that this relationship is referring to "higher burned fraction and lower tree cover for open canopies, however it is not found in the observations." Is this for figure 4? Also specify for what regions, as they are not consistent.

Page 12 line 1: Explain how increased bark thickness would be implemented in the model. Include discussion of the relationship between bark thickness and size-structure of trees, and species or regional variability in bark thickness characteristics, and how this might be accounted for in the model.

Page 12 line 4: "This feedback is included...but might be too weak." Support this statement with more detail. What information indicates that the feedback is too weak? Is this true for all regions? Which figures lead to this assertion?

Page 12 line 5-6: "...long-lived adult tree state could increase the survival of trees." How long do trees live in JSBACH? Provide some background on existing parameterization of tree life span and mortality mechanisms to support this statement. Include discussion of Hoffman's work on the 'fire-trap' within savanna systems.

Page 12 line 7: "For Australia...for both fire models is strong." Include the figure this is

referencing. Figure 4?

Page 13 line 3: Update to "The rank correlation...compared to model outputs (Table 1)." Include the reference to Table 1.

Page 13 -14 line 1: "adapts to changes in climate with usually PFT specific time scales." What does this mean? Are there variable PFT longevity within simulation?

Page 14 line 1-2: Include references to examples of DGVMs which include human dimensions.

Page 14 line 2: "...population density is a commonly used driver." Driver of what? Ignitions? Land use change? Please clarify.

Page 14 line 3: Start a new paragraph with the sentence beginning "Our model simulations..." and update this sentence to "Our model simulations also show that the modelled climate..."

Page 14 line 6: Update sentence to "...not affected by land use or by the type of fire model..."

Page 14 line 7: "...seasonality that is not resolved by the mean annual precipitation." The model has no seasonal variation in precipitation and is only using MAP? Please clarify.

Page 14 line 8-13: Include discussion of how the results differ due to the use of only preindustrial land use. Qualify the text in this section to clarify that the JSBACH simulations use preindustrial land use and these products use recent land use (Andela et al 2017 uses the past 18 years). Explain why the comparison is still valid.

Page 14 line 12: "The mechanism behind the reduction due to croplands..." Reduction of what? Fire occurrence? Please clarify.

Page 14 line 13: "...fragmentation of the landscape, which is not explicitly accounted for in the model." Include discussion of how fragmentation affects forests in reality, and

how this may be a challenge for models such as JSBACH. Is this an area for potential improvement?

Page 14 line 17: "...spatially varying ignitions." Do ignitions vary temporally?

Page 14 line 18: "...these differences in ignitions..." Differences between what? One is not spatially varied ignitions? Please clarify.

Page 14 line 32-33: Add at the end of the sentence what the values are for the satellite datasets. It is not possible to read them from the figures to compare to this measure of 100 mm and >650 mm per year.

Page 15 line 6-7: "...spatial scale needs to be considered..." Add discussion on how increased spatial scale (finer resolution) might improve the model results. Why not perform simulation at 1km similar to the Hirota dataset? Should simulation be finer than 1km? How small of a resolution can you achieve before you see compromised results for simulation?

Page 15 line 11-12: Are there plans to compare to biomass datasets? Identify potential datasets.

Page 15 line 26-28: "The multivariate comparison helped to . . ." Re-word this sentence. It is not clear what is meant by "too strong effect of fire on tree cover". Split into two sentences to identify problems, and then another to suggest improvements. Clarify where and how increased bark thickness can be included.

Page 15 -16 line 1: "although known variations in vegetation characteristics are not represented in models..." Provide a brief description of what is not represented? Bark thickness variability, size-structure? Consider adding a stronger concluding sentence to identify how these improvements will be helpful to models.

---

## Referee Comment (RC2) · Anonymous Referee #2 · 2 Mar 2018

The ability of JSBACH to reproduce the observed relationship between fire, tree and grass cover and mean annual precipitation (MAP) was assessed using two different coupled fire models, with the implicit aim of guiding future model development.
Analysis was split between continents to assess different regional climate-vegetation-fire relationships, and using present day and pre-industrial land use to assess the models ability to reproduce human impact on burnt area within bioclimate. The authors successfully demonstrates the potential of this approach by identifying too high tree cover at low precipitations, high burnt area in areas of low tree cover and cropland representation as key model weaknesses, before speculating on likely causes and so-

lutions. The approach is relatively simple but, as the authors point out, is also quite a novel way of identifying areas for improvements in vegetation-fire models which will hopefully be adopted by other modelling groups. I also like that the paper is solely dedicated to model assessment, despite the distraction of including JSBACH-standard (see comments below), and I look forward to seeing if this results in better informed, targeted model improvements in the future. If so, it could be a process the rest of us in the fire modelling community could learn from.

I do, however have a serious concern about the choice of driving data that needs to be addressed before I recommend publication. I also have a few other major comments, although some might just require brief clarification through author response and small changes to the m/s. Given the potential changes to the manuscript required to address the first major comment, I have only included a few key specific suggestion for now, largely for the introduction.

**Choice of JSBACH driving data**

JSBACH-fire was driven using simulated climate from the MIP Earth System Model. However, almost all the evaluation is of JSBACH-fire component alone. This is clearly a problem for the basic spatial evaluation in most section 3.1 and figure 1, where it is often unclear if mismatches in vegetation cover or burnt area is because of JSBACH itself or because of biases in the Earth System Models (ESM) climate simulation. As the rest of the paper is evaluating JSBACH in climate space, it could be argued that the choice of driving data doesn't matter. However, simulated climate biases could still be playing a role even here. For example, the authors only use MAP as a climate proxy. Inherent in MAPs influance on fire are the extreme conditions, specially the length dry periods, that increases susceptibility to burning. This is part of the reason for the wide range in fire and tree cover at a given MAP in all but the driest and wettest climates, and is invoked by the authors to explain different tree-MAP relationships in Australia.
General Circulation Models (GCMs) are notoriously poor at simulating dry periods, with many underestimating the length and/or severity of dry periods due to poor simulation of convective vs persistent rainfall and a problem with persistent dizzle (DeAngelis et al. 2013; Gutowski et al. 2003). Length of dry periods is fundamental in the calculation of ignition probability and each fire's area in SPITFIREs rate of spread model (Thonicke et al. 2010). The "standard" fire model used in this study sounds like it could be similar to GLOBFIRM? If so, this is also very sensitive to number of dry days, with a rapid increase in burnt area in longer dry seasons (Thonicke et al. 2001), which would explain at least part the underestimation of maximum burnt areas. Either way, driving and comparing JSBACH with ESM output could skew the MAP relationships with fire and potentially tree cover in figure 3-6 and A1-2. This is by no means the only problem with driving the model with ESM data, but it is the one that springs to mind. Using MPI also required the authors to make a rather awkward decision between performing comparisons on different time periods (1996-2005 from JSBASH runs; 2001-2010 from observations) or on the few years of model-observation overlap.

I have two suggestion for how the authors could address this problem:

1. Continue to use the MPI driven runs, but reframe the paper to evaluate the processes and identify weaknesses in simulation of tree cover and fire in the ESM as a whole. Some of the arguments I have made above as to how ESMs simulated climate could affect tree cover and fire could be included. However, there are likely many more, some specific to MPI. If there are any special required configuration of JSBACH to simulate vegetation dynamics under MPI then these should also be included. The authors briefly touch on two arguments that could also be expanded: lines 11-14 on page 6 uses figure 1c to briefly discuss whether MPIs MAP biases as a reason for some of the mismatches between observed and simulated burnt area; and lines 7-15 on page 13 where the mismatches between driving data introduced straight into the fire model (pop

density, lightning etc) and those driven from MPI climate output. Section 3.1 should just need re-formulating with no new analysis. Subsequent sections may require fresh analysis, potentially looking at multivariate relationships within the space of MPIs driving data.

2. Run JSBACH with climate observations, including using common precipitation observations for driving data, and analysis of observed and simulated fire, MAP and vegetation cover. According to (Rabin et al. 2017), JSBACH model output should be included in fireMIP, in which case, vegetation cover by PFT and burnt area from observation driven JSBACH will be available from fireMIP.

**Choice of fire dataset.**

Is there a reason for use of GFED4 instead of GFED4 with small fires (GFED4s) (van der Werf et al. 2017)? There may be a good reason for not including small fires, but given the prevalence of GFED4s in other fire evaluation studies (Rabin et al. 2017; Kelley et al. 2014; Kloster & Lasslop 2017), it might be worth including some justification. Also, are the certain weaknesses in fire detection in GFED that might affect the results? The missing smalls fire's for example should be mentioned as a caveat in relation to results from figure 3.

**Quantification of similarity in multivariate relationships.**

The observed relationship between MAP and tree cover is described as either "linear" for Australia and "sigmoid" for other continents, with the ability of each model to describe each curve used as evidence when identifying model weakness. However, I'm not sure I can see these relationships. Observed Australia looks more like the start of a

sigmoid (albeit with a shallower gradient when compared to e.g. Africa), "chopped" at low tree covers. South Africa looks more linear. A simple curve fitting and correlation could help determine how closely each continent resembles each function, and if the model is reproducing this relationship, which would place subsequent discussion on firmer ground.

The remaining multivariate comparisons is also largely based on visual comparisons on plots. While this is an important part of assessing differences in simulated vs observed relationships, I feel like the comparison could do with some quantification using some simple multivariate metric, expanding on the two-variable assessment in Table 1. I am by no means an expert in multivariate statistics though, and perhaps a "simple" comparison isn't possible. But if the authors have any thoughts on this, it would be good to hear (and perhaps include them in the m/s?)

**Use of two models**

More could be made of the use of the "standard model" (JASBACH-standard) to help analyse MPI, JSBACH or even SPITFIRE performance, which is obviously the model the authors will use in future studies. As a start, JSBACH-standard could do with a little bit more description to help inform later discussion. Are the curves describing relationship between relative humidity, fuel carbon and fire similar to GLOBFIRM (Thonicke et al. 2001), or are they more similar to those simpler rate of spread models such as CTEM (Arora & Boer 2005). Are parameters used by the model based on literature, site comparisons, or optimization of remote sensing? If the latter, is its poor performance likely due to biases in JSBACH simulation of vegetation or MPI climate and dizzle biases? If the former, is it additionally due to fire model structure or bad parameterizations? How much is PFT fraction remove after fire? Is 100% of burnt PFT removed, or just a fraction? If a fraction, does this vary? And does it vary by life form, PFT, burnt areas or some other relationship?

A better comparison between the two models in the discussion and/or conclusion might also further strength the case for use of multivariate approach. Despite its poor performance is there any part of the standard model multivariate relationship that could be used to guide development of SPITFIRE, particularly with respect PFT tree mortality? Is there any conclusion that can be drawn on the use of complex fire models to represent complex processes such as fire and fire-feedbacks, or does any part of the standard models performance (i.e, locations of fire occurrence) suggest that emergent behaviour of fire on coarse scales does not require the use of complex models? Does a comparison of strength and weakness of the two models say anything about the coupling to JSBACH or required configuration for use of JSBACH-fire in MPI?

Ofcourse, if the authors feel like nothing substantial can be learnt from comparing the two models, then they could consider removing JSBACH-standard from the m/s. However, there is nothing technically wrong with its inclusion, so I'll leave that for the authors to decide.

**Specific comments**

Page 2, line 24-25: The development of complex fire models actually started before widespread use of remotely sensed products. MC2 (Lenihan et al. 1998) forms the basis for most rate of spread models (Hantson et al. 2016), and SPITFIRE is itself a development of Reg-FIRM (Venevsky et al. 2002). Neither invoke the use of remote sensed burnt area.

Page 2, line 30: "the importance of benchmarking effects on vegetation has been noted". Not just noted, but also done (Kelley et al. 2014).

Page 3, line 11: Please use -180 to 180 coordinates longitudes.

Page 4, line 2-3: Replace (or include along side) "(Rabin et al. 2017)" with "(Thonicke et al. 2010; Lasslop et al. 2014)" . (Rabin et al. 2017) does provide description of SPITFIRE alongside several other fire models, but the authors should also give credit to the model developers.

Page 4, line 14: "During the 1000 year spin up period . . . " How was the spin-up determined? Where carbon or PFT fractions/burnt area in equilibrium by this point? How was this tested?

Page 5, line 31 - Page 4 line 1: Please add a citation to the r-package paper. I think (Tuck et al. 2014) is the correct reference, but the authors should check. Also include a direct reference to the r package used. Typing in the following in an R terminal should give you the require bibtex information:

» citation(«package name»)

Page 5, lines 12-13: Is any scaling applied when translating from LIS/OTD flash count to ignition sources? I.e, are cloud-cloud flashs removed?

Page 11, line 10 - Page 12, line 1: (Kelley et al. 2014) collected cite based bark thickness data to reparametrize bark thickness in a SPITFIRE based model. There might also be some Australia specific improvements in this paper that could be considered.

Page 14, lines 21 - 23: The non-independence of vegetation cover datasets should be included when introducing the datasets on page 4 and 5.

Page 17: Please complete author contributions.

**References**

Arora, V.K. & Boer, G.J., 2005. Fire as an interactive component of dynamic vegetation models. Journal of Geophysical Research: Biogeosciences, 110(G2). Available at: http://dx.doi.org/10.1029/2005jg000042.

DeAngelis, A.M., Broccoli, A.J. & Decker, S.G., 2013. A Comparison of CMIP3 Simulations of Precipitation over North America with Observations: Daily Statistics and Circulation Features Accompanying Extreme Events. Journal of climate, 26(10), pp.3209–3230.

Gutowski, W.J. et al., 2003. Temporal–Spatial Scales of Observed and Simulated Precipitation in Central U.S. Climate. Journal of climate, 16(22), pp.3841–3847.

Hantson, S. et al., 2016. The status and challenge of global fire modelling. Biogeosciences discussions , 13(11). Available at: http://dx.doi.org/10.5194/bg-2016-17.

Kelley, D.I., Harrison, S.P. & Prentice, I.C., 2014. Improved simulation of fire–vegetation interactions in the Land surface Processes and eXchanges dynamic global vegetation model (LPX-Mv1). Geoscientific Model Development, 7(5), pp.2411–2433.

Kloster, S. & Lasslop, G., 2017. Historical and future fire occurrence (1850 to 2100) simulated in CMIP5 Earth System Models. Global and planetary change, 150, pp.58–69.

Lasslop, G., Thonicke, K. & Kloster, S., 2014. SPITFIRE within the MPI Earth system model: Model development and evaluation. Journal of Advances in Modeling Earth Systems, 6(3), pp.740–755.

Lenihan, J.M. et al., 1998. Simulating broad-scale fire severity in a dynamic global vegetation model. Northwest science: official publication of the Northwest Scientific Association, 72(4), pp.:91–101.

Rabin, S.S. et al., 2017. The Fire Modeling Intercomparison Project (FireMIP), phase 1: experimental and analytical protocols with detailed model descriptions. Geoscientific Model Development, 10(3), pp.1175–1197.

Thonicke, K. et al., 2010. The influence of vegetation, fire spread and fire behaviour on biomass burning and trace gas emissions: results from a process-based model. Biogeosciences , 7(6), pp.1991–2011.

Thonicke, K. et al., 2001. The role of fire disturbance for global vegetation dynamics: coupling fire into a Dynamic Global Vegetation Model. Global ecology and biogeography: a journal of macroecology, 10(6), pp.661–677.

Tuck, S.L. et al., 2014. MODISTools - downloading and processing MODIS remotely sensed data in R. Ecology and evolution, 4(24), pp.4658–4668.

Venevsky, S. et al., 2002. Simulating fire regimes in human-dominated ecosystems: Iberian Peninsula case study. Global change biology, 8(10), pp.984–998.

van der Werf, G.R. et al., 2017. Global fire emissions estimates during 1997–2016. Earth System Science Data, 9(2), pp.697–720.

---

## Author Comment (AC1) · 23 May 2018

*We thank the reviewers for their detailed and constructive comments, which strongly helped to improve the manuscript. We included additional analysis on the correlation between burned fraction and tree cover and on the increase of maximum tree cover with increasing precipitation. We hope our replies to the reviewer comments and the modifications will make the manuscript suitable for publication in biogeosciences. Our reponses are inserted below the reviewer comments in italics.*

[Figure]

**1   Review 1**

General Comments:
Reviewer summary: The manuscript presents results from multi-variate comparisons between a simple fire model and complex fire model within JSBACH against those of remote sensing datasets for tree cover, grass cover, and burned fraction for regions within the tropics. The work finds that the resolution of the remote sensing datasets is important for setting precipitation limits on tree cover and burned fraction classifications. The fire models capture broad spatial patterns, but overall the complex fire model has improved performance. The analysis was completed for continental sub- sets and with and without preindustrial land use. Given the results the authors suggest C1 improving the drought response of vegetation, including more complex bark thickness for trees, and a representation of size-structure. The multi-variate analysis used here better identifies model-data mismatches to model processes.

Article contribution and overall impact: This study highlights the challenges of simulation of vegetation-fire interactions across the tropics. Strong climate vegetation relationships and a closely interacting fire regime make the vegetation state of this region difficult to simulate. The manuscript does a good job of presenting the challenges of capturing vegetation and fire in the tropics with simulation and with remote sensing datasets. The discussion would benefit from a more detailed description of the connections between recommended improvements and deficiencies of the simulations, as well as inclusion of more references. Please update the discussion to include a reference back to the figure or table being discussed (some of these are highlighted in detail comments). Specifically, more detailed discussion of size-structure and its importance as a mechanism for tree survival in fire prone regions should be included. A key component of the mortality of woody vegetation to fire is its size at the time of fire and the ability to accumulate size between fires. This is central to the work of many of W. Hoffman's papers in the region (Hoffman et al 2003, Hoffman et al 2009, Hoffman et al 2012). This type of work should be referenced as well as important differences
between the continents in terms of vegetation survival from fire.

*1) We thank the reviewer for identifying this lack of details. We extended the discussion to improve the connection between improvements and deficiencies of simulated patterns including the recommended inclusion of more references to literature but also to the figures and table in the results section.*

Detailed comments:
Page 6 line 15: Are burned area and burned fraction the same?

*2) Burned area refers to the area, burned fraction to the fraction of the grid cell. It therefore is the same parameter but with a different unit. We change the burned area to burned fraction here, as the burned fraction is displayed in the figure to avoid confusion.*

Page 7 line 12: "stronger relationship between low tree cover and high fire occurrence than observations" Explain this in more detail. By what measure and for which figure/table?

*3) We modified the paragraph to explain it in more detail and refer to figure 4 and table 1. We also performed an additional analysis and include the correlation between burned fraction and tree cover in table 1 to quantify this strength of the relationship.*
*"Models and observations generally agree on the absence of fire for very high tree cover (>0.8) and on the decrease of burned fraction for mean annual precipitation decreasing below 1000 mm. However for regions with tree cover < 0.8 and mean annual precipitation > 1000 mm we find strong differences. JSBACH-SPITFIRE shows a strong negative Spearman rank correlation between burned fraction and tree cover, the observations show a weak negative correlation, and JSBACH-standard shows a positive correlation (Table 1). This can also be seen in Figure 4 where*

*for the JSBACH-SPITFIRE simulation the highest burned fractions (> 50% of grid cells year$^{-1}$) are found in Africa for the lowest tree covers (0.1) and for precipitation between 1000-2000 mm year$^{-1}$. JSBACH-standard in many grid cells shows low fire occurrence for low tree cover, especially for South America (Figure 4), these grid cells have a high fraction of crops or pasture, which both are excluded from burning in JSBACH-standard (in SPITFIRE only crops are excluded). The observations (also Figure 4) show highest values of the burned fraction for tree cover values up to 0.3 for MODIS and up to 0.5 for LANDSAT."*

Page 9 line 6: Why use the preindustrial land use? The observation datasets are for the period of 1996-2005.

*4) The preindustrial state is a state with low influence of land use, the comparison with the historical simulation therefore indicates the effect land use has on the climate-vegetation-fire relationships. We add two sentences in the beginning of the paragraph to explain the purpose:*
*"The simulation with preindustrial land use represents a state with low influence of land use change. The comparison to the historical simulation allows to assess the influence of land use change since 1850."*
*We also include a description of the changes in statistical parameters listed in table 1:*
*"The impact of fire on tree cover as quantified by the Spearman rank correlation between burned fraction and tree cover is higher for the simulation with preindustrial land use (Table 1). Land use change did not affect the rank correlation between precipitation and temperature. The precipitation range for 80% of the burned area is only slightly narrower for the simulation including land use change (Table 1). Tree cover, however, is even higher for low precipiation and reaches canopy closure for lower precipitation (Table 1 and Figure 7 compared to Figure 4)."*

Page 10 line 4: Update "We here discuss. . ." to "Here we discuss. . ."

*5) Updated in the revised manuscript.*

Page 10 line 7: Clarify that improvements in the SPITFIRE version cannot improve this mismatch. The standard version does not capture the observations as shown in figure 3.

*6) We changed this part (which refers to the mismatch of tree cover in low precipitation areas) to:*
*"In these dry regions no or only very low burned fractions are observed, and SPITFIRE shows a good response to precipitation while JSBACH-standard already overestimates the burned area (Figure 3). The improved burned area pattern of SPITFIRE did not lead to an improvement in tree cover for these dry regions. It is therefore unlikely that further improvements in burned fraction will improve this model-data mismatch, satellite data however indicate that the intensity of fires increases in these regions and might help to explain the disappearance of trees (Hantson et al., 2017). The mechanisms however are not sufficiently understood to be included in a model."*

Page 11 line 3: update "too high tree cover" to "excessive tree cover",
and
Page 11 line 5: update "too high dominance" to "excessive dominance"

*7) Done as suggested, we also change "too high" to "excessive" in the abstract.*

Page 11 line 6-7: Explain how saplings being inferior to grasses would improve the representation of tree-grass competition? How would these saplings alter the resulting tree cover in areas where grasses exist? Are there processes in the model that would need to be added to include grass suppression of saplings?

*8) Whether or not the inclusion of saplings would improve the representation of tree cover is certainly a matter of the exact implementation and model tuning. Answers to these detailed questions would therefore be highly speculative. We are therefore careful with our statements and add:*
*"Including this mechanism could improve the balance between tree and grass cover, but it could also reduce the establishment rate of trees and, therefore, the tree cover in the dry regions with excessive tree cover. Including a PFT-specific rooting depth of vegetation would be an important extension of the model to improve the competition for water between grasses, saplings and adult trees." In the end of the paragraph on model improvements we also add: "How exactly these plausible modifications would change the patterns of tree cover, fire and their relation to climate likely strongly depends on the exact parameterization and needs to be tested with stepwise model development and factorial simulations." To make sure readers understand that these are only suggestions and that they first need to be tested to understand how exactly they change the simulation outcome.*

Page 11 line 8-9: Include the figure that this relationship is referring to "higher burned fraction and lower tree cover for open canopies, however it is not found in the observations." Is this for figure 4? Also specify for what regions, as they are not consistent.

*9) We updated the sentence with references to figures:*
*The absence of fire for closed canopies is captured well by JSBACH-SPITFIRE, the modelled strong relationship between higher burned fraction and lower tree cover for open canopies (Figure 4, with the exception of Australia, Table 1), however, is not found in the observations (Figure 2,4). See also reply 3.*

Page 12 line 1: Explain how increased bark thickness would be implemented in the model. Include discussion of the relationship between bark thickness and size-structure of trees, and species or regional variability in bark thickness characteristics, and how this might be accounted for in the model.

*10) There are several ways to increase bark thickness, the first would be to modify the PFT specific bark thickness which depends on the tree biomass. Bark thickness could also increase according to previously burned area, assuming tree invest more in bark in regions with high fire occurrence. We modified the existing paragraph to:*
*"Bark thickness is a key property of trees for the fire-related mortality. In JSBACH-SPITFIRE bark thickness is PFT specific and depends on the biomass. The adaptation of trees to frequent fires by increased bark thickness, and therefore higher resistance of trees to fire (Pellegrini et al., 2017) would increase the tree cover in regions with high burned fraction. This could be implemented in the model with more specific PFTs or by modifying the bark thickness according to the fire regime. Kelley and Harrison (2014) included bark thickness as an adaptive trait in the LPX model, which increased and improved the tree cover for Australia. Resprouting is another important mechanism that changes the balance between mortality and recovery and also leads to an increase in tree cover in fire affected areas in a modelling study (Kelley and Harrison, 2014)."*
*see also reply 45*

Page 12 line 4: "This feedback is included. . .but might be too weak." Support this statement with more detail. What information indicates that the feedback is too weak? Is this true for all regions? Which figures lead to this assertion?

*11) This still refers to the correlation between burned fraction and tree cover and the highest burned fractions for rather high tree cover in the observations. We suggest*

*two possibilities of what could cause the higher correlation in the model, adaptation of bark thickness to fire regime or the feedback between fuel load and tree mortality. We slightly changed the sentence to clarify.*

Page 12 line 5-6: ". . .long-lived adult tree state could increase the survival of trees." How long do trees live in JSBACH? Provide some background on existing parameter- ization of tree life span and mortality mechanisms to support this state- ment. Include discussion of Hoffman's work on the 'fire-trap' within savanna systems.

*12) We include discussion on the general priciples proposed and supported by the work of Hoffmann, however his studies do not offer an explanation why the highest burned fractions are observed for rather high tree cover, which is the main surprise in the comparison here and the subject of the paragraph. The general principles dealt with in the work of Hoffmann are included in the model already. We include his work now in the discussion: "The absence of fire for closed canopies is captured well by JSBACH-SPITFIRE, the modelled strong relationship between higher burned fraction and lower tree cover for open canopies (Figure 4, with the exception of Australia, Table 1), however, is not found in the observations (Figure 2, 4,Table 1). Many general processes determining the savanna-forest boundary are included in the JSBACH-SPITFIRE model: Increased tree cover leads to a suppression of fire by excluding grasses, higher flammability of grasses leads to increases in fire occurrence with increasing grass biomass (Hoffmann et al., 2012). In JSBACH-SPITFIRE bark thickness is PFT specific and depends on the biomass. Tropical trees are represented by two PFTs one of them has a lower sensitivity to fire due to a higher bark thickness and a higher stem leading to a lower probability of crown scorch. This is also observed in field studies where savanna species show a higher ratio of bark thickness to stem diameter (Hoffmann et al., 2003). "*

Page 12 line 7: "For Australia. . .for both fire models is strong." Include the figure this is referencing. Figure 4?

*13) Yes, the reference to figure 4 is included now.*

Page 13 line 3: Update to "The rank correlation. . .compared to model outputs (Table 1)." Include the reference to Table 1.

*14) We included the reference to the table and updated the sentence.*

Page 13 -14 line 1: "adapts to changes in climate with usually PFT specific time scales." What does this mean? Are there variable PFT longevity within simulation?

*15) Changes in PFT distributions are not instantanious the response of vegetation to any climate change is therefore delayed and the delay depends on the PFT specific time scale. We change the sentence to "constant PFT specific time scales".*

Page 14 line 1-2: Include references to examples of DGVMs which include human dimensions.

*16) most of the DGVMs and land surface models do as representing land use change is now really a standard, we therefore do not think that references here are useful. The reference to Hantson et al. (2016) already includes a number of models including human properties. Listing individual model references would only lengthen the references section and would be an arbitrary choice of models.*

Page 14 line 2: ". . .population density is a commonly used driver." Driver of what? Ignitions? Land use change? Please clarify.

*17) We extend the sentence with: ...commonly used driver for human ignitions and suppression of fires.*

Page 14 line 3: Start a new paragraph with the sentence beginning "Our model simulations. . ." and update this sentence to "Our model simulations also show that the modelled climate. . ."

*18) This paragraph was rewritten.*

Page 14 line 6: Update sentence to ". . .not affected by land use or by the type of fire model. . ."

*19) The sentence was removed.*

Page 14 line 7: ". . .seasonality that is not resolved by the mean annual precipitation." The model has no seasonal variation in precipitation and is only using MAP? Please clarify.

*20) This part of the discussion was removed. For clarification: the model uses daily precipitation however the comparison was based on MAP."*

Page 14 line 8-13: Include discussion of how the results differ due to the use of only preindustrial land use. Qualify the text in this section to clarify that the JSBACH sim- ulations use preindustrial land use and these products use recent land use (Andela et al 2017 uses the past 18 years). Explain why the comparison is still valid.

*21) We do not compare the simulations with preindustrial land use to recent satellite products. The decrease in burned area due to land use however is supported by several satellite data analysis. We add a sentence to explain that this is not a direct comparison but refers to the isolation of the effect of land use change: "We seperated the effect of land use change by comparing the historical simulation to a simulation with preindustrial land use. We find that land cover change is influencing the differences in the modelled fire regime between Africa and South America."*

Page 14 line 12: "The mechanism behind the reduction due to croplands. . ." Reduction of what? Fire occurrence? Please clarify.

*22) We added "burned area" to clarify*

Page 14 line 13: ". . .fragmentation of the landscape, which is not explicitly accounted for in the model." Include discussion of how fragmentation affects forests in reality, and how this may be a challenge for models such as JSBACH. Is this an area for potential improvement?

*23) Fragmentation can certainly affect forests and for instance their biodiversity in many ways, as the sentence is about the effects of fragmentation on burned area, we indicate how fragmentation affects fire. Fragementation effects on forests are not mentioned here and for a model such as JSBACH we don't see a direct benefit. The fragmentation effects on fire are however very direct as fragmentation often stops fires from spreading. We add:*
*"Fragmentation of the landscape by for instance roads can act as a fire break and therefore reduce the potential fire size. The exact relationships between humans, land use and vegetation fires are still unknown and therefore not well represented in models."*

[Figure]

Page 14 line 17: ". . .spatially varying ignitions." Do ignitions vary temporally?

*24) The paragraph is about differences in the spatial patterns between continents. The lightning ignitions vary seasonally and the human ignitions vary annually due to changes in population density. However, as we do not address temporal variability of burned area at all in the manuscript and we only evaluate spatial patterns we do not see a benefit of dicussing the temporal variability.*

Page 14 line 18: ". . .these differences in ignitions. . ." Differences between what? One is not spatially varied ignitions? Please clarify.

*25) This again refers to the spatial variations in ignitions, we update the sentence to: "...these spatial differences in ignitions..."*

Page 14 line 32-33: Add at the end of the sentence what the values are for the satellite datasets. It is not possible to read them from the figures to compare to this measure of 100 mm and >650 mm per year.

*26) We add:*
*The remote sensing datasets show for Africa an absence of tree cover for precipitation less than ca. 300 mm and canopy closure for 1500 mm year$^{-1}$ in the model resolution (Figure 4).*
Page 15 line 6-7: ". . .spatial scale needs to be considered. . ." Add discussion on how increased spatial scale (finer resolution) might improve the model results. Why not perform simulation at 1km similar to the Hirota dataset? Should simulation be finer than 1km? How small of a resolution can you achieve before you see compromised results for simulation?

*27) Also here the reviewer asks for answers that can only be adressed by doing simulations in high resolution and testing the influence of model resolution on the simulation results. The only thing we can conclude from our analysis is that some metrics of the comparison depend on the spatial resolution. Such high resolution simulations are also still very computationally intensive and we would not have the necessary computation time available. We now add a sentence to suggest that running the model in higher spatial resolution could improve the performance as the thresholds in the model are closer to the ones found for higher resolution or local scale observations:*
*"Moreover, as the thresholds found for the model are closer to the ones found for site-level and high resolution satellite datasets the model performance could improve if the spatial resolution of the model is increased."*

Page 15 line 11-12: Are there plans to compare to biomass datasets? Identify potential datasets.

*28) Including our plans for future work is an uncommon suggestion and we do not see any use of including them. We add references to datasets (SAATCHI, AVITABILE, BACCINI*:
"... and pan-tropical datasets are available (Saatchi et al., 2011; Baccini et al., 2012; Avitabile et al., 2016)".

Page 15 line 26-28: "The multivariate comparison helped to . . ." Re-word this sentence. It is not clear what is meant by "too strong effect of fire on tree cover". Split into two sentences to identify problems, and then another to suggest improvements. Clarify where and how increased bark thickness can be included.

*29) We changed the sentence as suggested to: "The multivariate comparison revealed a too strong impact of fire on tree cover for gridcells with very high fire occurrence, which leads to too low tree cover. Possible model modifications to boost the tree cover in exactly these regions with high fire occurrence are an adaptation of trees to fire by increasing bark thickness in reponse to high fire frequencies or a stronger negative feedback between fire occurrence and fuel load. This stronger feedback should then reduce fire intensity and consequently fire mortality."*

Page 15 -16 line 1: "although known variations in vegetation characteristics are not represented in models. . ." Provide a brief description of what is not represented? Bark thickness variability, size-structure?   Consider adding a stronger concluding sentence to identify how these improvements will be helpful to models.

*30) This is meant to refer to the differences found for instance by (Lehmann et al., 2014).  These known differences are however not well enough understood to be implemented in models. We changed the sentence to: "Known variations in vegetation are not sufficiently understood to be represented in models.  However, our finding that models do show differences in the fire-vegetation-climate relationships between continents shows that further exploration why models show differences can be helpful to better understand causes for intercontinental differences."  We can only suggest improvements whether they will really be helpful or not needs to be tested with such modifications implemented in models and comparisons of simulations with and without these modifications.*
*We add as a last concluding sentence: "Overall the multivariate model evaluation highlights the potential for more targeted model improvements with respect to the interactions between climate, vegetation and fire, which are crucial for our understanding of future vegetation projections."*

**2 Reviewer 2**

The ability of JSBACH to reproduce the observed relationship between fire, tree and grass cover and mean annual precipitation (MAP) was assessed using two different coupled fire models, with the implicit aim of guiding future model development. Analysis was split between continents to assess different regional climate-vegetation-fire relationships, and using present day and pre-industrial land use to assess the models ability to reproduce human impact on burnt area within bioclimate. The authors successfully demonstrates the potential of this approach by identifying too high tree cover at low precipitations, high burnt area in areas of low tree cover and cropland representation as key model weaknesses, before speculating on likely causes and solutions. The approach is relatively simple but, as the authors point out, is also quite a novel way of identifying areas for improvements in vegetation-fire models which will hopefully be adopted by other modelling groups. I also like that the paper is solely dedicated to model assessment, despite the distraction of including JSBACH-standard (see comments below), and I look forward to seeing if this results in better informed, targeted model improvements in the future. If so, it could be a process the rest of us in the fire modelling community could learn from.

I do, however have a serious concern about the choice of driving data that needs to be addressed before I recommend publication. I also have a few other major comments, although some might just require brief clarification through author response and small changes to the m/s. Given the potential changes to the manuscript required to address the first major comment, I have only included a few key specific suggestion for now, largely for the introduction.

JSBACH-fire was driven using simulated climate from the MIP Earth System Model. However, almost all the evaluation is of JSBACH-fire component alone. This is clearly

a problem for the basic spatial evaluation in most section 3.1 and figure 1, where it is often unclear if mismatches in vegetation cover or burnt area is because of JSBACH itself or because of biases in the Earth System Models (ESM) climate simulation. As the rest of the paper is evaluating JSBACH in climate space, it could be argued that the choice of driving data doesn't matter. However, simulated climate biases could still be playing a role even here. For example, the authors only use MAP as a climate proxy. Inherent in MAPs influance on fire are the extreme conditions, specially the length dry periods, that increases susceptibility to burning. This is part of the reason for the wide range in fire and tree cover at a given MAP in all but the driest and wettest climates, and is invoked by the authors to explain different tree-MAP relationships in Australia.

*31) We agree that biases in the forcing can have an influence on the model evaluation and that the same simulation driven with reanalysis data would have different results. While the traditional variable by variable evaluation for instance shown in figure 1 is highly dependend on spatial biases our approach presented here largely overcomes this limitation. The focus of this paper is on the multivariate comparison that evaluates the model in climate space. We use the standard JSBACH setup, which is the combination of JSBACH with MPI-ESM meteorology. As the fire is sensitive to a number of variables, evaluation of the model in a different setup wouldn't help to guide model development for a model that is almost only run in the coupled setup. The evaluation of the model within the climate space helps to reduce the impact of climate model biases on the model evaluation and therefore to focus on biases in the land surface model. Moreover, our motivation here is to evaluate the land surface model, a detailed evaluation of climate biases in the ECHAM model is therefore out of scope. Understanding potential influences of certain climate biases (such as extremes) on the simulation would require specific factorial experiments. While this would certainly increase our knowledge, it would not lead to an improvement of the coupled model system unless the climate biases can be improved. Mean annual precipitation explains*

*a large part of the tree cover variability and therefore is a useful proxy for climate. Moreover we can relate model-data mismatches to this simple proxy, it is therefore informative. While certainly more parameters influence tree cover distribution an increasing number of variables included to explain patterns would require a totally different approach, as ours is largely based on the possibility to visualize the relationship between the variables. Three variables are a natural limit here (x-y scatter plot + color scale). We introduce our motivation for showing the geographic patterns and for our evaluation approach now in the beginning of the results section: "We first give an overview over the geographical distribution of the used observation and model output datasets. The comparison of geographical patterns is an important assessment of model performance, it is however difficult to assess whether the interactions between precipitation, fire and tree cover are well captured. Moreover as the JSBACH model is usually used as a land surface model for the MPI-ESM and therefore also here forced with MPI-ESM output, biases in model forcing can cause geographical biases of vegetation and fire variables even with a perfect fire and vegetation model. To reduce the influence of biases in forcing data on the model-data comparison and allow to more closely evaluate the interactions between model components we propose a multivariate evaluation of climate-fire-vegetation relationships. We assess the robustness of observed relationships for two tree cover datasets and two spatial resolutions and compare them to the model simulations. The last paragraph of this section adresses the influence of land use change on the simulated relationships."*

**2.1 Choice of JSBACH driving data**

JSBACH-fire was driven using simulated climate from the MIP Earth System Model. However, almost all the evaluation is of JSBACH-fire component alone. This is clearly a problem for the basic spatial evaluation in most section 3.1 and figure 1, where it is
often unclear if mismatches in vegetation cover or burnt area is because of JSBACH itself or because of biases in the Earth System Models (ESM) climate simulation. As the rest of the paper is evaluating JSBACH in climate space, it could be argued that the choice of driving data doesn't matter. However, simulated climate biases could still be playing a role even here. For example, the authors only use MAP as a climate proxy. Inherent in MAPs influance on fire are the extreme conditions, specially the length dry periods, that increases susceptibility to burning. This is part of the reason for the wide range in fire and tree cover at a given MAP in all but the driest and wettest climates, and is invoked by the authors to explain different tree-MAP relationships in Australia. General Circulation Models (GCMs) are notoriously poor at simulating dry periods, with many underestimating the length and/or severity of dry periods due to poor simulation of convective vs persistent rainfall and a problem with persistent dizzle (DeAngelis et al. 2013; Gutowski et al. 2003). Length of dry periods is fundamental in the calculation of ignition probability and each fire's area in SPITFIREs rate of spread model (Thonicke et al. 2010). The "standard" fire model used in this study sounds like it could be similar to GLOBFIRM? If so, this is also very sensitive to number of dry days, with a rapid increase in burnt area in longer dry seasons (Thonicke et al. 2001), which would explain at least part the underestimation of maximum burnt areas. Either way, driving and comparing JSBACH with ESM output could skew the MAP relationships with fire and potentially tree cover in figure 3-6 and A1-2. This is by no means the only problem with driving the model with ESM data, but it is the one that springs to mind. Using MPI also required the authors to make a rather awkward decision between performing comparisons on different time periods (1996-2005 from JSBASH runs; 2001-2010 from observations) or on the few years of model-observation overlap.

I have two suggestion for how the authors could address this problem:

1. Continue to use the MPI driven runs, but reframe the paper to evaluate the processes and identify weaknesses in simulation of tree cover and fire in the ESM as a whole. Some of the arguments I have made above as to how ESMs simulated climate

could affect tree cover and fire could be included. However, there are likely many more, some specific to MPI. If there are any special required configuration of JSBACH to simulate vegetation dynamics under MPI then these should also be included. The authors briefly touch on two arguments that could also be expanded: lines 11-14 on page 6 uses figure 1c to briefly discuss whether MPIs MAP biases as a reason for some of the mismatches between observed and simulated burnt area; and lines 7-15 on page 13 where the mismatches between driving data introduced straight into the fire model (popdensity, lightning etc) and those driven from MPI climate output. Section 3.1 should just need re-formulating with no new analysis. Subsequent sections may require fresh analysis, potentially looking at multivariate relationships within the space of MPIs driving data.

2. Run JSBACH with climate observations, including using common precipitation observations for driving data, and analysis of observed and simulated fire, MAP and vegetation cover. According to (Rabin et al. 2017), JSBACH model output should be included in fireMIP, in which case, vegetation cover by PFT and burnt area from observation driven JSBACH will be available from fireMIP.

*32) See our previous reply. We include some of the points mentioned by the reviewer to improve the discussion on our setup choice (see below). Evaluating the details of climate biases of the MPI-ESM is out of scope of this manuscript. Clearly there can be biases due to climate biases in the simulations of JSBACH. However evaluating the model in a different setup seems less promising and less targeted to us than our approach to evaluate the model in climate space using the setup it is usually used with. As the reviewer also acknowledges we mention the limitation of the input datasets determining the ignitions as inconsistency. However regarding the conclusions we draw from our comparison we don't see a strong point that they would be strongly affected. The reviewer states that the rainfall seasonality is especially important for the "wide range in fire and tree cover at a given MAP in all but the driest and wettest climates". We show that the relation between precipitation and burned area is captured quite*

*nicely at least whith SPITFIRE and for the relation between MAP and tree cover we look at exactly the thresholds of these driest and wettest climates, not at the variability inbetween. For the intermediate rainfall regions we focus on the relationship between tree cover and fire. We don't see a good reason why other climate biases should decrease the correlation between fire and tree cover, which is the point of our focus here. FireMIP simulations in this setup are unfortunately not available. Within the first round of FireMIP simulations (Rabin et al. 2017) the model was set up with prescribed vegetation. For recent simulations we did also similar simulations with dynamic vegetation however this model includes a number of changes, such that the versions are not comparable anymore. We include a paragraph on the model biases in the discussion of model improvements:*

*"Many climate models have problems to represent extremes, length of dry periods and tend to generate a permanent drizzle (DeAngelis et al., 2013; Gutowski et al., 2003). With our approach we only include mean annual precipitation, other aspects of the modelled climate are neglected but might contribute to model-data mismatches in the relationship between precipitation and other variables. Mean annual precipitation is however a strong driver of vegetation patterns especially in the tropics and including more climate parameters would require an entirely different approach and possibly limit visualization and interpretation of the results. Including more climatic parameters could especially help to interpret more of the variability for mean annual precipitation amounts that allow tree establishment but do not lead to complete canopy closure. The reasonable relationship of mean annual precipitation and burned area however indicates either that additional climate biases are not important as fire is quite sensitive to the length of dry seasons or that that the fire model cancels out additional climate biases."*

**2.2 Choice of fire dataset**

Is there a reason for use of GFED4 instead of GFED4 with small fires (GFED4s) (van der Werf et al. 2017)? There may be a good reason for not including small fires, but given the prevalence of GFED4s in other fire evaluation studies (Rabin et al. 2017; Kelley et al. 2014; Kloster & Lasslop 2017), it might be worth including some justification. Also, are the certain weaknesses in fire detection in GFED that might affect the results? The missing smalls fire's for example should be mentioned as a caveat in relation to results from figure 3.

*33) We use here the global burnt area dataset with the highest accuracy (Padilla et al., 2015). The dataset does underestimate small fires, and a recent version of GFED4 (GFED4s) tries to take these into account. However, the small fire detection procedure has been strongly criticized and is highly uncertain see eg, (interactive discussion van der Werf et al. (2017)). Therefore, we decide to use GFED4 for the moment as it has a proven high quality and refrain from using GFED4s until its accuracy has been shown. The spatial patterns of the two datasets are very similar (see Randerson et al., 2012, Figure 7), the main difference is that the GFED4s has a 25% higher burned area. These small fires are often related to croplands or deforestation. The models used here do not model deforestation or cropland fires, therefore aiming at this high burned area that includes these fires would not be an advantage. As far as we know there is no quantification of other weaknesses of the burned area datasets, therefore speculating about the extent and whether they would influence our results would be difficult. We add a reference to an evalutation study and mention the main sources of uncertainties. We add in the discussion section:*
*"The latest release of the GFED burned area and emissions datasets includes an extension for small fires (Randerson et al., 2012). However these small fires are often related to cropland fires or deforestation fires. Neither of these fire types are modelled*

*explicitly in our model approaches and therefore could cause an unwanted mismatch. Cropland fires are not expected to strongly influence the vegetation cover, while deforestation is prescribed as described in the model and simulation paragraphs and therefore the influence on vegetation cover is considered. Burned area datasets are generally uncertain mainly due to the limited spatial and temporal resolution (Padilla et al., 2015), the difference in global burned area between the dataset including small fires and the one not including small fires is 25%. The spatial patterns are less affected, but missed burned areas due to high cloud cover certainly introduces also spatial biases. How important such errors are for a comparison as present here is unknown."*

2.3   Quantification of similarity in multivariate relationships.

The observed relationship between MAP and tree cover is described as either "linear" for Australia and "sigmoid" for other continents, with the ability of each model to describe each curve used as evidence when identifying model weakness. However, I'm not sure I can see these relationships. Observed Australia looks more like the start of a sigmoid (albeit with a shallower gradient when compared to e.g. Africa), "chopped" at low tree covers. South Africa looks more linear. A simple curve fitting and correlation could help determine how closely each continent resembles each function, and if the model is reproducing this relationship, which would place subsequent discussion on firmer ground.

*34) This is an interesting idea and indeed the purely visual comparison was indeed not that firm. As likely only the maximum tree cover for a certain precipitation amount is limited by precipitation and lower tree covers are likely modified by other factors we used a quantile regression to characterize the relationship between precpitation and*

[Figure]

*maximum tree cover. We use a linear regression and a local regression to illustrate the difference between the linear and nonlinear/sigmoid increase. We include a paragraph in the methods section:*

*We use quantile regressions to characterize the relationship between precipitation and maximum tree cover. The quantile regressions were computed with the R package quantreg (Koenker, 2018). We use the local quantile regression to characterize the shape of the increase in maxmimum tree cover for increasing precipitation. Moreover we quantify the deviation from a linear increase by also including the linear qunantile regression. Both regressions were computed for the 0.9 quantile. For the local quantile regression the bandwidth parameter was set to 300 and the number of points where the function was estimated was set to 10.*

*Adopted the paragraph in the results section:*

*"Models and observations show differences between continents in the relationship between precipitation and maximum tree cover (Figure 5). For Africa, South America and Asia the relationship between maximum tree cover and precipiation shows a saturation for high precipitation. For Australia maximum tree cover increases linearly with increasing precipitation for models and observations, but the precipitation range also does not reach values where a clear saturation is reached for the other conti- nents. For JSBACH-standard the curves are very similar for the different continents. JSBACH-SPITFIRE shows a stronger variation, this must be due to the differences in fire as the model is otherwise the same. The observations show an even stronger variation between continent, with clearly lower tree cover valsue for Australia followed by Asia. For Africa local quantile regression clearly differs from the linear quantile regression for the satellite data, indicating a sigmoid shape, while the other continents show a rather linear increase until the saturation (Figure 5). JSBACH-SPITFIRE reproduces the higher tree cover for South America compared to Africa for mean annual precipitation lower than 1000 mm, but also JSBACH-standard shows a small difference."*

*In the discussion we remove the paragraph on the disucssion of the linear increase*

*for Australia in comparison to the Lehmann et al. (2014) and focus on the point that models do also show some differences between the continents. We add in the end of the discussion:*

*The comparison of the increase in maximum tree cover with increasing precipiation shows that the model shows some variability in climate-vegetation-fire relationships between continents, it misses a large part of the variability. Finding the correct balance of the many influencing factors, e.g. climate, fire, land use, evolutionary differences, will remain a challenge for the future.*

The remaining multivariate comparisons is also largely based on visual comparisons on plots. While this is an important part of assessing differences in simulated vs observed relationships, I feel like the comparison could do with some quantification using some simple multivariate metric, expanding on the two-variable assessment in Table 1. I am by no means an expert in multivariate statistics though, and perhaps a "simple" comparison isn't possible. But if the authors have any thoughts on this, it would be good to hear (and perhaps include them in the m/s?)

*35) This is a very interesting suggestion. However we believe that such a metric would still need to be developed. We did not find an applicable, promising approach in a web search. Also we are not sure what this multivariate metric could represent. Correlations can only capture linear, rank correlations monotonic relationships. This made sense for precipitation and tree cover, the relationship with fire however is more complex. Probably an approach based on regression methods, including also nonlinearities could be a way forward. This seems promising to us, however it would deserve more attention and in depth testing. Nevertheless we now also include the rank correlation between fire and tree cover for a certain precipitation and tree cover range in the Table 1 in addition to the correlation between precipitation and tree cover. This quantifies the stronger impact of SPITFIRE on tree cover compared to the observations and also reveals that JSBACH-standard has a reversed relationship*

*between fire and tree cover, likely due to the exclusion of pastures for burning.*

2.4   Use of two models

More could be made of the use of the "standard model" (JASBACH-standard) to help analyse MPI, JSBACH or even SPITFIRE performance, which is obviously the model the authors will use in future studies. As a start, JSBACH-standard could do with a little bit more description to help inform later discussion. Are the curves describing relation- ship between relative humidity, fuel carbon and fire similar to GLOBFIRM (Thonicke et al. 2001), or are they more similar to those simpler rate of spread models such as CTEM (Arora & Boer 2005). Are parameters used by the model based on literature, site comparisons, or optimization of remote sensing? If the latter, is its poor perfor- mance likely due to biases in JSBACH simulation of vegetation or MPI climate and dizzle biases? If the former, is it additionally due to fire model structure or bad pa- rameterizations? How much is PFT fraction remove after fire? Is 100% of burnt PFT removed, or just a fraction? If a fraction, does this vary? And does it vary by life form, PFT, burnt areas or some other relationship?

*36) The simple fire parameterization is described as: "The JSBACH-standard fire computes burned area based on a minimum burned fraction which increases as a function of the litter carbon pools and relative humidity averaged over the last three weeks." And there is really not more in terms of burned area. So probably it is closer to GLOBFIRM, but it is also unclear how to quantify whether it is close to one or the other model. We add that the model was tuned to yield reasonable global emissions estimates and improve the tree cover, there was no comparison with site level or remote sensing products. We already included that:*

*"In the JSBACH-standard fire scheme the burned area directly translates into a reduction of the cover fractions of the plant functional types (PFTs) ..."*
*we add: "(100% of the cover fractions on burned area are removed)"*

A better comparison between the two models in the discussion and/or conclusion might also further strength the case for use of multivariate approach. Despite its poor per- formance is there any part of the standard model multivariate relationship that could be used to guide development of SPITFIRE, particularly with respect PFT tree mor- tality? Is there any conclusion that can be drawn on the use of complex fire models to represent complex processes such as fire and fire-feedbacks, or does any part of the standard models performance (i.e, locations of fire occurrence) suggest that emer- gent behaviour of fire on coarse scales does not require the use of complex models? Does a comparison of strength and weakness of the two models say anything about the coupling to JSBACH or required configuration for use of JSBACH-fire in MPI?
Ofcourse, if the authors feel like nothing substantial can be learnt from comparing the two models, then they could consider removing JSBACH-standard from the m/s. However, there is nothing technically wrong with its inclusion, so I'll leave that for the authors to decide.

*37) The comparison with the simple (poor permorming) fire model mainly shows that improvements in the fire model lead only to small improvements in vegetation patterns. We think that answers to the very specific questions of the reviewer would be highly speculative and would require additional analysis. It is also often unclear how much difference in performance is due to better tuning and how much due to a better model structure, as none of the models is optimized.*

[Figure]

**2.5   Specific comments**

Page 2, line 24-25: The development of complex fire models actually started before widespread use of remotely sensed products. MC2 (Lenihan et al. 1998) forms the basis for most rate of spread models (Hantson et al. 2016), and SPITFIRE is itself a development of Reg-FIRM (Venevsky et al. 2002). Neither invoke the use of remote sensed burnt area.

*38) We modify the sentence, the recent implementations of SPITFIRE for instance have made strong use of satellite data: "The development of remotely sensed global burned area products facilitated the implementation and evaluation of complex fire models within DGVMs (Hantson et al., 2016). "*

Page 2, line 30: "the importance of benchmarking effects on vegetation has been noted". Not just noted, but also done (Kelley et al. 2014).

*39) We extended the sentence in the revised manuscript with:*
*... and applied in model development studies (Kelley and Harrison, 2014; Lasslop et al., 2014)*

Page 3, line 11: Please use -180 to 180 coordinates longitudes.

*40) We updated the coordinates for the South America region.*

Page 4, line 2-3: Replace (or include along side) "(Rabin et al. 2017)" with "(Thonicke et al. 2010; Lasslop et al. 2014)". (Rabin et al. 2017) does provide description of SPITFIRE alongside several other fire models, but the authors should also give credit to the model developers.

*41) We refer to the two older publications in the beginning of the paragraph, where SPITFIRE is mentioned the first time. (Thonicke et al. 2010 is included there now too.) Rabin et al. provides the most up-to-date, complete and detailed description and therefore deserves a reference too.*

Page 4, line 14: "During the 1000 year spin up period . . . " How was the spin-up determined? Where carbon or PFT fractions/burnt area in equilibrium by this point? How was this tested?

*42) The spin-up period was determined based on experience, we did not apply a formal test criterium. PFTs are largely in equilibrium after 1000 years, small changes especially between woody PFTs can still take place in some grid cells. Global tree cover on the other hand equilibrates after around 300 years. We included:*
*"At the end of the 1000 years PFT distribution was largely in equilibrium with only minor shifts between woody PFTs in few grid cells."*

Page 5, line 31 - Page 4 line 1: Please add a citation to the r-package paper. I think (Tuck et al. 2014) is the correct reference, but the authors should check. Also include a direct reference to the r package used. Typing in the following in an R terminal should give you the require bibtex information:
Âż citation(Âńpackage nameÂż)

*43) We included the publication indicated with the citation command (Mattiuzzi and Detsch, 2018). Tuck et al. 2014 seems to be the citation for the MODISTools package.*

Page 5, lines 12-13: Is any scaling applied when translating from LIS/OTD flash count to ignition sources? I.e, are cloud-cloud flashs removed?

*44) Cloud-cloud flashes are removed, however, we refer to the model description papers for the exact model formulations and here only document the input files. We also don't detail how population density is converted into ignitions and want to keep a similar level of details for the lightning ignitions.*

Page 11, line 10 - Page 12, line 1: (Kelley et al. 2014) collected cite based bark thick- ness data to reparametrize bark thickness in a SPITFIRE based model. There might also be some Australia specific improvements in this paper that could be considered.

*45) Yes, we include the reference to Kelley et al. 2014 for the bark thickness as an adaptive trait and the resprouting mechanism which acts in a similar way to increase tree cover:*
*"Kelley and Harrison (2014) included bark thickness as an adaptive trait in the LPX model, which increased and improved the tree cover for Australia. Resprouting is another important mechanism that leads to an increase in tree cover in fire affected areas (Kelley and Harrison, 2014)." see also reply 10.*

Page 14, lines 21 - 23: The non-independence of vegetation cover datasets should be included when introducing the datasets on page 4 and 5.

*46) We mention the similarity of the two datasets in the methods section in the revised manuscript.*
*"The datasets rely on different sensors, however, the algorithms to derive vegeta- tion cover are very similar and the datasets therefore not completely independent. Nevertheless using the two datasets can give a first insight on the robustness of the investigated patterns."*

Page 17: Please complete author contributions.

*47) We completed the author contributions.*

**References**

[revised manuscript text omitted]

van der Werf, G. R., Randerson, J. T., Giglio, L., van Leeuwen, T. T., Chen, Y., Rogers, B. M., Mu, M., van Marle, M. J. E., Morton, D. C., Collatz, G. J., Yokelson, R. J., and Kasibhatla, P. S.: Global fire emissions estimates during 1997–2016, Earth Syst. Sci. Data, 9, 697–720, https://doi.org/10.5194/essd-9-697-2017, http://www.earth-syst-sci-data-discuss.net/essd-2016-62/https://www.earth-syst-sci-data.net/9/697/2017/, 2017.

---

## Referee Report (RR1)

Most of the author response and/or manuscript changes adequately address the major comments from the last review. I particularly like the quantile regression analysis to determine the maximum tree cover for a given MAP (see comments below). I am also satisfied with the responses and changes to the paper addressing the choice of fire data and the analysis of both fire models. However, I remain concerned about the response to the "choice of JSBACH driving data", although after clarification on the aims of the analysis, I am no longer sure if the this is an issue with using ESM model output itself or the way this is used to infer areas for improvement of JSBACH. Below I respond to each original point in turn, before including a small number of additional minor corrections.

**Choice of JSBACH driving data**
The authors have clarified the three aims:
1. Develop a simple multivariate technique to explore the difference between modelled and observed vegetation, fire and climate.
2. Use these differences to evaluate the simulation of, and coupling between, tree cover and burnt area in JSBACH
3. To do this within the MPI-ESM framework, achieved by driving JSBACH offline but with MPI-ESM model output.

There is nothing wrong with the aims, and I like that the authors attempt to at keep the methodology relatively simple. Driving the JSBACH as configured for use in MPI-ESM, offline with ESM output also makes sense, and in their response and suggested changes to the manuscript, the authors have justified the choice of driving data. The authors also discuss the weaknesses associated with this method in the revision of section 4.1, which is also a welcome addition to the paper. However, to critique solely the land surface component in an ESM setup such as this, as if it were independent of other potential climate model biases seems to contradicts the 3rd aim above and introduces the methodological inconsistency which I don't feel have been adequately addressed. To phrase in terms of the multivariate approach, the authors have diagnosed the vegetation cover and fire axis, but not the climate axis. The authors state in their response that "*regarding the conclusions we draw from our comparison we don't see a strong point that they would be strongly affected.*" Here are just some examples from the (revised) paper where climate biases could potentially affect either the results, discussion and/or conclusion:

1. *Surprisingly the observations show a higher Spearman correlation between tree cover and precipitation than the models (Table 1). The lower correlation of the modelled relationship most likely originates from the lower precipitation regions (<500 mm year-1 where the maximum tree cover is very low in the observations and both models strongly overestimate the maximum tree cover (Figure 4).*

   The correlation between MAP and other climate variables that influence tree cover could also break down in the MPI-ESM driving JSBACH. As already noted, length of dry seasons are likely to be shorter in seasonal climates. Most GCMS models (although I

don't know if MPI-ESM is amongst them)  also suffer from biases in downward SW  (Li et al. 2013) which could influence tree cover, particularly at the higher tree cover range, where figure 5 also indicates mismatches beetween model and observation in some continents, particularly Asia.

2. *JSBACH overestimates tree cover for low precipitation on all tropical continents.*

The drizzle problem already discussed seems like an obvious candidate to affect vegetation cover at low precipitations, either through decreasing the length of dry periods or due to associated changes in cloud cover changing evaporative demand and hence available moisture. Despite not ruling out additional climate problems, the authors use this simulated mismatch at low tree covers to justify planned changes to tree to vegetation dynamics:

*only if a 5 year average of NPP turns negative, drought effects on the dynamic vegetation take effect. Other models require a minimum of 100 mm year-1 precipitation for sapling establishment (Sitch et al., 2003). The too high excessive tree cover could be partly improved by improving the non-vegetated fraction which decreases too fast with increasing precipitation*

and

*Tree-grass competition for water could for example be improved in the model by introducing the a sapling stage of trees, which are competitively inferior to grasses (D'Onofrio et al., 2015). Including this mechanism could improve the balance between tree and grass cover, but it could also reduce the establishment rate of trees and therefore the tree cover in the dry regions with excessive tree cover. Including a PFT-specific rooting depth of vegetation would be an important extension of the model to improve the competition for water between grasses saplings and adult trees.*

These three fundamental changes to the dynamics of JSBACH are suggested without establishing that the problem is with JSBACH itself. While it is often necessary reparameterize components of ESMs to compensate for biases in other model components, this should always been done in the knowledge that it is to compensate for other these other biases, and the suggested changes to JSBACH above go beyond a standard re-parameterization.

3. *For Australia underestimation of burned area for both fire models is strong (Figure 4). In a previous evaluation where the model was forced with observed climate and vegetation cover was prescribed (in contrast to the dynamic vegetation cover and climate modelled by the MPI-ESM) JSBACH-SPITFIRE showed better results for Australia (Hantson et al., 2015). An improved response of vegetation cover dynamics to precipitation will therefore likely improve the patterns of burned area.*

The better simulation of fire in Hantson et al. 2015 could also be due to better representation of rainfall timing and distribution, temperatures or any number of climate factors from being driven by observed climate. Also, better representation of vegetation cover would hopefully have been achieved in Hantson et al. 2015 with observed rather than simulated climate. Again, parameterization of either JSBACH or SPITFIRE to account for additional climate biases may be necessary in an Earth System model, but here the author imply the the problem is with JSBACH itself.

4. *This indicates that not an improvement of the fire model but improved modelling of drought effects on the vegetation dynamics will improve the response of vegetation to climate in dry regions.*

Again, another likely explanation is MPI-ESM rainfall distribution or the impact of other climate factors on available moisture etc.

5. *Intercontinental variation in the relationship between precipitation and maximum tree cover is much smaller for the models compared to the observations. Known variations in vegetation are not sufficiently understood to be represented in models. However our finding that models do show differences in the fire-vegetation-climate relationships between continents shows that further exploration why models show differences can be helpful to better understand causes for intercontinental differences.*

If this is meant purely for land surface modelling, then there is little in the results of this paper to justify this statement. That there is a modelled difference in fire-vegetation-climate relationships between continents would be more valid if the authors made it clear that this statement is about the ESM setup as a whole.

6. *Overall the multivariate model evaluation highlights the potential for more targeted model improvements with respect to the interactions between climate vegetation and fire, which are crucial for our understanding of future vegetation projections.*

Again, this is fine as a statement about the ESM setup as a whole, but not focusing solely on the land surface component.

Of course, there are more suggest model improvements in the manuscript where inherent climate biases from MPI-ESM have (to my mind at least) no obvious impact. However, even in these cases, the authors should be careful at presenting potential new model processes without first checking for the influence in other climate biases. The apparently stronger correlation between fire and tree cover compared to observations, for example, is used to suggest inclusion of resprouting and adaptive bark thickness or fuel feedbacks that might influence fire intensity and hence tree mortality. Again, there are no end of climate biases that could affect intensity which would not be picked up by

a straight MAP-tree cover-bunt area comparison. And again, these changes go far beyond standard reparameterization of a land surface model in an ESM. To be fair to the authors, they have included the statement "*exact parameterization and needs to be tested with stepwise model development and factorial simulations*" which does help mitigate some concern with model changes such as this.

The authors suggest in their response that only way to address this contradiction is the do detailed assessment of the atmospheric component of the model, or perform complex experiments or analysis using additional model driving data. This is almost certainly not the case, and it would be a shame if further revisions did make the analysis more complicated. However, some additional, simple analysis might resolve the issue. Here are some examples based on the author responses:

- *We use the standard JSBACH setup, which is the combination of JSBACH with MPI-ESM meteorology. As the fire is sensitive to a number of variables, evaluation of the model in a different setup wouldn't help to guide model development for a model that is almost only run in the coupled setup.*

  A run with observed climate obviously wouldn't be used as a basis for further model development if your aiming to improve JSBACH when driven with ESM meteorology. But it would help the authors determine if the deficiencies already identified are due to simulated climate biases or due to the vegetation component, and would place their discussion on much firmer ground. This is part of the justification for offline land surface model runs required for MIPs associated with CMIP6, e.g (van den Hurk et al. 2016; Lawrence et al. 2016).

- *our motivation here is to evaluate the land surface model, a detailed evaluation of climate biases in the ECHAM model is therefore out of scope*

  There is no need to do a detailed evaluation of climate biases in ECHAM (which would indeed be out of the scope of this paper). However, the authors should ascertain if problems in land surface simulation are caused by either problems in the simulation of the land surface or problems with the information it receives from the atmospheric component - a rather basic first order assessment of any land surface model within an earth system framework. I was able to give a few pointer to potential climate biases from my limited knowledge of climate intermodel comparison literature in my last review. The authors should be able to identify other MPI-specific climate biases that they could at least discuss if not to test. As stated in the last review, there are two of instances where climate model deficiencies are discussed (i.e, when explaining discrepancies in simulated spatial patterns and when discussing calculation of lightning ignitions). At the very least, these types of discussions should be included when critiquing the rest of JSBACH.

- *Understanding potential influences of certain climate biases (such as extremes) on the simulation would require specific factorial experiments*

  and

  *While certainly more parameters influence tree cover distribution an increasing number of variables included to explain patterns would require a totally different appraoch*

  Not necessarily. A first step could be to simply show if other climate information (no. dry days, downward SW etc) are causing some of the relationships you see using the exact same approach used for MAP. It may well be that this shows that using MAP alone does do a sufficient enough job as a proxy for climate space, which will then support the rest of the papers discussion. If not, then any additional climate variable that explains some discrepancy could be included in the same way that grass and tree cover are interchanged at various stages in the manuscript.

- *Mean annual precipitation explains a large part of the tree cover variability and therefore is a useful proxy for climate*

  Obviously not enough in JSBACH when driven by MPI climate data - the range of TC at a given MAP is one of the features JSBACH as driven by MPI data does not replicate, and there is no other result to help indicate how much of this discrepancy is due to simulation of vegetation cover or other climate biases.

While I do not expect the authors to address all these points in the manuscript (that would be a very long paper!), I hope that I have demonstrated they are certainly not without options. Picking up on one or two of these point, or anything else which can either show MAP really is enough by itself to account for all other climate biases or that can truly attribute problems with model performance to either JSBACH or MPI-ESM climate, will be sufficient.

The authors do make a good point in their response regarding the performance of SPITFIRE along the MAP gradient. If an aspect of the model is ineed performing well in this analysis, then given that the authors aim is to have an improve JSBACH for use within an ESM, it makes sense no other investigation is indicated - with the normal caveat that there could be a degradation in performance after development of another aspect of the model (which in this case would include MPI-ESM as a whole). The authors also make it clear that fireMIP runs of JSBACH would not be an appropriate direct source of analysis.

As the authors are only able to use MPI-ESM model output till 2005 to drive JSBACH, they have to make a rather awkward choice about comparison time periods, as identified in the last review. An ideal solution to this would be to run MPI-ESM beyond 2005, something that could be happening as part of CMIP6 simulations? However, I realise that this is probably not possible,

and the MPI-ESM may well be configured differently for CMIP6 simulations.  I would like to hear to authors thoughts changing the comparison periods though. The authors state that  *"Using only the overlapping period (2001-2005) would decrease the robustness of the mean fire regime and climate characterization".* This is certainly true for fire regime. However, tree cover is normally more stable, and as trees take a few years to establish, the cover found during 2001-2005 would of also be a consequence of burnt area and climate before this period. Perhaps a better choice is to split comparison periods based on variable rather then on model/observation. i.e, when performing analysis, take modelled and observed burnt area and climate from 1996-2005 (climate data, MPI and GFED overlap) and tree cover from 2001-2005. While this is still a rather pragmatic solution to the mismatch in modelled and observed time periods, it might make more sense then the pragmatic solution outlined in the manuscript?

**2.2 Choice of fire dataset**
The explanation in the revised manuscript demonstrates that GFED without small fire is the most appropriate choice for this study.

**2.3 Quantification of similarity in multivariate relationships**
The quantile and local quantile regression, while simple, help strength some of the main discussion points about intercontinental differences in multivariate relationships. To be honest, I thought the authors might include a description of this in future work, so it was a pleasant surprise to see that analysis was actually included in the paper.

**Re: multivariate metric**
Thank you for investigating this. I agree that any further attempt to develop a multivariate metric is too much for one paper, and given an appropriate metric could not be found by either the authors or myself, the inclusion of the extra spearman rank coefficient between fire and tree cover is certainly enough for this paper.

**2.4 Use of two models**
It appears I overestimated the complexity of the standard model, and the additional information is more than enough. I also accept that additional comparisons between JSBACH-SPITFIRE and JSBACH-standard could become too speculative. However, I note the authors have started to make more use of JSBACH-standard through revisions in response to reviewer 1 - when describing changes in burnt area and tree cover in dry regions in section 4.1 for example.

**2.5 Specific comments**
I am happy with all changes except for a couple of small details:
   ● 39 and 45) Kelley and Harrison 2014 should probably be changed to Kelley et al. 2014:

      Kelley, D. I., Sandy P. Harrison, and I. C. Prentice. "Improved simulation of fire–vegetation interactions in the Land surface Processes and eXchanges dynamic global vegetation model (LPX-Mv1)." Geoscientific Model Development 7.5 (2014):

2411-2433.

Kelley and Harrison 2014 looks at future changes in fire, whereas Kelley et al. 2014 is the paper describing model development and benchmarking.

- 42) Was the "minor shifts between woody PFTs in a few cells" quantified? Quantifying equilibrium during spin-up should really be a requirement for any modelling study, and if authors did quantify equilibrium in any way, then it would help the cause to state how this was assessed. However, I do understand that quantifying equilibrium is unfortunately not standard practice, and finding a way to do so is well outside the scope of this paper. So if it was not quantified, then leave this sentence as is.

- 43) Thank you for including citation this. And thank you for including the reference to the r-package used for quantile regression. Statistical software development is a tough job, and I'm sure the developers will appreciate the extra citations for their work.

Hurk, Bart van den, Hyungjun Kim, Gerhard Krinner, Sonia I. Seneviratne, Chris Derksen, Taikan Oki, Hervé Douville, et al. 2016. "LS3MIP (v1.0) Contribution to CMIP6: The Land Surface, Snow and Soil Moisture Model Intercomparison Project – Aims, Setup and Expected Outcome." *Geoscientific Model Development* 9 (8): 2809–32.

Lawrence, David M., George C. Hurtt, Almut Arneth, Victor Brovkin, Kate V. Calvin, Andrew D. Jones, Chris D. Jones, et al. 2016. "The Land Use Model Intercomparison Project (LUMIP) Contribution to CMIP6: Rationale and Experimental Design." *Geoscientific Model Development* 9 (9): 2973–98.

Li, J-L F., D. E. Waliser, G. Stephens, Seungwon Lee, T. L'Ecuyer, Seiji Kato, Norman Loeb, and Hsi-Yen Ma. 2013. "Characterizing and Understanding Radiation Budget Biases in CMIP3/CMIP5 GCMs, Contemporary GCM, and Reanalysis." *Journal of Geophysical Research, D: Atmospheres* 118 (15): 8166–84.

---

## Referee Report (RR2)

The authors have included new figure to determine if there is a bias in rainfall distribution simulated by ECHAM. This has helped rebalance the analysis across variables in their multivariate analysis. I do have some specific issues with the new figure, which I also feel could be used more in discussing weaknesses in the model performance. I also pick up on some of the author responses that could have some bearing for revision in the rest of the paper, and provide some examples of quick analysis to assess or rule out two remaining climate biases described in the last review.

*Figure B1*

1. The figure uses CRU-NCEP precip observations, whereas TMPA is used for observed precip elsewhere in the paper. The same climate observations should be used for both, especially as the new figure is used to diagnose differences in climate relationships between the two different climate axes in figures 2, and 4-7. Apologies if I implied in an earlier review that it would be okay to use different precip data for different parts of the analysis - I used fireMIP as an example of an offline JSBACH-SPITFIRE simulation, and thought that it would self evident that if used, observations should still be consistent across different parts of the analysis.

   While the authors may argue that choice of dataset might make little difference to the relationship, the disagreement in precipitation between observed datasets in notorious (Beck et al. 2017; Weedon et al. 2014). A quick plot of MAP vs no. dry days I conducted with CRU TS3.2 (Harris et al. 2013) (data I chose for no other reason that I already had it downloaded, not because I'm recommending it for use in the m/s) compared to CRU-NCEP used in figure B1 shows what I mean:

[Figure]

Figure 1: CRUTS3.2 MAP vs no. dry days for tropics and Australia, based on coordinates provided in section 2.

The relationship between MAP and no. drys days for CRU by itself is clearly different, and would actually agree more with ECHAM annual precip. TMPA may also show a significantly difference relationship as well.

2.  The authors already produces an excellent style of figure to diagnose burnt area vs precip in figure 3 which could have been used here with the x-axis displaying precip and the y-axis cumulative days at a given precip level. This would provide more information on rainfall distribution biases. However, there is nothing particularly wrong with the simple scatter plot used, so I'll leave this as a suggestion rather than a requirement.

3.  If the author's choice to stick with the scatter plots, then please add a trend line.

4.  Remember that, for SPITFIRE, impact of dry days in cumulative, so cumulative dry days might also be worth considering, especially as this is another area MPI has been shown to sometimes struggle with (Sillmann et al. 2013). If the authors are able to use the variables already in figure B1 effectively though, then again this won't be

required.

5. The authors need to use this figure to help diagnose climate relationships in more detail. In their response, the authors state that "*This analysis ... shows that the number of dry days in dry regions is well comparable between model and CRUNCEP, for moister regions the number of dry days is even higher in the forcing dataset (MPI-ESM output) used here. We therefore confirm that our conclusions are unlikely affected by biases of rainfall seasonality.*" If this relationship holds once the figure is redrawn with TMPA, then the "anti-drizzle" bias in MPI is surprising. However, it is still a climate bias that will affect simulated fire and possible vegetation, and should be discussed as such in the main text. If it turns out that MPI-ESM agrees with TMPA dry days, than the text will be fine as it is.

**Author responses**

Author responses in italics. My response to the responses in normal font.

*The main concern of the reviewer with respect to the climate biases is the seasonality of the rainfall.*

I use seasonality as an example, and it was not the only or main concern.

*Of courser biases always exist, here, however, it is important whether the climate biases could have such a strong effect as the reviewer claims.*

This is correct. I have no idea how strong an affect the climate biases have. As the authors are presenting a new way of evaluating land surface in ESMs, they need to demonstrate that the impact of other climate biases is either negligible or can be accounted for.

*Shortwave radiation does not affect the tree cover in JSBACH, we quickly tested it by applying a multivariate regression, precipitation is highly significant, radiation is not significant if only these two variables are used in a multivariate linear regression. As so far there is no discussion on shortwave radiation and how it influences the model in the paper, we did not include this in the manuscript as it would require several paragraphs to be added.*

And

*Radiation could have a considerable influence on the productivity of PFTs, but is very unlikely to influence tree cover in JSBACH for the tropics based on the way the model is build. We tested this also quickly with a multivariate regression TC=a1\*P+a2\*R for the modelled variables where the influence of radiation is not significant. It is therefore unlikely that biases in radiation would show up in tree cover. We now show that the number of dry days is not less in the ECHAM forcing. See also reply 11 and 12.*

Was this test with just JSBACH, or for observed tree cover/climate as well? Obviously if SW does not have a significant effect on JSBACHs simulated tree cover but does on observed, then this would be a useful missing climate-vegetation relationship that would need to exploring. If it was tested for both model and observation, then the authors point stands.

*Our proposed method clearly goes beyond the normal variable by variable comparison. Including all variables that might be important in the coupled system of fire, vegetation and climate would be optimal in a certain sense but would then suffer from the complexity of the necessary approach and difficulties in interpretation. As stated in the manuscript we use precipitation as a proxy for climate and precipiation is included as one of the axis. The same critisim, that there could be biases not in the mean but in another characteristic of precipitation, could apply to fire and vegetation cover. We simply use annual burned area as a proxy for the fire regime, but fire intensity and seasonality and extremes can be important characteristics too. For tree and grass cover we also summarized two PFTs into one variable*

Although the authors have only used burnt area for the fire axis, assessment and suggested improvements have borrowed a lot from previous model assessment and literature. In response to reviewer 1s comments, they also have started exploring fire intensity (figure C1). Obviously PFT fractions are always going to be grouped into just three (tree, grass, bare) fraction types for observational comparison, but each were assessed, which gave some grounding for suggested changes in vegetation dynamics, at least from the land surface bias side. Land use experiments also help explore this impact of changing anthropogenic land cover in JSBACH - again part of the vegetation axis. There is also extensive discussion of changes in plant physiological traits and vegetation dynamics and vegetation-fire feedbacks. And this maybe the key to the problem. i.e, the number observed datasets + number of

variables assessed + past model evaluation + literature + suggested model deficiencies and potential development that has gone into the fire and vegetation axis is extensive, but there is much less detail on the climate axis. And that any mismatches in the multivariate pattern compared to observations are almost always assumed to be because of vegetation and fire biases and not climate. This can be properly balanced by proper discussion of figure B1, and/or reference to MPI climate assessment and climate biases.

*A reduction in tree cover would lead to an increase in burned area, therefore what we write is correct. Or vice versa the high burned fraction observed in Australia cannot be achieved with SPITFIRE if such a high tree cover is present.*

The argument that burnt area would increase with reduced tree cover is fine. That the ESM needs to reduce simulated tree cover in Australia is also fine. The problem is the statement that "*An **improved response of vegetation cover dynamics to precipitation** will therefore likely improve the patterns of burned area*" has not been demonstrated. I suspect improved vegetation response would be useful, but I also suspect that biases in MPI climate also share some of the blame. If a change in vegetation cover dynamics is induced with is used to improve fire by compensating for any climate bias, then this is not an improved response but a pragmatic tuning and should be identified as such. Figure 1c shows too much rainfall in Northern Australia, so the authors could already use some of their original analysis to diagnose precip as one potential climate bias that would affect tree cover and burnt area.

In terms of regional climate biases not taken into account by MAP, it might be that figure B1 isn't very helpful yet. In figure 1 in this review, for example, the slope of the fit line, spread of the data, and deviation from linear fit at low precips is different for Australia compared to the spread for the whole tropics.

*Also the reviewer does not give any references that climate model biases can have such a big effect. Of course any of the climate parameters used can be wrong, but the same would be true for any observational dataset used as model forcing.*

Apologies for not providing references in the previous review. The authors may want look at and cite (Ahlström et al. 2017). Although exploring the carbon cycle rather than vegetation cover, they did show a significant impact of precip, temperature and SW biases on simulated vegetation in CMIP5 models. Focussing on the Amazon, (Ahlström et al. 2017) showed MAP, SW and temp climate biases explain most of the simulated GPP, above ground biomass and tree cover. (Ahlström et al. 2012) also showed similar results for disagreement in projected changes in different climate variables into the future. These are just the ones I can think of off the top of my head, there is probably many more. As GCMs have been around for a lot longer, there is of course extensive literature on climate biases that could potentially lead to problems with vegetation dynamics once enabled. (Sillmann et al. 2013) might be a good starting point.

The authors could use (Li et al. 2013) to support their view that only MAP needs to be considered for tropical vegetation distribution, as they use observational constraints to show MAP is the main driver of disagreement in vegetation productivity across models in a region of similar extent to southern America used in this study. However, it should be noted that other climate biases appear to become more important at high MAPs, where vegetation productivity is predominantly limited by available radiation (Nemani et al. 2003). I don't know enough about vegetation dynamics (in model or real world) to know if this tipping point between MAP and SW limited production occurs when tree cover is already saturated. If it does, then maybe (Li et al. 2013) would suggest that other tree cover controls don't need to be considered, at least for this region.

I'm not so sure about the impact of climate biases on fire, as this is a little outside my area of expertise. However, I get the impression that, even with wind speed limitation, SPITFIRE is sensitive to variations in windspeed, especially at lower speeds (Lasslop et al. 2014), which again, GCMs struggle to adequately simulate.

*In regions where fire is absent trees always win the competition in JSBACH, it is therefore impossible that other climate factors can solve this, the only reasonable reason is the absence of drought effects on vegetation cover in the model.*

Again, the authors need to back these statements up by showing in some way that other climate biases are not the issue here. As they are unable to run JSBACH with climate

observations, perhaps offline runs could be referenced in other papers. For example, JSBACH seems to simulate too much tree cover at low MAPs in the offline study by (Baudena et al. 2014). If this was an appropriate test with no fundamental developmental changes compared to the JSBACH configuration used in the m/s, then the authors could cite this study to back up their suggestion of improved vegetation dynamics at MAP. The authors should have a much better idea of published JSBACH and MPI experiments and evaluation, so might also be able to think of better examples.

*This comment is unclear, the variations that are mentioned are observed and the model also shows some variations. We do not see how the ESM setup as a whole comes in here.*

I was just reinforcing that fact that the climate axis should be considered as much as the vegetation and fire axis. I meant "ESM setup" as a land surface model driven by ESM output that is emulating a full ESM, obviously without the land-atmosphere feedback (I'm not sure that makes it any clearer…?).

*Climate biases can clearly influence the burned area, and its spatial patterns, but I do not see a way that climate biases will turn around the impact of fire on tree cover that much in SPITFIRE, except for the fire-fuel feedback mentioned by the reviewer here. This feedback is already included in the model and different climate forcing leading to different fuel loads could maybe strengthen the feedback. However, in that case it would make sense to reparameterize the model to strengthen the feedback in the Earth system model setting*

The point is more to show that the cause of low tree cover is fire feedback in the first place, and not other climate biases (though the author are right that maybe the impact of climate biases on fire-feedbacks should also be a concern...?) If the authors can show climate biases beyond MAP isn't to blame, then the suggested changes fire-feedback are fine.

*Precipitation is the main driver of vegetation cover in the tropics. Removing the main driver from this analysis and exchanging it with other potential climatic drivers that are correlation with Precipitation would likely lead to correlations between vegetation and the climatic driver mainly because of the correlation between the two drivers. The relationship would then still*

*be caused by precipitation. We do not see a way for a useful interpretation of such relationships without removing the effect of precipitation, which would require a more complex approach. Exchanging tree and grass cover is different as both are mainly driven by precipitation and fire.*

I was more thinking of some like this:

[Figure]

Figure 2 (Apologies for the messy style). The 2 left hand columns of the figure shows CRU TS3.2 cloud cover (roughly used a not-so-great inverse proxy for SW) vs MAP, middle shows MAT vs

MAP, and the two right show number of wetdays vs MAP. Again, I'm not recommending CRU, but just using it as a readily available example. Green column 1 and 3 shows tree cover from VCF (Dimiceli et al. 2015), and red coloured columns 2 and 4 show burnt area from GFED4s (van der Werf et al. 2017). The regions (all tropics, Africa, Southern America, Asia and Australia) are the same used in the m/s.

Even from this example, it is clear that MAP is important but not the only control on either variable. Tree cover does increase with MAP as expected, but the extent of the increase is modulated by temperature, with an ideal MAT occuring around 25 degrees C, and with a rapid drop off at warmer temperatures. The relationship can be exaggerated further in some regions. Australia in particularly has tree cover extending into very dry areas when it is cool enough. Number of wetdays also seems important for tree control in Asia and Australia. Although some of this might be explained by fire feedbacks, that only goes to show that these variables are important for the fire axis also. As I'm using different data to the authors, I won't dwell on the details in the figure above - but it is an example of using on of the technique the authors have already developed to account for more climate controls and identify which biases are appropriate to consider when. A figure like this does not need to be included in the m/s, but it could serve as a starting point to help identify important climate biases. The authors could also think about using spearman's rank or the multivariate regression they used with JBACH to rule out significant effects of short wave.

*We prefer to keep the same averaging periods for all variables. If the goal was to only evaluate tree cover it would likely be a good idea. The goal here is however to evaluate the interactions. Tree cover influences the fire regime therefore having the same averaging period for these two variables seems plausible to us. Also the GFED data have more problems for the earlier years and are more reliable from 2001 on.*

This is a very good point and I'm happy for averaging period to be kept as is.

**References**

Ahlström, Anders, Josep G. Canadell, Guy Schurgers, Minchao Wu, Joseph A. Berry, Kaiyu Guan, and Robert B. Jackson. 2017. "Hydrologic Resilience and Amazon Productivity." *Nature Communications* 8 (1): 387.

Ahlström, Anders, Guy Schurgers, and Benjamin Smith. 2017. "The Large Influence of Climate Model Bias on Terrestrial Carbon Cycle Simulations." *Environmental Research Letters: ERL [Web Site]* 12 (1): 014004.

Ahlström, A., G. Schurgers, A. Arneth, and B. Smith. 2012. "Robustness and Uncertainty in Terrestrial Ecosystem Carbon Response to CMIP5 Climate Change Projections." *Environmental Research Letters: ERL [Web Site]* 7 (4): 044008.

Baudena, M., S. C. Dekker, P. M. van Bodegom, B. Cuesta, S. I. Higgins, V. Lehsten, C. H. Reick, et al. 2014. "Forests, Savannas and Grasslands: Bridging the Knowledge Gap between Ecology and Dynamic Global Vegetation Models." *Biogeosciences Discussions* 11 (6): 9471–9510.

Beck, Hylke E., Noemi Vergopolan, Ming Pan, Vincenzo Levizzani, Albert I. J. M. van Dijk, Graham P. Weedon, Luca Brocca, Florian Pappenberger, George J. Huffman, and Eric F. Wood. 2017. "Global-Scale Evaluation of 22 Precipitation Datasets Using Gauge Observations and Hydrological Modeling." *Hydrology and Earth System Sciences* 21 (12): 6201–17.

Dimiceli, C., M. Carroll, R. Sohlberg, D. H. Kim, M. Kelly, and J. R. G. Townshend. 2015. "MOD44B MODIS/Terra Vegetation Continuous Fields Yearly L3 Global 250m SIN Grid V006." *NASA EOSDIS Land Processes DAAC*.

Harris, I., P. D. Jones, T. J. Osborn, and D. H. Lister. 2013. "Updated High-Resolution Grids of Monthly Climatic Observations - the CRU TS3.10 Dataset." *International Journal of Climatology* 34 (3): 623–42.

Lasslop, Gitta, Kirsten Thonicke, and Silvia Kloster. 2014. "SPITFIRE within the MPI Earth System Model: Model Development and Evaluation." *Journal of Advances in Modeling Earth Systems* 6 (3): 740–55.

Li, J-L F., D. E. Waliser, G. Stephens, Seungwon Lee, T. L'Ecuyer, Seiji Kato, Norman Loeb, and Hsi-Yen Ma. 2013. "Characterizing and Understanding Radiation Budget Biases in CMIP3/CMIP5 GCMs, Contemporary GCM, and Reanalysis." *Journal of Geophysical Research, D: Atmospheres* 118 (15): 8166–84.

Nemani, Ramakrishna R., Charles D. Keeling, Hirofumi Hashimoto, William M. Jolly, Stephen C. Piper, Compton J. Tucker, Ranga B. Myneni, and Steven W. Running. 2003. "Climate-Driven Increases in Global Terrestrial Net Primary Production from 1982 to

1999." *Science* 300 (5625): 1560–63.

Sillmann, J., V. V. Kharin, X. Zhang, F. W. Zwiers, and D. Bronaugh. 2013. "Climate
Extremes Indices in the CMIP5 Multimodel Ensemble: Part 1. Model Evaluation in the
Present Climate." *Journal of Geophysical Research, D: Atmospheres* 118 (4): 1716–33.

Weedon, Graham P., Gianpaolo Balsamo, Nicolas Bellouin, Sandra Gomes, Martin J. Best,
and Pedro Viterbo. 2014. "The WFDEI Meteorological Forcing Data Set: WATCH
Forcing Data Methodology Applied to ERA-Interim Reanalysis Data." *Water Resources
Research* 50 (9): 7505–14.

Werf, Guido R. van der, James T. Randerson, Louis Giglio, Thijs T. van Leeuwen, Yang
Chen, Brendan M. Rogers, Mingquan Mu, et al. 2017. "Global Fire Emissions Estimates
during 1997–2016." *Earth System Science Data* 9 (2): 697–720.

---

## Referee Report (RR3)

**Review**
**of «Tropical climate-vegetation-fire relationships: multivariate evaluation of the land surface model JSBACH» by G. Lasslop, T. Moeller, D. D'Onofrio, S. Hantson, and S. Kloster**
**(Manuscript number bg-2018-48).**

Lasslop et al. presented a study on assessment of climate-vegetation-fire relationships in tropical regions based on both observational and model data. The choice of the topic for the research is undoubtedly worthwhile owing to importance of correct representations of climate (precipitation), vegetation, and fire in global climate models. Authors presented some interesting results that will be valuable for scientific community. The manuscript can be recommend to publish with minor revisions.

Here is the specific comments:

1) When authors discuss precipitation-vegetation-fire relationship, they miss a link between precipitation rate and lightning activity (thus, fire ignition) (see, e.g. Romps et al., doi: 10.1126/science.1259100). At least a short discussion on this point should be added to the paper.
2) Correlation coefficients in the Table 1 should be accompanied with statistical significance estimates (e.g. to show statistically significant coefficients with the bold font).
3) Figures 4, 6, and 7 are 'blind' and hard to read. Is it possible to increase dots and chose more contrast-to-white colors?
4) The word 'surprisingly' (Introduction, Section 3.3) seems to be unsuitable since there was made no particular assumptions on any expectations.
5) English should be improved, mostly in terms of punctuation.

---

## Author Response (AR2)

*We thank the reviewers for taking the time to provide additional comments. Our reponses are inserted below the reviewer comments in italics.*

**1 Review 1**

5  Page 14 line 10: Update to "NPP turns negative does drought effect the vegetation."
*1) Updated to include the next comment.*

Page 14 line 9-10: Add in how drought impacts the vegetation currently. What are the effects of drought that take effect after NPP turns negative for 5 years? Diminished productivity? Through percent reduction biomass? Total die-off?
10  *2) NPP is the productivity, lower productivity of course also leads to lower biomass, the effect of five years negative NPP is that PFTs stop establishing. We changed the sentence, (also addresses previous comment) to:*
*...only if a 5 year average of NPP turns negative, PFTs stop to establish.*

Page 14 line 11-12: "The excessive tree cover could be partly improved by improving the non-vegetated fraction which
15  decreases too fast with increasing precipitation." The non-vegetated fraction decreases too quickly, implying that the vegetation increases and controlling this vegetation response is important to addressing the excessive tree cover. What could be a mechanism of improvement be for the drought response?
*3) From this analysis it is not possible to draw conclusions about the mechanisms. The non-vegetated fraction depends on vegetation productivity, therefore improvements in productivity might help, also the hydrology is an important process for the*
20  *drought response, we therefore mention the relation to these two processes as examples now:*
*This non-vegetated fraction depends on the productivity of vegetation. Further investigation of effects of the soil moisture memory not only on climate (Hagemann and Stacke, 2015) but also on the vegetation might also lead to useful insights.*

Page 14 line 25: I think you mean "lower sensitivity to fire due to higher bark thickness and taller crown leading to lower
25  probability of crown scorch." Update.
*4) We removed the part on the crown, as the paragraph mostly focusses on the bark thickness.*

Page 14 -15, pg14 lines 21 to pg 15 line 14. There are duplicate sentences and redundancies in this paragraph. Improve the organization of the paragraph and remove redundancies. Specifically, the sentence about JSBACH-SPITFIRE and bark thick-
30  ness is duplicated.
*5) We removed the duplicate sentence and restructured and shortened the paragraph.*

Page 15 line 11: SPITFIRE quantifies fire intensity. Are the fires more frequent and of lower intensity once the tree cover has decreased?
35  *6) Fire frequency increases with decreasing tree cover as seen here in Figure 4,6 and 7 and in Lasslop et al. 2015. Fire line intensity however does not decrease with increasing burned area (Figure 1. We add the figure to the supplement.*

Page 15 line 15-21: Complete this paragraph with a concluding statement of how including saplings and adult long lived trees may impact the balance of tree cover to bare fraction or fire behavior in SPITFIRE.
40  *7) We included:*
*Including a sapling state could therefore increase tree cover in frequently burned areas, while decreasing tree cover (as described above) in areas that are too dry to provide fuel for frequent burning.*

Page 16 line 7: Update to "...climate models have problems representing extremes..."
45  *8) We updated the text as suggested.*

[Figure]

**Figure 1.** Relationship between annual burned area and fire line intensity. The expected decrease in fire line intensity for frequently burning areas is not found in the simulation results and might indicate that the feedback between fire occurrence, fuel load and fire intensity is too weak.

    Page 16 line 15: " fire is quite sensitive to the length of dry seasons" add references supporting this statement.
*9) change to number of dry days reference to Bistinas, also reference to correlation between dry days and mean precipitation in supplement.*

5      Page 17 line 1-3: The evidence that increases in managed land leads to decreased burned area is supported in the recent literature as referenced in the manuscript. Remove the sentence about roads as a fire break or provide a citation recognizing this link within the tropics. Roads are often a source of ignitions that also impact the spatial variability of burned area (Loboda and Csiszar 2007 RSE; Syphard et al 2007 Eco. Appl.; Syphard et al 2008 Int J Wild Fire; Narayanaraj and Wimberly 2012 Appl. Geog; Faivre et al 2014 Int J Wild Fire).

10   Additionally, there is evidence that forest fragmentation due to land cover change by humans leads to increases in fires. This type of fragmentation is shown to alter micro-climate conditions within forest canopies and lead to increased understory ignitions and fires. (Morton et al 2013 Philos Trans R Soc; Brando et al 2014 PNAS; Soares-Filho B et al. 2012 Landscape Ecol.) This connection is not well represented in models, but, as detailed in these references, representation of forest fragmentation specifically is an important component of capturing the relationship between humans, land use and fire. Further, as demon-

15   strated by and detailed within these references there is an expanding understanding of the relationships between humans, land use and fire. Acknowledge that this research is progressing, alongside the need to improve this representation in models.
*10) We remove the sentence about roads. The studies mentioned here are on a much smaller scale than our simulations, global large scale analysis so far only support the decrease of burned area due to humans. We mention that work on local scale helps to increase the understanding, however a generalization to the large scale is still needed to be able to represent it in global*

20   *models. As this is not the working scale of our model we do not include all 8 additional references, but only the most recent ones:*
*The mechanism behind the reduction in burned area due to croplands is however likely a fragmentation of the landscape, which is not explicitly accounted for in the model. On local scale understanding on these relationships is increasing, for instance the relation between fire and roads (Faivre et al., 2014; Narayanaraj and Wimberly, 2012) or between fire and land management*

25   *(Morton et al., 2013; Brando et al., 2014). However, a generalization to an approach that would be suitable for global models*

*is still missing.*

**2 Review 2**

*We only include the points where the reviewer disagrees to shorten the text and time needed to read it.*

The remaining main concern of the reviewer is summarized in the beginning of the review:
However, I remain concerned about the response to the "choice of JSBACH driving data", although after clarification on the aims of the analysis, I am no longer sure if the this is an issue with using ESM model output itself or the way this is used to infer areas for improvement of JSBACH.

*11) The main concern of the reviewer with respect to the climate biases is the seasonality of the rainfall. We therefore performed an additional analysis comparing the number of dry days and the rainfall seasonality of the forcing data used here to the CRUNCEP observational model forcing dataset (used in the FireMIP simulations, a setup that was suggested by the reviewer). We define rainfall seasonality as the number of days needed to reach 80% of the annual precipitation, and dry days as days with less rainfall than 3 mm. The CRUNCEP dataset is a reanalysis dataset commonly used in offline model comparisons (Rabin et al., 2017). This analysis (, now a figure in the supplement) shows that the number of dry days in dry regions is well comparable between model and CRUNCEP, for moister regions the number of dry days is even higher in the forcing dataset (MPI-ESM output) used here. We therefore confirm that our conclusions are unlikely affected by biases of rainfall seasonality. The reviewer now also mentions existing biases in shortwave radiation. Of courser biases always exist, here, however, it is important whether the climate biases could have such a strong effect as the reviewer claims. Shortwave radiation does not affect the tree cover in JSBACH, we quickly tested it by applying a multivariate regression, precipitation is highly significant, radiation is not significant if only these two variables are used in a multivariate linear regression. As so far there is no discussion on shortwave radiation and how it influences the model in the paper, we did not include this in the manuscript as it would require several paragraphs to be added.*

Choice of JSBACH driving data

The authors have clarified the three aims:

1. Develop a simple multivariate technique to explore the difference between modelled and observed vegetation, fire and climate.

2. Use these differences to evaluate the simulation of, and coupling between, tree cover and burnt area in JSBACH

3. To do this within the MPI-ESM framework, achieved by driving JSBACH offline but with MPI-ESM model output.

There is nothing wrong with the aims, and I like that the authors attempt to at keep the methodology relatively simple. Driving the JSBACH as configured for use in MPI-ESM, offline with ESM output also makes sense, and in their response and suggested changes to the manuscript, the authors have justified the choice of driving data. The authors also discuss the weaknesses associated with this method in the revision of section 4.1, which is also a welcome addition to the paper. However, to critique solely the land surface component in an ESM setup such as this, as if it were independent of other potential climate model biases seems to contradicts the 3rd aim above and introduces the methodological inconsistency which I don't feel have been adequately addressed. To phrase in terms of the multivariate approach, the authors have diagnosed the vegetation cover and fire axis, but not the climate axis.

*12) We also already mentioned climate biases in the revised version of the manuscript in more detail. The expected drizzle, which seems to be the main concern of the reviewer, however, is not present in our climate forcing as our new analysis shows. See also reply 11. Our proposed method clearly goes beyond the normal variable by variable comparison. Including all variables that might be important in the coupled system of fire, vegetation and climate would be optimal in a certain sense but would then suffer from the complexity of the necessary approach and difficulties in interpretation. As stated in the manuscript we use precipitation as a proxy for climate and precipitation is included as one of the axis. The same critisim, that there could be biases not in the mean but in another characteristic of precipitation, could apply to fire and vegetation cover. We simply use annual burned area as a proxy for the fire regime, but fire intensity and seasonality and extremes can be important characteristics too. For tree and grass cover we also summarized two PFTs into one variable. This is the compromise we did to allow a*

[Figure]

**Figure 2.** Relationship between annual precipitation and precipitation seasonality and number of dry days for the ECHAM simulation used as meteorological forcing for the JSBACH simulations used here and the CRUNCEP dataset.

*simple interpretable approch.*

The authors state in their response that "regarding the conclusions we draw from our comparison we don't see a strong point that they would be strongly affected." Here are just some examples from the (revised) paper where climate biases could potentially affect either the results, discussion and/or conclusion:

1. Surprisingly the observations show a higher Spearman correlation between tree cover and precipitation than the models (Table 1). The lower correlation of the modelled relationship most likely originates from the lower precipitation regions (<500 mm year-1 where the maximum tree cover is very low in the observations and both models strongly overestimate the maximum tree cover (Figure 4).

The correlation between MAP and other climate variables that influence tree cover could also break down in the MPI-ESM driving JSBACH. As already noted, length of dry seasons are likely to be shorter in seasonal climates. Most GCMS models (although I don't know if MPI-ESM is amongst them) also suffer from biases in downward SW (Li et al. 2013) which could influence tree cover, particularly at the higher tree cover range, where figure 5 also indicates mismatches beetween model and observation in some continents, particularly Asia.

*13) Radiation could have a considerable influence on the productivity of PFTs, but is very unlikely to influence tree cover*

*in JSBACH for the tropics based on the way the model is build. We tested this also quickly with a multivariate regression TC=a1\*P+a2\*R for the modelled variables where the influence of radiation is not significant. It is therefore unlikely that biases in radiation would show up in tree cover. We now show that the number of dry days is not less in the ECHAM forcing. See also reply 11 and 12.*

    2. JSBACH overestimates tree cover for low precipitation on all tropical continents. The drizzle problem already discussed seems like an obvious candidate to affect vegetation cover at low precipitations, either through decreasing the length of dry periods or due to associated changes in cloud cover changing evaporative demand and hence available moisture. Despite not ruling out additional climate problems, the authors use this simulated mismatch at low tree covers to justify planned changes

10    to tree to vegetation dynamics: only if a 5 year average of NPP turns negative, drought effects on the dynamic vegetation take effect. Other models require a minimum of 100 mm year-1 precipitation for sapling establishment (Sitch et al., 2003). The too high excessive tree cover could be partly improved by improving the non-vegetated fraction which decreases too fast with increasing precipitation and Tree-grass competition for water could for example be improved in the model by introducing the a sapling stage of trees, which are competitively inferior to grasses (D'Onofrio et al., 2015). Including this mechanism could

15    improve the balance between tree and grass cover, but it could also reduce the establishment rate of trees and therefore the tree cover in the dry regions with excessive tree cover. Including a PFT-specific rooting depth of vegetation would be an important extension of the model to improve the competition for water between grasses saplings and adult trees. These three fundamental changes to the dynamics of JSBACH are suggested without establishing that the problem is with JSBACH itself. While it is often necessary reparameterize components of ESMs to compensate for biases in other model components, this should always

20    been done in the knowledge that it is to compensate for other these other biases, and the suggested changes to JSBACH above go beyond a standard re-parameterization.

*14) Our analysis shows that there is no drizzle problem see reply 11. The suggestions made here are also based on the too strong dominance of trees in dry regions, grasses can only exist if fire is present, effects of climate on productivity would not make a difference about this in the model. Even if there were other climate biases or other problems in the model, the processes*

25 *discussed here are known to be crucial for the vegetation composition in dry areas, therefore suggesting that including them could help the model be better is in our opinion reasonable. We add in the manuscript:*

*The suggested processes are known to be important for the vegetation distribution and it seems plausible that they can help to improve the vegetation distribution.*

30    3.For Australia underestimation of burned area for both fire models is strong (Figure 4). In a previous evaluation where the model was forced with observed climate and vegetation cover was prescribed (in contrast to the dynamic vegetation cover and climate modelled by the MPI-ESM) JSBACH-SPITFIRE showed better results for Australia (Hantson et al., 2015). An improved response of vegetation cover dynamics to precipitation will therefore likely improve the patterns of burned area. The better simulation of fire in Hantson et al. 2015 could also be due to better representation of rainfall timing and distribution,

35    temperatures or any number of climate factors from being driven by observed climate. Also, better representation of vegetation cover would hopefully have been achieved in Hantson et al. 2015 with observed rather than simulated climate. Again, parameterization of either JSBACH or SPITFIRE to account for additional climate biases may be necessary in an Earth System model, but here the author imply the the problem is with JSBACH itself.

*15) A reduction in tree cover would lead to an increase in burned area, therefore what we write is correct. Or vice versa the*

40 *high burned fraction observed in Australia cannot be achieved with SPITFIRE if such a high tree cover is present. The JSBACH model is parameterized for the coupled setting not for the observational dataset. The model shows between 10 and 30% tree cover for any precipitation below 500mm per year, while the observations a maximum of 10%. The discrepancy is rather large and it seems unlikely that a different distribution of rainfall can explain the difference. Also the reviewer does not give any references that climate model biases can have such a big effect. Of course any of the climate parameters used can be wrong,*

45 *but the same would be true for any observational dataset used as model forcing.*

    4.This indicates that not an improvement of the fire model but improved modelling of drought effects on the vegetation dynamics will improve the response of vegetation to climate in dry regions. Again, another likely explanation is MPI-ESM rainfall distribution or the impact of other climate factors on available moisture etc.

*16) In regions where fire is absent trees always win the competition in JSBACH, it is therefore impossible that other climate factors can solve this, the only reasonable reason is the absence of drought effects on vegetation cover in the model.*

5.Intercontinental variation in the relationship between precipitation and maximum tree cover is much smaller for the models compared to the observations. Known variations in vegetation are not sufficiently understood to be represented in models. However our finding that models do show differences in the fire-vegetation-climate relationships between continents shows that further exploration why models show differences can be helpful to better understand causes for intercontinental differences. If this is meant purely for land surface modelling, then there is little in the results of this paper to justify this statement. That there is a modelled difference in fire-vegetation-climate relationships between continents would be more valid if the authors made it clear that this statement is about the ESM setup as a whole.

*17) This comment is unclear, the variations that are mentioned are observed and the model also shows some variations. We do not see how the ESM setup as a whole comes in here.*

6.Overall the multivariate model evaluation highlights the potential for more targeted model improvements with respect to the interactions between climate vegetation and fire, which are crucial for our understanding of future vegetation projections. Again, this is fine as a statement about the ESM setup as a whole, but not focusing solely on the land surface component. Of course, there are more suggest model improvements in the manuscript where inherent climate biases from MPI-ESM have (to my mind at least) no obvious impact. However, even in these cases, the authors should be careful at presenting potential new model processes without first checking for the influence in other climate biases. The apparently stronger correlation between fire and tree cover compared to observations, for example, is used to suggest inclusion of resprouting and adaptive bark thickness or fuel feedbacks that might influence fire intensity and hence tree mortality. Again, there are no end of climate biases that could affect intensity which would not be picked up by a straight MAP-tree cover-bunt area comparison. And again, these changes go far beyond standard reparameterization of a land surface model in an ESM. To be fair to the authors, they have included the statement "exact parameterization and needs to be tested with stepwise model development and factorial simulations" which does help mitigate some concern with model changes such as this.

*18) Climate biases can clearly influence the burned area, and its spatial patterns, but I do not see a way that climate biases will turn around the impact of fire on tree cover that much in SPITFIRE, except for the fire-fuel feedback mentioned by the reviewer here. This feedback is already included in the model and different climate forcing leading to different fuel loads could maybe strengthen the feedback. However, in that case it would make sense to reparameterize the model to strengthen the feedback in the Earth system model setting.*

The authors suggest in their response that only way to address this contradiction is the do detailed assessment of the atmospheric component of the model, or perform complex experiments or analysis using additional model driving data. This is almost certainly not the case, and it would be a shame if further revisions did make the analysis more complicated. However, some additional, simple analysis might resolve the issue. Here are some examples based on the author responses:

We use the standard JSBACH setup, which is the combination of JSBACH with MPI-ESM meteorology. As the fire is sensitive to a number of variables, evaluation of the model in a different setup wouldn't help to guide model development for a model that is almost only run in the coupled setup.

A run with observed climate obviously wouldn't be used as a basis for further model development if your aiming to improve JSBACH when driven with ESM meteorology. But it would help the authors determine if the deficiencies already identified are due to simulated climate biases or due to the vegetation component, and would place their discussion on much firmer ground. This is part of the justification for offline land surface model runs required for MIPs associated with CMIP6, e.g (van den Hurk et al. 2016; Lawrence et al. 2016).

*19) Using observed climate is obviously most useful when looking at the spatial patterns, which we avoid with our method. It is certainly an important approach, our aim here was to evaluate the model for the Earth system model setup.*

our motivation here is to evaluate the land surface model, a detailed evaluation of climate biases in the ECHAM model is therefore out of scope

There is no need to do a detailed evaluation of climate biases in ECHAM (which would indeed be out of the scope of this

paper). However, the authors should ascertain if problems in land surface simulation are caused by either problems in the simulation of the land surface or problems with the information it receives from the atmospheric component - a rather basic first order assessment of any land surface model within an earth system framework. I was able to give a few pointer to potential climate biases from my limited knowledge of climate intermodel comparison literature in my last review. The authors should

5   be able to identify other MPI-specific climate biases that they could at least discuss if not to test. As stated in the last review, there are two of instances where climate model deficiencies are discussed (i.e, when explaining discrepancies in simulated spatial patterns and when discussing calculation of lightning ignitions). At the very least, these types of discussions should be included when critiquing the rest of JSBACH.

*20) We adressed the problem of the drizzle now, showing that the seasonality and number of dry days is comparable in ECHAM*
10  *and a observational model forcing dataset. We already discuss model deficiencies in the revised version in appropriate sec-*
*tions, more specifically in the section where discuss potential model improvements, repeating these in more parts would lead*
*to redundancies.*

Understanding potential influences of certain climate biases (such as extremes) on the simulation would require specific
15  factorial experiments
and
While certainly more parameters influence tree cover distribution an increasing number of variables included to explain patterns would require a totally different appraoch
Not necessarily. A first step could be to simply show if other climate information (no. dry days, downward SW etc) are causing
20  some of the relationships you see using the exact same approach used for MAP. It may well be that this shows that using MAP alone does do a sufficient enough job as a proxy for climate space, which will then support the rest of the papers discussion. If not, then any additional climate variable that explains some discrepancy could be included in the same way that grass and tree cover are interchanged at various stages in the manuscript.

*21) Precipitation is the main driver of vegetation cover in the tropics. Removing the main driver from this analysis and ex-*
25  *changing it with other potential climatic drivers that are correlation with Precipitation would likely lead to correlations be-*
*tween vegetation and the climatic driver mainly because of the correlation between the two drivers. The relationship would*
*then still be caused by precipitation. We do not see a way for a useful interpretation of such relationships without removing the*
*effect of precipitation, which would require a more complex approach. Exchanging tree and grass cover is different as both are*
*mainly driven by precipitation and fire.*

Mean annual precipitation explains a large part of the tree cover variability and therefore is a useful proxy for climate
Obviously not enough in JSBACH when driven by MPI climate data - the range of TC at a given MAP is one of the features JSBACH as driven by MPI data does not replicate, and there is no other result to help indicate how much of this discrepancy is due to simulation of vegetation cover or other climate biases. While I do not expect the authors to address all these points
35  in the manuscript (that would be a very long paper!), I hope that I have demonstrated they are certainly not without options. Picking up on one or two of these point, or anything else which can either show MAP really is enough by itself to account for all other climate biases or that can truly attribute problems with model performance to either JSBACH or MPI-ESM climate, will be sufficient.

*22) The reviewer agrees that JSBACH is not enough driven by mean annual precipitation, our previous comment cited here ref-*
40  *ered to observed relationships (although this may have been not exactly clear). He also agrees that adressing all his comments*
*is outside of the scope of this paper or would make the paper too long. We hope that the additional analysis with the number*
*of dry days, which supports that our main discrepancy between model and observations (the overestimation of tree cover for*
*dry regions is a problem of JSBACH not the forcing data, adresses most of the reviewers concerns.*

45    As the authors are only able to use MPI-ESM model output till 2005 to drive JSBACH, they have to make a rather awkward choice about comparison time periods, as identified in the last review. An ideal solution to this would be to run MPI-ESM beyond 2005, something that could be happening as part of CMIP6 simulations? However, I realise that this is probably not possible, and the MPI-ESM may well be configured differently for CMIP6 simulations. I would like to hear to authors thoughts changing the comparison periods though. The authors state that "Using only the overlapping period (2001-2005) would decrease the robustness of the mean fire regime and climate characterization". This is certainly true for fire regime. However, tree cover is normally more stable, and as trees take a few years to establish, the cover found during 2001-2005 would of also be a consequence of burnt area and climate before this period. Perhaps a better choice is to split comparison periods based on variable rather then on model/observation. i.e, when performing analysis, take modelled and observed burnt area and climate

5 from 1996-2005 (climate data, MPI and GFED overlap) and tree cover from 2001-2005. While this is still a rather pragmatic solution to the mismatch in modelled and observed time periods, it might make more sense then the pragmatic solution outlined in the manuscript?

*23) We prefer to keep the same averaging periods for all variables. If the goal was to only evaluate tree cover it would likely be a good idea. The goal here is however to evaluate the interactions. Tree cover influences the fire regime therefore having the*

10 *same averaging period for these two variables seems plausible to us. Also the GFED data have more problems for the earlier years and are more reliable from 2001 on.*

I am happy with all changes except for a couple of small details:

39 and 45) Kelley and Harrison 2014 should probably be changed to Kelley et al. 2014:

15 Kelley, D. I., Sandy P. Harrison, and I. C. Prentice. "Improved simulation of fire–vegetation interactions in the Land surface Processes and eXchanges dynamic global vegetation model (LPX-Mv1)." Geoscientific Model Development 7.5 (2014): 2411-2433.

Kelley and Harrison 2014 looks at future changes in fire, whereas Kelley et al. 2014 is the paper describing model development and benchmarking.

20 *24) We thank the reviewer for finding this mistake, the reference is corrected.*

42) Was the "minor shifts between woody PFTs in a few cells" quantified? Quantifying equilibrium during spin-up should really be a requirement for any modelling study, and if authors did quantify equilibrium in any way, then it would help the cause to state how this was assessed. However, I do understand that quantifying equilibrium is unfortunately not standard practice,

25 and finding a way to do so is well outside the scope of this paper. So if it was not quantified, then leave this sentence as is.

*25) The minor shifts were not used to quantify equilibrium. We therefore leave the sentence as is.*

**References**

[revised manuscript text omitted]

---

## Author Response (AR3)

Dear editor, 1) we here reply to the third review of the reviewer. We updated the figure evaluating the rainfall seasonality and number of dry days with the TMPA dataset instead of CRUNCEP and find a better agreement between MPI-ESM and the observational dataset than previously. Although we don't agree that it is necessary to look at all other climate biases as with the multivariate analysis we already account for spatial biases in precipitation (which is the main driver of tree cover) we now show that bias in mean temperature and bias in mean shortwave radiation does not explain any of the tree cover bias (close to zero, non-significant correlation). We adjust the paragraph on the discussion of climate biases and moved it to the discussion section on "Limitations in the comparability between observations and modeled variables".

We are surprised about the reviewer's grading. Following the reviewer's advice of his second review apparently led to a degradation of the scientific significance. The reviewer's assessment of scientific quality dropped following the reviewer's advice of the first review. With this third review it remains largely unclear what exactly the reviewer wants us to change and especially how to make an end to this process.

Below we reply to the reviewer as positive as possible. Our replies are in italic. Our replies cited by the reviewer are set in bold.

**1 Review**

The authors have included new figure to determine if there is a bias in rainfall distribution simulated by ECHAM. This has helped rebalance the analysis across variables in their multivariate analysis. I do have some specific issues with the new figure, which I also feel could be used more in discussing weaknesses in the model performance. I also pick up on some of the author responses that could have some bearing for revision in the rest of the paper, and provide some examples of quick analysis to assess or rule out two remaining climate biases described in the last review.

Figure B1

1. The figure uses CRU-NCEP precip observations, whereas TMPA is used for observed precip elsewhere in the paper. The same climate observations should be used for both, especially as the new figure is used to diagnose differences in climate relationships between the two different climate axes in figures 2, and 4-7. Apologies if I implied in an earlier review that it would be okay to use different precip data for different parts of the analysis - I used fireMIP as an example of an offline JSBACH-SPITFIRE simulation, and thought that it would self evident that if used, observations should still be consistent across different parts of the analysis. While the authors may argue that choice of dataset might make little difference to the relationship, the disagreement in precipitation between observed datasets in notorious (Beck et al. 2017; Weedon et al. 2014). A quick plot of MAP vs no. dry days I conducted with CRU TS3.2 (Harris et al. 2013) (data I chose for no other reason that I already had it downloaded, not because I'm recommending it for use in the m/s) compared to CRU-NCEP used in figure B1 shows what I mean:Figure 1: CRUTS3.2 MAP vs no. dry days for tropics and Australia, based on coordinates provided in section 2. The relationship between MAP and no. drys days for CRU by itself is clearly different, and would actually agree more with ECHAM annual precip. TMPA may also show a significantly difference relationship as well.

*2) We downloaded the daily version of the TMPA dataset (the one used in the manuscript is monthly) and redid the figures showing an even better agreement with the model and only small underestimation of seasonality or number of dry days, mostly for high rainfall regions which are not important to our conclusions. See also reply 1.*

2. The authors already produces an excellent style of figure to diagnose burnt area vs precip in figure 3 which could have been used here with the x-axis displaying precip and the y-axis cumulative days at a given precip level. This would provide more information on rainfall distribution biases. However, there is nothing particularly wrong with the simple scatter plot used, so I'll leave this as a suggestion rather than a requirement.

*3) We prefer to use the plot as it is.*

3. If the author's choice to stick with the scatter plots, then please add a trend line.

*4) See reply 3. We added a trend line and even the slope of the line.*

4. Remember that, for SPITFIRE, impact of dry days in cumulative, so cumulative dry days might also be worth considering, especially as this is another area MPI has been shown to sometimes struggle with (Sillmann et al. 2013) . If the authors are able to use the variables already in figure B1 effectively though, then again this won't berequired.

*5) As the burned area pattern shows a good relation with precipitation we don't see a need to include a third measure of rainfall seasonality.*

5. The authors need to use this figure to help diagnose climate relationships in more detail. In their response, the authors state that " This analysis ... shows that the number of dry days in dry regions is well comparable between model and CRUNCEP, for moister regions the number of dry days is even higher in the forcing dataset (MPI-ESM output) used here. We therefore confirm that our conclusions are unlikely affected by biases of rainfall seasonality." If this relationship holds once the figure is redrawn with TMPA, then the "anti-drizzle" bias in MPI is surprising. However, it is still a climate bias that will affect simulated fire and possible vegetation, and should be discussed as such in the main text. If it turns out that MPI-ESM agrees with TMPA dry days, than the text will be fine as it is.

*6) See reply 2 and 5. MPI-ESM agrees better with TMPA, especially the agreement in the dry regions, which we discuss the most, is good. We modified the text according to the new comparison.*

Author responses
We cite our old response set in bold, the reviewers comments on it normal and our replies in italics.
**main concern of the reviewer with respect to the climate biases is the seasonality of the rainfall.**
I use seasonality as an example, and it was not the only or main concern.
*7) OK.*

**Of courser biases always exist, here, however, it is important whether the climate biases could have such a strong effect as the reviewer claims.**
This is correct. I have no idea how strong an affect the climate biases have. As the authors are presenting a new way of evaluating land surface in ESMs, they need to demonstrate that the impact of other climate biases is either negligible or can be accounted for.

*8) See reply 1. The multivariate evalutation takes into account spatial biases of precipitation. We discuss the presence and impact of climate biases, include analysis on climate biases of precipitation seasonality, and now temperature and shortwave radiation. Our results show no influence of climate biases on the tree cover.*

**Shortwave radiation does not affect the tree cover in JSBACH, we quickly tested it by applying a multivariate regression, precipitation is highly significant, radiation is not significant if only these two variables are used in a multivariate linear regression. As so far there is no discussion on shortwave radiation and how it influences the model in the paper, we did not include this in the manuscript as it would require several paragraphs to be added.And Radiation could have a considerable influence on the productivity of PFTs, but is very unlikely to influence tree cover in JSBACH for the tropics based on the way the model is build. We tested this also quickly with a multivariate regression TC=a1\*P+a2\*R for the modelled variables where the influence of radiation is not significant. It is therefore unlikely that biases in radiation would show up in tree cover. We now show that the number of dry days is not less in the ECHAM forcing. See also reply 11 and 12.**
Was this test with just JSBACH, or for observed tree cover/climate as well? Obviously if SW does not have a significant effect on JSBACHs simulated tree cover but does on observed, then this would be a useful missing climate-vegetation relationship that would need to exploring. If it was tested for both model and observation, then the authors point stands.

*9) We show that radiation has no influence of the modelled tree cover, therefore biases in radiation cannot have an influence on our results. We now additionally compute the correlation between the radiation bias and tree cover bias and find a zero, not significant correlation. See reply 1 and 8.*

**Our proposed method clearly goes beyond the normal variable by variable comparison. Including all variables that might be important in the coupled system of fire, vegetation and climate would be optimal in a certain sense but would then suffer from the complexity of the necessary approach and difficulties in interpretation. As stated in the manuscript**

**we use precipitation as a proxy for climate and precipiation is included as one of the axis. The same critisim, that there could be biases not in the mean but in another characteristic of precipitation, could apply to fire and vegetation cover. We simply use annual burned area as a proxy for the fire regime, but fire intensity and seasonality and extremes can be important characteristics too. For tree and grass cover we also summarized two PFTs into one variable**

5   Although the authors have only used burnt area for the fire axis, assessment and suggested improvements have borrowed a lot from previous model assessment and literature. In response to reviewer 1s comments, they also have started exploring fire intensity (figure C1). Obviously PFT fractions are always going to be grouped into just three (tree, grass, bare) fraction types for observational comparison, but each were assessed, which gave some grounding for suggested changes in vegetation dynamics, at least from the land surface bias side. Land use experiments also help explore this impact of changing anthropogenic

10  land cover in JSBACH - again part of the vegetation axis. There is also extensive discussion of changes in plant physiological traits and vegetation dynamics and vegetation-fire feedbacks. And this maybe the key to the problem. i.e, the number observed datasets + number ofvariables assessed + past model evaluation + literature + suggested model deficiencies and potential development that has gone into the fire and vegetation axis is extensive, but there is much less detail on the climate axis. And that any mismatches in the multivariate pattern compared to observations are almost always assumed to be because of vegetation

15  and fire biases and not climate. This can be properly balanced by proper discussion of figure B1, and/or reference to MPI climate assessment and climate biases.

   *10) We discuss that figure B1 does not show a concern that climate biases are important. We do not aim at discussing the impact of climate biases in the same way as the biases in the vegetation and fire parameters, as we aim to evaluate vegetation and fire not the climate. This disbalance is therefore intended, discussion on the biases in MPI-ESM forcing is included in the*

20  *manuscript.*

**A reduction in tree cover would lead to an increase in burned area, therefore what we write is correct. Or vice versa the high burned fraction observed in Australia cannot be achieved with SPITFIRE if such a high tree cover is present.**
   The argument that burnt area would increase with reduced tree cover is fine. That the ESM needs to reduce simulated tree cover

25  in Australia is also fine. The problem is the statement that " An  improved response of vegetation cover dynamics to precipitation   will therefore likely improve the patterns of burned area  " has not been demonstrated. I suspect improved vegetation response would be useful, but I also suspect that biases in MPI climate also share some of the blame. If a change in vegetation cover dynamics is induced with is used to improve fire by compensating for any climate bias, then this is not an improved response but a pragmatic tuning and should be identified as such. Figure 1c shows too much rainfall in Northern Australia, so

30  the authors could already use some of their original analysis to diagnose precip as one potential climate bias that would affect tree cover and burnt area. In terms of regional climate biases not taken into account by MAP, it might be that figure B1 isn't very helpful yet. In figure 1 in this review, for example, the slope of the fit line, spread of the data, and deviation from linear fit at low precips is different for Australia compared to the spread for the whole tropics.

   *11) We account for spatial biases in mean annual precipitation by evaluating tree cover and burned area for a given mean*

35  *annual precipitation.*
   *In Australia tree cover is too high, burned area is too low. We know that if tree cover decreases burned area increases in the model, therefore the burned area will increase. There is no further demonstration necessary. Our statement is correct and sufficiently supported by previous sensitivity analysis (Lasslop et al., 2016) and our knowledge of the model equations.*
   *We modify the text to improve the clarity and include that observed climate might contribute to the improved pattern:*

40  *An improved response of vegetation cover dynamics to precipitation will reduce the underestimation of burned area as in SPIT-FIRE tree cover and burned area are closely related (Lasslop et al., 2016). Part of the better performance in the previous study might also be due to the use of observed climate forcing.*

**Also the reviewer does not give any references that climate model biases can have such a big effect. Of course any**
45 **of the climate parameters used can be wrong, but the same would be true for any observational dataset used as model forcing.**
   Apologies for not providing references in the previous review. The authors may want look at and cite (Ahlström et al. 2017). Although exploring the carbon cycle rather than vegetation cover, they did show a significant impact of precip, temperature and SW biases on simulated vegetation in CMIP5 models. Focussing on the Amazon, (Ahlström et al. 2017) showed MAP,

SW and temp climate biases explain most of the simulated GPP, above ground biomass and tree cover. (Ahlström et al. 2012) also showed similar results for disagreement in projected changes in different climate variables into the future. These are just the ones I can think of off the top of my head, there is probably many more. As GCMs have been around for a lot longer, there is of course extensive literature on climate biases that could potentially lead to problems with vegetation dynamics once enabled. (Sillmann et al. 2013) might be a good starting point. The authors could use (Li et al. 2013) to support their view that only MAP needs to be considered for tropical vegetation distribution, as they use observational constraints to show MAP is the main driver of disagreement in vegetation productivity across models in a region of similar extent to southern America used in this study. However, it should be noted that other climate biases appear to become more important at high MAPs, where vegetation productivity is predominantly limited by available radiation (Nemani et al. 2003). I don't know enough about vegetation dynamics (in model or real world) to know if this tipping point between MAP and SW limited production occurs when tree cover is already saturated. If it does, then maybe (Li et al. 2013) would suggest that other tree cover controls don't need to be considered, at least for this region. I'm not so sure about the impact of climate biases on fire, as this is a little outside my area of expertise. However, I get the impression that, even with wind speed limitation, SPITFIRE is sensitive to variations in windspeed, especially at lower speeds (Lasslop et al. 2014), which again, GCMs struggle to adequately simulate.

*12) We already showed that radiation has no influence on the tree cover, this is already clear from the model equations. We now include the correlation between temperature and shortwave radiation bias with tree cover bias, which is close to zero and insignificant. See reply 1,8,9.*

**In regions where fire is absent trees always win the competition in JSBACH, it is therefore impossible that other climate factors can solve this, the only reasonable reason is the absence of drought effects on vegetation cover in the model.** Again, the authors need to back these statements up by showing in some way that other climate biases are not the issue here. As they are unable to run JSBACH with climateobservations, perhaps offline runs could be referenced in other papers. For example, JSBACH seems to simulate too much tree cover at low MAPs in the offline study by (Baudena et al. 2014). If this was an appropriate test with no fundamental developmental changes compared to the JSBACH configuration used in the m/s, then the authors could cite this study to back up their suggestion of improved vegetation dynamics at MAP. The authors should have a much better idea of published JSBACH and MPI experiments and evaluation, so might also be able to think of better examples.

*13) Our statement is based on the knowledge of the model equations, it is impossible that climate biases affect the dominance of trees in regions without fire. We include a sentence in the model description to emphasize this more:*
*"In gridcells without disturbance and positive NPP trees prevail."*
*The cited study of Baudena et al. uses results from a coupled simulation. No comparable study with offline JSBACH and observations exists. We account for spatial biases by evaluating tree cover for a given mean precipitation. With our additional analysis we do not find any indications that the other climate biases affect our results. See reply 1,8,9, and 12*

**This comment is unclear, the variations that are mentioned are observed and the model also shows some variations. We do not see how the ESM setup as a whole comes in here.**
I was just reinforcing that fact that the climate axis should be considered as much as the vegetation and fire axis. I meant "ESM setup" as a land surface model driven by ESM output that is emulating a full ESM, obviously without the land-atmosphere feedback (I'm not sure that makes it any clearer...?).

*14) We do not intend to adress the climate axis in a similar detail as the vegetation parameters. The latter are the focus of this study.*

**Climate biases can clearly influence the burned area, and its spatial patterns, but I do not see a way that climate biases will turn around the impact of fire on tree cover that much in SPITFIRE, except for the fire-fuel feedback mentioned by the reviewer here. This feedback is already included in the model and different climate forcing leading to different fuel loads could maybe strengthen the feedback. However, in that case it would make sense to reparameterize the model to strengthen the feedback in the Earth system model setting**
The point is more to show that the cause of low tree cover is fire feedback in the first place, and not other climate biases (though

the author are right that maybe the impact of climate biases on fire-feedbacks should also be a concern...?) If the authors can show climate biases beyond MAP isn't to blame, then the suggested changes fire-feedback are fine.

*15) That fire is the reason can easily be seen from the figure 4 where fire and low tree cover are clearly related, most notably*
*for Africa. Moreover, with the old fire model, that does not include a feedback between fire and vegetation the tree cover is*
5 *higher, it is therefore caused by the different fire model. We could not identify any support of the reviewers idea that climate*
*biases impact our results. see reply 1, 8,9,12 and 13*

**Precipitation is the main driver of vegetation cover in the tropics. Removing the main driver from this analysis and exchanging it with other potential climatic drivers that are correlation with Precipitation would likely lead to cor-**
10 **relations between vegetation and the climatic driver mainly because of the correlation between the two drivers. The relationship would then still be caused by precipitation. We do not see a way for a useful interpretation of such relationships without removing the effect of precipitation, which would require a more complex approach. Exchanging tree and grass cover is different as both are mainly driven by precipitation and fire.**

I was more thinking of some like this: Figure 2 (Apologies for the messy style). The 2 left hand columns of the figure shows
15 CRU TS3.2 cloud cover (roughly used a not-so-great inverse proxy for SW) vs MAP, middle shows MAT vsMAP, and the two right show number of wetdays vs MAP. Again, I'm not recommending CRU, but just using it as a readily available example. Green column 1 and 3 shows tree cover from VCF (Dimiceli et al. 2015) , and red coloured columns 2 and 4 show burnt area from GFED4s (van der Werf et al. 2017) . The regions (all tropics, Africa, Southern America, Asia and Australia) are the same used in the m/s. Even from this example, it is clear that MAP is important but not the only control on either variable. Tree cover
20 does increase with MAP as expected, but the extent of the increase is modulated by temperature, with an ideal MAT occuring around 25 degrees C, and with a rapid drop off at warmer temperatures. The relationship can be exaggerated further in some regions. Australia in particularly has tree cover extending into very dry areas when it is cool enough. Number of wetdays also seems important for tree control in Asia and Australia. Although some of this might be explained by fire feedbacks, that only goes to show that these variables are important for the fire axis also. As I'm using different data to the authors, I won't dwell on
25 the details in the figure above - but it is an example of using on of the technique the authors have already developed to account for more climate controls and identify which biases are appropriate to consider when. A figure like this does not need to be included in the m/s, but it could serve as a starting point to help identify important climate biases. The authors could also think about using spearman's rank or the multivariate regression they used with JBACH to rule out significant effects of short wave.

*16) We agree that using regressions is a very interesting way of analysing model results. There is no evidence of an influence*
30 *of cloud cover on tree cover literature to our knowledge. Knowing that cloud cover/short wave does not influence tree cover*
*in the model we do not consider including a discussion useful. The reviewer mentions that such a figure does not need to be*
*included, but we should use regressions or correlations. We computed the correlation between radiation bias and tree cover*
*bias; as expected the correlation is not significant. We did the same for temperature and also found no indication that a*
*temperature bias could help to explain tree cover differences. We indicate the lack of correlation/association between climate*
35 *biases and tree cover biases in the manuscript now. See reply 1,8,9,12,13 and 15.*

**References**

[revised manuscript text omitted]

---

## Author Response (AR4)

Dear editor,

We thank you and the anonymous reviewer for the helpful comments. Below we reply to the specific comments of the review and detail our changes in the document.

**1 Replies to the specific comments**

1) When authors discuss precipitation-vegetation-fire relationship, they miss a link between precipitation rate and lightning activity (thus, fire ignition) (see, e.g. Romps et al., doi: 10.1126/science.1259100). At least a short discussion on this point should be added to the paper.

*The manuscript already included a discussion on the connection with lightning on p.16 l. 7-15. We add the recommended reference to this paragraph:*

*Lightning strikes are strongly related to precipitation (Romps et al., 2014).*

2) Correlation coefficients in the Table 1 should be accompanied with statistical significance estimates (e.g. to show statistically significant coefficients with the bold font).

*We indicate the significance with bold font now.*

3) Figures 4, 6, and 7 are 'blind' and hard to read. Is it possible to increase dots and chose more contrast-to-white colors?

*The student who prepared these figures left the institute a while ago. Although we agree that the contrast could be improved, we do not see a major problem in the graphical appearance of the graph and therefore keep it as it is.*

4) The word 'surprisingly' (Introduction, Section 3.3) seems to be unsuitable since there was made no particular assumptions on any expectations.

*We removed the word 'surprisingly'.*

5) English should be improved, mostly in terms of punctuation.

*We corrected punctuation and languange in some places, however, the journal offers also an editing service that is paid with the publication fees we therefore believe that the journal will take care of this.*

**References**

Romps, D. M., Seeley, J. T., Vollaro, D., and Molinari, J.: Projected increase in lightning strikes in the United States due to global warming, Science, 346, 851–854, https://doi.org/10.1126/science.1259100, http://www.sciencemag.org/cgi/doi/10.1126/science.1259100, 2014.